# CRITICEVAL: Evaluate Large Language Model as Critic

**Tian Lan**[1*]    **Wenwei Zhang**[2*]    **Chen Xu**[4]    **Heyan Huang**[1]
**Dahua Lin**[2,3,5]    **Kai Chen**[2†]    **Xian-Ling Mao**[1†]

[1]School of Computer Science and Technology, Beijing Institute of Technology
[2]Shanghai AI Laboratory    [3]MMLab, The Chinese University of Hong Kong
[4]Key Laboratory of Brain Health Intelligent Evaluation and Intervention,
Ministry of Education, Beijing Institute of Technology    [5]CPII under InnoHK
https://github.com/open-compass/CriticEval

## Abstract

Critique ability, *i.e.,* the capability of Large Language Models (LLMs) to identify and rectify flaws in responses, is crucial for their applications in self-improvement and scalable oversight. While numerous studies have been proposed to evaluate critique ability of LLMs, their comprehensiveness and reliability are still limited. To overcome this problem, we introduce CRITICEVAL, a novel benchmark designed to comprehensively and reliably evaluate critique ability of LLMs. Specifically, to ensure the comprehensiveness, CRITICEVAL evaluates critique ability from four dimensions across nine diverse task scenarios. It evaluates both scalar-valued and textual critiques, targeting responses of varying quality. To ensure the reliability, a large number of critiques are annotated to serve as references, enabling GPT-4 to evaluate textual critiques reliably. Extensive evaluations of open-source and closed-source LLMs first validate the reliability of evaluation in CRITICEVAL. Then, experimental results demonstrate the promising potential of open-source LLMs, the effectiveness of critique datasets and several intriguing relationships between the critique ability and some critical factors, including task types, response qualities and critique dimensions.

## 1 Introduction

Critique ability is crucial for the self-improvement [1, 2] of LLMs, as it enables the effective analysis and correction of flaws in responses [3, 4]. This capability also facilitates a more robust framework, *i.e.*, scalable oversight [5, 6], for ensuring the AI systems growing in scale and capability remain aligned with human-desired outcomes and ethical standards.

So far, while numerous works have been proposed to evaluate critique ability of LLMs in downstream tasks, like common NLP tasks [7, 8] and reasoning tasks [4, 9], their comprehensiveness and reliability are limited. Specifically, existing works [10, 11, 12] typically evaluate only specific aspects of critique ability, resulting in limited evaluated critique dimensions [13, 11, 12, 10, 14], insufficient analysis of response qualities and task types [4, 9]. Besides, while GPT-4 is frequently used to evaluate textual or natural language critiques [3, 10, 13], its reliability across all critique dimensions and tasks remains unverified [10, 15]. In summary, a comprehensive and reliable benchmark for assessing critique capability of LLMs is still under-explored, significantly impeding the in-depth analysis.

---

[*]    Equal contributions
[†]    Corresponding author

38th Conference on Neural Information Processing Systems (NeurIPS 2024).

Table 1: Comparison between the **test and dev set** of benchmarks and CRITICEVAL. A complete list are described in Appendix C. The response quality in some benchmarks is unclassified (-). PR denotes the Pass Rate on reasoning and coding tasks. Our concurrent works are marked with a †.

| Benchmarks | Critique Format | Critique Dimension | Response Quality | Test NL Data Size | Subjective Metric | Objective Metric | Human Anno. | Release |
|---|---|---|---|---|---|---|---|---|
| CRITICBENCH [9] | Scalar | 1 | 2 | 0 | - | Accuracy | ✗ | ✗ |
| Shepherd [10] | NL | 1 | - | 352 | GPT-4 | - | ✗ | ✗ |
| Auto-J [11] | NL/Scalar | 2 | - | 232 | GPT-4 | Accuracy | ✗ | ✓ |
| UltraFeedback [12] | NL | 1 | - | 450 | GPT-4 | - | ✗ | ✗ |
| CRITICBENCH [16] † | Scalar | 2 | 2 | 0 | - | F1,PR | ✓ | ✓ |
| MetaCritique [13] † | NL | 1 | - | 300 | GPT-4 w/ Ref. | - | ✓ | ✓ |
| SummEval [8] | Scalar | 1 | - | 0 | - | Correlation | ✓ | ✓ |
| WMT-22 (zh-en) [7] | Scalar | 1 | - | 0 | - | Correlation | ✓ | ✓ |
| MT-Bench [17] | Scalar | 1 | - | 0 | - | Accuracy | ✓ | ✓ |
| CRITICEVAL (Ours) | NL/Scalar | 4 | 4 | 3,608 | GPT-4 w/ Ref. | Correlation,PR | ✓ | ✓ |

To fill this gap, we propose a novel benchmark, CRITICEVAL, designed to comprehensively and reliably measure critique capability of LLMs. Specifically, to ensure comprehensiveness, CRITICEVAL evaluates critique ability of LLMs from following dimensions: evaluating a single response (feedback), comparing pairs of responses (comparison), correcting the response based on feedback (correction) and evaluating one feedback of LLM (meta-feedback). These critique dimensions cover all categories of critiques in previous works [11, 13, 4] and the necessary capabilities for self-improvement of LLM [1] and scalable oversight [6]. These critique dimensions are measured under nine diverse task scenarios, including three common NLP tasks, two alignment tasks, and four math reasoning and coding tasks. Moreover, evaluated responses in each critique dimension and each task are collected using various open-source and closed-source LLMs with different capabilities, with human annotation ensuring varied quality levels. Furthermore, since both the scalar-valued and textual formats of critique are commonly used in these scenarios [3], CRITICEVAL evaluate the critiques in both formats, equipped with objective [18, 19] and subjective [10, 11] evaluations, respectively. Note that scalar-valued critiques typically refer to Likert scores and preference labels, while textual critiques refer to more fine-grained textual analysis about response quality [11, 3]. Overall, as shown in Table 1, CRITICEVAL exhibits significant advantages in comprehensiveness compared to previous benchmarks. It demonstrates great diversity in critique formats, critique dimensions, response qualities and the data size of textual critique.

To ensure the reliability of evaluating textual critiques in CRITICEVAL, a large number of high-quality critiques are annotated, serving as references for GPT-4 to evaluate textual critiques automatically. To annotate these textual critiques efficiently, we employ a human-in-the-loop pipeline [20], first generated by GPT-4 and then rigorously reviewed and refined by human experts.

Extensive evaluations of 35 widely used open-source and closed-source LLMs prove the reliability of CRITICEVAL. Specifically, GPT-4 with human-annotated reference critiques achieves close correlations with human judgments, while removing them results in significant performance loss. Additionally, critiques with higher scores consistently lead to superior improvements, illustrating a clear correlation between the real critique ability of LLMs and their evaluation scores within CRITICEVAL. Then, extensive evaluations results also demonstrate that some open-source LLMs, such as Qwen [21] and InternLM2 [22], are approaching state-of-the-art closed-source LLMs in critique capabilities, and their critique ability could be further improved through scaling strategy. Besides, the effectiveness of critique datasets is also validated. Finally, these evaluation results also reveal several intriguing phenomena:

- Critique difficulty varies by task type. For instance, math reasoning and coding tasks are more challenging for feedback, comparison, while they are easier for meta-feedback.

- There is an inverse relationship between the quality of critiques and responses. For example, high-quality responses pose a greater challenge to critique effectively.

- Critique difficulty correlates with the critique dimensions; notably, comparison and meta-feedback dimensions present greater challenges than feedback dimension.

These observations and phenomena promote an in-depth understanding of critique ability of LLMs. We hope the discoveries could spur future research in this field.

## 2 Related Work

### 2.1 Application of Critique Ability

**Automatic Evaluation** Automatic evaluation, also known as critique ability in recent works [3, 10], has been well studied in the past few years [23, 18]. It aims to accurately judge the evaluated responses in numerous NLP tasks and reduce the high cost of human annotations [18, 17, 24, 25, 26]. Recently, advanced LLMs, like GPT-4, have exhibited very close correlation with human judgments [18, 27, 13], assign textual critiques with corresponding quality scores, *i.e.,* the scalar-valued score, in a chain-of-thought inference manner [28]. To further mitigate the high inference cost of closed-source LLMs, numerous works propose to improve critique ability of open-source LLMs by fine-tuning them on critique datasets generated by GPT-4 [14, 29, 30], like Auto-J [11] and UltraFeedback [12].

**LLM Self-improvement** So far, critique ability has been widely used for self-improvement of LLMs in two stages: (1) **Inference stage**: Given textual critiques that analyze the flaws in the response and provide suggestions, LLMs can iteratively improve the response quality [6, 15, 31, 32, 33].; (2) **Training stage**: Scalar-valued critiques are frequently used to compile responses with a clear performance gap for rejective fine-tuning (RFT) or preference learning (RLHF [34]), which further enhances LLM capabilities [1, 2, 5, 35, 36]. For instance, Self-rewarding [1] improves Llama-2-70B by fine-tuning it on samples selected based on its rewards. Similarly, ChatGLM-Math [36] fine-tunes a math-critique model for scoring generated answers, which are used for rejective fine-tuning [37] and direct preference optimization [38].

### 2.2 Benchmarking Critique Ability

So far, numerous meta-evaluation benchmarks have been proposed to evaluate the critique ability of models [18]. Early benchmarks mainly focus on evaluating scalar-valued critiques [30, 24] on common NLP tasks [18, 23], like translation [7] and summary [8] by computing the correlations between model and human judgments. Recent works also assess scalar-valued critiques of LLMs on reasoning and coding tasks [24, 9]. For example, CRITICBENCH [9] built from 3 reasoning tasks, analyzes important properties of critique ability of LLMs. Our concurrent work, CRITICBENCH [4] analyzes several intriguing findings among generation, critique and correction capability on responses collect from five reasoning tasks.[3] Compared with these existing works, our proposed CRITICEVAL exhibits advantages on several crucial factors, like critique dimensions, response qualities and diverse task types, leading to more comprehensive evaluations for critique ability. Although CRITICBENCH [9] collect high-quality responses, their quality levels are still unclassified.

Beyond scalar-valued critiques, evaluating textual critiques is more challenging [10, 15]. Most existing works coarsely evaluated textual critiques using GPT-4 [11, 12], proven unreliable [10, 13]. Unlike them, our extensive results prove that GPT-4 with human-annotated critiques is reliable for evaluating textual critiques. Although our concurrent work, MetaCritique [13], demonstrates the reliability of evaluating textual critiques by verifying their Atomic Information Units, it is unclear whether their conclusions could be extended to more critique dimensions and tasks.

## 3 Preliminaries

We first formally define the key concepts and their corresponding notions in CRITICEVAL. Figure 1 shows a specific case to understand these concepts.

**Task Input ($I$) and Response ($R$)** represent the queries and generations of LLMs, respectively.

**Critique** aims to analyze and refine the generated responses. Formally, this paper studies the critique capabilities in four dimensions: (1) feedback $F_s$ involves textual analysis and a quality score. Good feedback should not only find flaws in responses but also provide helpful suggestions for correction [6]; (2) correction or refinement $CR$ aims to revise responses with or without feedback. Previous evaluations [11, 12, 9] overlook this dimension, although it is an inevitable step when letting the model improve itself [35]; (3) comparison $F_c$ contains a textual critique and a preference label for a pair of responses $(R_a, R_b)$; (4) Meta-feedback $F_s(F_s)$, *i.e.*, the feedback of feedback itself [6], involves a rating score reflecting the quality of $F_s$ and corresponding textual analysis, which is a high-level critique dimension. Such an ability is necessary to improve critique ability of LLMs [1, 13]. Due to the complexity of the meta-feedback dimension, textual critiques are not collected in this paper, and we leave it for future research.

---

[3]This work has the same name as the previous CRITICBENCH [9].

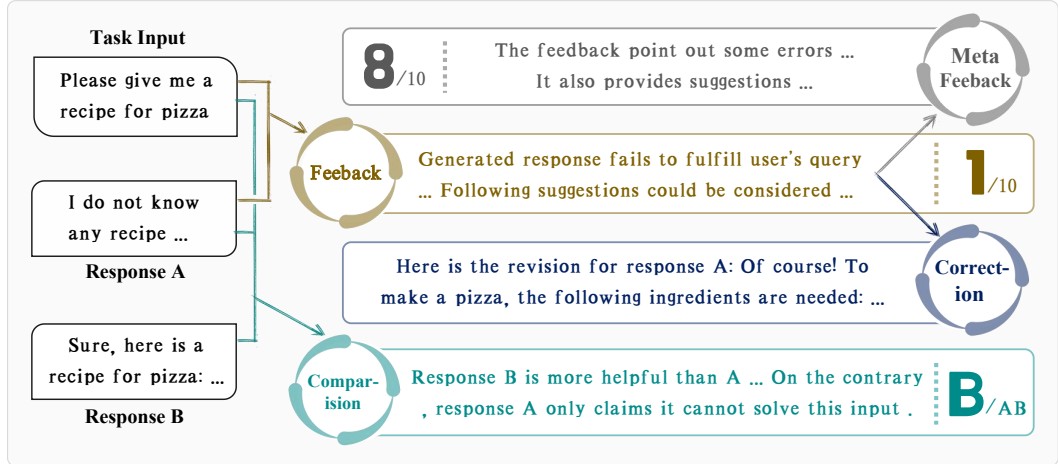

Figure 1: Cases of four critique dimensions. Scalar-valued critiques are scores and preference labels.

To the best of our knowledge, these four critiques cover all categories of critiques examined in prior research [11, 4, 13]. Although the feedback of correction and comparison dimensions are also important, they are not essential for the self-improvement of LLMs. Thus, this study mainly focuses on studying the feedback of feedback $F_s(F_s)$ and leaving the rest of them for our future work.

## 4 CRITICEVAL Construction

Given the challenge of crafting scalar-valued and textual critiques from scratch, we construct CRITICEVAL using a human-in-the-loop data construction pipeline as shown in Figure 2.[4]

### 4.1 Task Input Collection

Task inputs for 9 distinct tasks are collected to evaluate critique capabilities comprehensively (Step 1 in Figure 2). Specifically, CRITICEVAL includes three widely used tasks for evaluating critique ability: (1) representative classical language tasks: summary [39], translation [40], and question-answering [41]; (2) LLM alignment: general chat scenarios [19] and harmlessness cases [35]; (3) reasoning and code capabilities: math reasoning with chain-of-thought (CoT) and program-of-thought (PoT), and coding with and without execution results. We hereinafter refer to "code w/ execution" as "CodeExec" and "code w/o execution" as "CodeNE". For each task, we collect around 100 task inputs from the test sets of some widely used benchmark datasets to ensure the task input quality and avoid data contamination. Please refer to Appendix D for more details about the data source.

### 4.2 Response and Critique to be Evaluated

For each collected $I$ in each task, LLMs of different scales and capabilities are first employed to generate responses with diverse flaws (Step 2 (a) in Figure 2). The complete list of LLMs is in Appendix E. Then, low-, medium-, and high-quality responses with diverse quality differences are collected according to the quality score annotated by the human raters with GPT-4-turbo as the assistant (Step 2 (b)). Moreover, we also collect golden or correct responses, which have been proven challenging for critiques [15]. More details about how to select low-, medium, high-quality and correct responses can be found in Appendix F.

After collecting responses, we further collect critiques to be evaluated for the meta-feedback dimension by utilizing four LLMs that are known powerful for critiques (Step 3 (d) in Figure 2): (1) GPT-4; (2) GPT-3.5-turbo; (3) Auto-J-13B [11]; (4) UltraCM-13B [12].

### 4.3 Reference Critique Generation and Annotation

After collecting task inputs and responses, four kinds of reference critiques are collected on these responses to make the objective and subjective evaluation in our proposed CRITICEVAL more reliable.

**Feedback and Correction**    GPT-4-turbo is utilized to generate feedback and corrections sequentially (Step 3 (c) and (e) in Figure 2). The scalar-valued and textual critiques for feedback dimension

---

[4]More considerations for utilizing human-in-the-loop annotation pipeline are described in Appendix G.

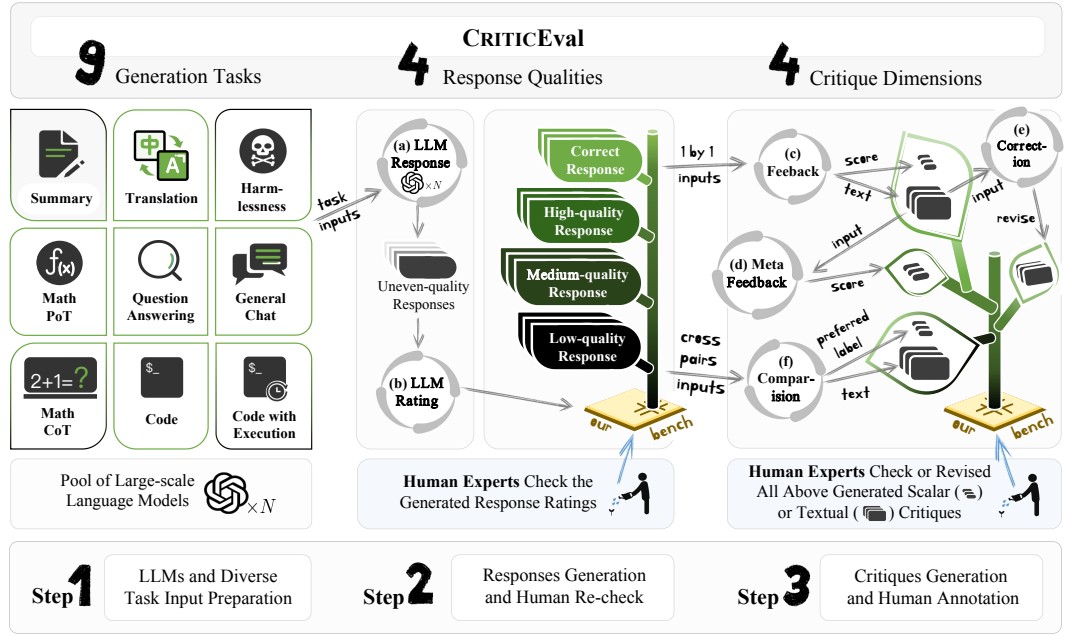

Figure 2: The data construction pipeline for our proposed CRITICEVAL. **Step 1**: 9 tasks and numerous LLMs are prepared. **Step 2**: LLMs are employed to generate responses, which are then meticulously reviewed by human experts. **Step 3**: Critiques are generated by LLMs with strong critique ability, and human experts annotate them.

are collected, denotes as **"score"** and **"text"** in Figure 2. Since responses in math reasoning and coding tasks pose significant challenges for critiques during our annotation, ground-truth answers are provided for GPT-4 as references to generate high-quality feedback and corrections. Then, they are carefully reviewed and revised by human annotators.

**Comparison** Our empirical finding suggests that pairs of responses pose greater challenges for comparison if they perform similarly. Therefore, two kinds of pairs are first created: $(R_{\text{low}}, R_{\text{high}})$ and $(R_{\text{med}}, R_{\text{high}})$, designated as the easy and hard samples, respectively. Then, GPT-4-turbo is prompted to provide scalar-valued and text critiques on these pairs (Step 3 (f) in Figure 2). These outputs, labeled as **"preferred label"** and **"text"** in Figure 2, are then refined by human annotators.

**Meta-Feedback** Since GPT-4 has been proven unreliable to evaluate critiques [10, 15], three human experts are asked to provide their quality scores for generated critiques (Step 3 (d) in Figure 2).

During human annotation, multiple human experts are asked to follow a rigorous annotation protocol, detailed in Appendix H.1, and the statistics of human annotation for reference critiques are described in Appendix H.6. Besides, several case studies are shown in Appendix I to facilitate a clear understanding of our proposed CRITICEVAL.

## 5 Evaluation Metrics

### 5.1 Objective Evaluation

**Feedback and meta-feedback evaluation** aim to evaluate the consistency between generated scores and human judgments. This setup facilitates the generation in a chain-of-thought manner, followed by the quality score of the evaluated critiques. For the meta-feedback dimension, LLMs are prompted with annotated reference critiques. The widely-used Spearman correlations [42] are computed [23, 24], which ranges from $-1$ to $1$ (normalize to $(-100, 100)$). Higher scores indicate a higher consistency with human judgments. The $p$-value of spearman correlation are recorded, and $p < 0.05$ is typically considered to be statistically significant [23, 43].

**Comparison evaluation** assesses the accuracy of LLM in deciding preferences between two responses. It is well known that current LLMs exhibit significant **positional bias** [17, 44, 45], *i.e.,* LLMs tend to prefer responses based on their specific position in the prompt. We implement a rigorous verification process to mitigate the effects of positional bias. Specifically, given responses $R_a$ and $R_b$ to be compared, we obtain the comparison based on two orders, noted as $F_c^a = F_c(R_a, R_b)$

Table 2: Subjective and objective evaluation results on the test set of CRITICEVAL. **Dark gray and shade gray in this and the following tables highlight the best and worst performance.** Objective feedback and meta-feedback scores with $> 0.05$ $p$-value are marked with †. $F_s, CR, F_c, F_s(F_s)$ represent feedback, correction, comparison and meta-feedback critique dimensions, respectively. **Overall** column denotes the overall score over multiple critique dimensions.

| Models | Subjective Evaluation | | | | Objective Evaluation | | | | |
|---|---|---|---|---|---|---|---|---|---|
| | $F_s$ | $CR$ | $F_c$ | Overall | $F_s$ | $CR$ | $F_c$ | $F_s(F_s)$ | Overall |
| *Closed-source LLM* | | | | | | | | | |
| **GPT-4-turbo** | 7.84 | 7.69 | 7.89 | 7.81 | 63.54 | 69.67 | 57.33 | 62.90 | 72.55 |
| **GPT-3.5-turbo** | 5.21 | 7.55 | 4.92 | 5.89 | 51.44 | 64.00 | 40.67 | 28.71 | 60.83 |
| **Claude-instant-1** | 5.88 | 7.72 | 5.76 | 6.45 | 42.78 | 50.00 | 44.89 | 38.89 | 58.93 |
| *Open-source Qwen Series LLMs [47]* | | | | | | | | | |
| **Qwen-72B-Chat** | 5.57 | 7.45 | 5.02 | 6.01 | 42.64 | 54.67 | 44.00 | 27.86 | 58.48 |
| **Qwen-14B-Chat** | 4.81 | 7.25 | 3.98 | 5.35 | $14.32^{\dagger}$ | 38.00 | 15.78 | $10.72^{\dagger}$ | 41.58 |
| **Qwen-7B-Chat** | 4.05 | 6.38 | 3.47 | 4.63 | $-8.09^{\dagger}$ | 32.33 | 5.33 | $11.73^{\dagger}$ | 34.87 |
| *Open-source InternLM2 Series LLMs [48]* | | | | | | | | | |
| **InternLM2-20B** | 6.03 | 7.48 | 5.10 | 6.20 | 58.61 | 50.50 | 44.67 | $3.95^{\dagger}$ | 56.61 |
| **InternLM2-7B** | 5.20 | 7.17 | 4.62 | 5.66 | 49.09 | 36.17 | 23.78 | $3.17^{\dagger}$ | 46.52 |
| *Open-source Mistral Series LLMs [49]* | | | | | | | | | |
| **Mixtral-8x7B** | 5.31 | 7.33 | 4.62 | 5.75 | 51.00 | 43.34 | 43.78 | 26.66 | 56.49 |
| **Mistral-7B** | 4.70 | 7.20 | 4.28 | 5.39 | 43.66 | 38.17 | 27.88 | 31.68 | 50.93 |
| *Open-source Llama-2 Series LLMs [37]* | | | | | | | | | |
| **Llama2-70B-Chat** | 4.12 | 7.11 | 3.95 | 5.06 | 32.79 | 42.34 | 21.11 | 28.32 | 48.50 |
| **Llama2-13B-Chat** | 3.70 | 7.11 | 3.32 | 4.71 | 30.61 | 24.67 | 22.67 | 31.02 | 44.54 |
| **Llama2-7B-Chat** | 3.44 | 6.02 | 3.21 | 4.22 | 20.81 | 21.00 | 5.33 | $5.67^{\dagger}$ | 34.89 |

and $F_c^b = F_c(R_b, R_a)$. The objective scores are computed by: $s = \frac{1}{N}\sum_{i=1}^{N} \mathbb{1}(L(F_c^a, F_c^b))$, where $\mathbb{1} \to \{0, 1\}$ is the indicator function. $L(F_c^a, F_c^b)$ is true if and only if $F_c^a \neq F_c^b$ and $F_c^a, F_c^b$ align with ground-truth preference label. $N$ is the number of test samples.

**Correction evaluation** is only conducted on math reasoning and coding tasks since the revision could be easily verified with the ground-truth answers and the test cases [4, 9]. Thus, the objective evaluation metric is implemented as the pass rate: $\frac{N_{\text{Pass}}}{N}$, where $N$ and $N_{\text{Pass}}$ are the number of the total samples and passed samples, respectively.

### 5.2 Subjective Evaluation

The subjective evaluation aims to evaluate the quality of the generated textual critiques. Since responses in math reasoning and coding tasks can be verified, we only conduct subjective evaluations on other 5 tasks for the correction dimension. In our work, GPT-4 evaluates the generated critiques by generating the chain of thought followed by the score, with our human-annotated critiques as references. It is well-known that LLMs prefer longer generations during their automatic evaluation [46, 45]. However, Figure 7 in Appendix J proves that there is no clue that GPT-4, with our concise and precise reference critiques as input, would give higher scores to longer critiques. The subjective scores range from 1 to 10. Following previous work [20], the human-annotated reference critiques are anchored to 8, serving as a relative scoring pivot.

### 5.3 Overall Score

The overall score of subjective and objective evaluation is computed as averaging on all the critique dimensions, and more details about computing the overall score can be found in Appendix K.

## 6 Evaluation and Analysis

The critique abilities of representative LLMs are analyzed in this section, and the overview results are shown in Table 2. Firstly, the reliability of evaluation in CRITICEVAL are proven in Section 6.2

and Section 6.3. Then, overall analysis is described in Section 6.4. Furthermore, several intriguing phenomena about some critical factors are described, including task types (Section 6.5), the response quality (Section 6.6) and the critique dimensions (Section 6.7). Finally, we elaborate and analyze the fine-grained error patterns of model-generated critiques in Section 6.8. The complete experimental results of all evaluated LLMs on the test/dev set for each task and each critique dimension are placed in Appendix L.

## 6.1 LLMs to be Evaluated

35 widely used open-source and closed-source LLMs of different sizes are evaluated on CRITICEVAL, including the instruction-tuned LLMs [21, 50, 22, 51], critique-tuned LLMs that fine-tuned on critique datasets generated by GPT-4 [11, 12, 30], and reward models [12, 52, 53]. Please refer to Appendix N for all evaluated LLMs and the inference details. The prompt templates for LLMs on critique dimensions are shown in Appendix I with score rubrics listed in Figure 18 in Appendix H.3.

## 6.2 Reliability of Subjective Evaluation in CRITICEVAL

As LLMs are prompted with human-annotated critiques, their performance in meta-feedback could reveal their reliability for evaluating generated textual critiques. As shown in Table 2 ($F_s(F_s)$ column) and Table 3, GPT-4-turbo achieves very high correlations (62.90, and 66.18) with human judgment. Although there is still a gap compared to the average human level ($66.18 < 79.03$), the strong correlations ensure the reliable evaluation for tex-

Table 3: Results of meta-feedback dimension in CRITICEVAL dev set. $p$-value $< 0.05$.

| Models | $F_s(F_s)$ | $F_s(F_s)$ w/o ref. |
|---|---|---|
| **GPT-4-turbo** | **66.18** | **47.26 (-18.92)** |
| **Qwen-1.5-72B** | 38.97 | 22.35 (**-16.62**) |
| **Claude-instant-1** | 36.88 | 19.88 (**-17.00**) |
| **GPT-3.5-turbo** | 17.28 | 16.38 (**-0.90**) |

tual critique ability [23]. Furthermore, we also conduct the ablation study to prove the contribution of our human-annotated reference critiques. As shown in Table 3, it can be found that all LLMs perform worse when the reference critiques (**ref.**) are removed (average -13.36 performance decrease), proving their significant contribution for reliable subjective evaluation in our proposed CRITICEVAL.

Moreover, except for the feedback critique dimension, we also test the reliability of subjective evaluation on the correction and comparison critique dimensions. Specifically, we ask three human annotators to annotate the quality score of 450 critiques generated by five representative LLMs (GPT-3.5-turbo, Qwen-72B-Chat, InternLM2-20B-Chat, Mistral-7B and ChatGLM3-6B) from 9 tasks in CRITICEVAL, and all the human annotators

Table 4: Correlations in $CR$ and $F_c$ dimensions. $p$-value $< 0.05$.

| - | $CR$ | $F_c$ |
|---|---|---|
| **Human Avg.** | 87.04 | 76.55 |
| **GPT-4 w/ ref.** | 82.10 | 70.27 |

are guided by the same evaluation protocol in our subjective evaluation. The results are shown in Table 4. It can be found that GPT-4-turbo, with our human-annotated critiques as references, could achieve a very strong correlation with human judgments, close to the average human level. This observation proves the robust and reliable subjective evaluation of the textual critiques in the correction and comparison dimensions. Besides, the correlation scores on the correction and comparison critique dimension are higher than the feedback dimension. This phenomenon suggests that the feedback of the feedback is more challenging than the feedback of correction and comparison.

## 6.3 More Effective Critiques Consistently Lead to Superior Corrections

Table 5: The quality of corrections $CR$ increases as the quality of feedback increases.

| Models | Source of Feedbacks | Objective | | Subjective | |
|---|---|---|---|---|---|
| | | $F_s$ | $CR$ | $F_s$ | $CR$ |
| **InternLM2-20B-Chat** | **Llama2-70B-Chat** | 2.24 | 7.15 | 5.63 | 5.71 |
| **InternLM2-20B-Chat** | **InternLM2-20B-Chat** | 7.53 | 10.33 | 6.85 | 5.80 |
| **InternLM2-20B-Chat** | **Human-Annotated** | 8.00 | 50.50 | 8.00 | 7.48 |
| **Llama2-70B-Chat** | **Llama2-70B-Chat** | 2.24 | 5.33 | 5.63 | 5.54 |
| **Llama2-70B-Chat** | **InternLM2-20B-Chat** | 7.53 | 12.47 | 6.85 | 6.32 |
| **LLama2-70B-Chat** | **Human-Annotated** | 8.00 | 42.34 | 8.00 | 7.11 |

Although the reliability of subjective evaluation has been proven in Section 6.2, it is still unknown whether **real feedback critique ability of LLMs is consistent with the evaluation results in CRITICEVAL**. To explore this, we prompt the InternLM2-20B-Chat and Llama2-70B-Chat models to revise responses from CRITICEVAL using three sources of feedback with varying quality levels. As

illustrated in Table 5, a clear and consistent trend emerges: as the quality of the feedback increases, both the objective and subjective revision performance improves. This finding underscores that real critique ability of LLMs aligns closely with the evaluation results in our proposed CRITICEVAL, *i.e.,* critiques of LLMs with higher scores are more accurate and effective for corrections.

In summary, the reliability of evaluation in CRITICEVAL has been well proven. Following sections will describe the overall analysis and relationships between critique ability and several crucial factors.

## 6.4 Overall Analysis of LLMs

As shown in Table 2, GPT-4 significantly outperforms other LLMs on most critique dimensions, while slightly underperforms our human-annotated critiques ($7.81 < 8$). Surprisingly, open-source LLMs are approaching state-of-the-art closed-source LLMs. For example, InternLM2-20B-Chat surpasses GPT-3.5-turbo on overall subjective scores ($6.20 > 5.89$). Furthermore, there is a clear relationship that the critique ability of LLMs improves steadily as the number of parameters increases (Table 2), suggesting that the critique ability of LLMs highly correlates with their capability.

Table 6: Critique-tuned LLMs results in feedback dimension.

| Models | Sub. | Obj |
|---|---|---|
| Llama-2-13B | 3.70 | 30.61 |
| Llama-2-70B | 4.12 | 32.79 |
| Auto-J-13B | **4.21** | **36.05** |
| UltraCM-13B | 4.12 | 21.51 |
| TigerScore-13B | 3.31 | 17.87 |

We also provide a clear diagram to show this relationship in Figure 14 in Appendix R. Beyond the average scores, we also categorize the textual critiques of LLMs into multiple quality intervals for more interpretable analysis, which are described in Appendix O.

The results of critique-tuned LLMs in the feedback dimension on the test set are shown in Table 6. It can be found that critique-tuned LLMs fine-tuned from Llama-2-13B significantly outperform even the Llama-2-70B-Chat model, proving the effectiveness of critiques datasets [11, 12]. The results of representative reward models are shown in Table 7. From these results, it can be found that reward models like UltraRM-13B achieve impressive performance in scoring the quality of responses, significantly outperforming GPT-3.5-turbo. This observation aligns with findings in recent works [14].

Table 7: Reward model objective results in $F_s$ and $F_c$ dimensions.

| Models | $F_s$ | $F_c$ |
|---|---|---|
| GPT-3.5-turbo | 51.44 | 40.67 |
| UltraRM-13B | **52.33** | **54.67** |
| Ziya-7B | 25.81 | 40.00 |

## 6.5 Relationship with Task Type

Effective critiques usually require domain knowledge and understanding of given tasks. We analyze the relationship between critique ability and task type in Table 8, which shows the average performance of all evaluated LLMs.

**Feedback, Comparison** LLMs achieve much higher scores in the first five tasks than on math reasoning and coding tasks, indicating math reasoning and code tasks are more challenging.

**Meta-Feedback** LLMs achieve much higher consistency with human judgments on code and math reasoning tasks, indicat-

Table 8: Two **Avg.** rows represent the average scores of all LLMs on the first 5 tasks and the last 4 tasks.

| Tasks | $F_s$ | | $F_c$ | | $CR$ | | $F_s(F_s)$ |
|---|---|---|---|---|---|---|---|
| | Sub. | Obj. | Sub. | Obj. | Sub. | Obj. | Obj. |
| Translate | 4.43 | 31.14 | 3.78 | 18.28 | 5.31 | - | -2.93 |
| Chat | 5.09 | 20.60 | 4.97 | 32.60 | 5.66 | - | 1.80 |
| QA | 5.20 | 30.75 | 5.05 | 27.67 | 6.42 | - | 13.50 |
| Summary | 4.76 | 28.93 | 4.63 | 37.12 | 5.99 | - | 0.54 |
| Harmless. | 5.12 | 25.04 | 3.97 | 19.35 | 7.51 | - | 2.71 |
| Avg. | **4.92** | **27.29** | **4.48** | **27.00** | 6.18 | - | 3.12 |
| MathCoT | 3.55 | 22.56 | 2.80 | 12.42 | - | 29.36 | 19.63 |
| MathPoT | 3.35 | 27.80 | 3.05 | 14.98 | - | 24.98 | 22.73 |
| CodeExec | 3.07 | 13.38 | 2.74 | 7.72 | - | 32.20 | 25.50 |
| CodeNE | 2.77 | 10.37 | 2.80 | 10.33 | - | 29.50 | 24.38 |
| Avg. | 3.19 | 18.53 | 2.85 | 11.36 | - | 29.01 | **23.06** |

ing that evaluating textual critiques in math reasoning and code tasks is more reliable.

**Correction** Math reasoning tasks are more challenging than coding tasks, and CodeExec is easier to revise than CodeNE due to the richer information in execution results. Except for math reasoning and coding tasks, the translation is the most challenging task because professional domain knowledge is required, while harmlessness is the easiest to refine since most LLMs have been trained to avoid harmful generations [35]. Furthermore, we explore the variance in correction quality on reasoning and coding tasks (**Obj.** in Table 9)

Table 9: Average performance of evaluated LLMs on test set.

| Dimension | Sub. | Obj. |
|---|---|---|
| $CR$ w/ HF | **7.12** | **43.66** |
| $CR$ w/ SF | 5.48 | 13.01 |
| $CR$ w/ EF | 5.16 | 14.44 |

and other subjective tasks (**Sub.** in Table 9).[5] Specifically, three kinds of feedback are used for correction: (1) **H**uman-annotated **F**eedback (**HF**); (2) **E**mpty **F**eedback (**EF**), where LLMs are prompted to

---

[5]Subjective tasks represent five tasks (translation, general chat, QA, summary and harmlessness).

improve responses without any feedback; and (3) LLMs **S**elf-generated **F**eedback (**SF**). As shown in Table 9, it can be found that self-generated feedback is beneficial to corrections on subjective evaluation (HF(7.12) > SF(5.48) > EF(5.16)), while it might negatively affect corrections on objective evaluation of math reasoning and coding tasks (SF(13.01) < EF(14.44) < HF(43.66)). This observation proves that LLMs struggle in self-improvement on challenging reasoning tasks, aligning with recent findings [54, 4].

## 6.6 Relationship with Response Quality

Before analyzing the relationship between response quali-
ties and critique ability, it is essential to categorize the error
patterns in responses. We highlight that the error patterns
are related to the task type, complicating the classification
of errors. To conduct a representative analysis of errors in
all tasks, human annotators are asked to categorize errors
into three patterns, which collectively encompass nearly

Table 10: Error pattern distribution (%).

| Error Pattern | Low | Med. | High |
|---|---|---|---|
| Obvious | **74.68** | 29.48 | 20.42 |
| Complex | 16.46 | **45.51** | 31.69 |
| Subtle | 8.86 | 25.00 | **47.89** |

all the cases: (1) **Obvious error** is easy to critique and correct, like apparent misuses of words in translation task; (2) **Complex error** is challenging to correct, regardless of whether critiques are easy to critique, like logical reasoning error in reasoning tasks; (3) **Subtle error** is hard to critique, while it is usually easier to revise than complex error, like slight misunderstandings of context in general chat. The distribution presented in Table 10 reveals distinct primary errors across different response qualities. More details about these error patterns in each task are described in Appendix P.

Given the distribution of error patterns, we ana-
lyze critique ability of LLMs on responses with
varying qualities. As shown in Table 11, high-
quality responses are the hardest for feedback
since they contain lots of subtle errors (Table
10). Note that the medium-quality responses
have higher objective feedback scores than low-
quality ones, which is inconsistent with our ex-
pectations. This phenomenon is because low-

Table 11: Average performance of LLMs on the different response qualities (test set).

| Quality | Subjective | | Objective | | |
|---|---|---|---|---|---|
| | $F_s$ | $CR$ | $F_s$ | $CR$ | $F_s(F_s)$ |
| Low | **5.14** | **7.17** | 21.93 | **46.04** | 22.73 |
| Medium | 4.76 | 7.08 | **23.10** | 40.58 | 19.78 |
| High | 4.66 | 7.15 | 20.62 | 45.19 | **28.84** |

quality responses often receive very low human-annotated quality scores (near 1), while the scoring of LLMs tends to be higher, leading to a discrepancy. For the correction dimension, low-, and high-quality responses are easier to correct than medium-quality due to the most obvious and subtle errors. There are two kinds of qualities for comparison dimension: easy and hard. Most LLMs perform better on easy samples than on hard samples. Specifically, the subjective and objective scores of easy samples are 4.78 and 39.73, respectively, higher than those of hard samples (4.55 and 29.80). For the meta-feedback dimension, LLMs achieve the highest consistency with human judgments on high-quality responses while performing worst on medium-quality responses.

## 6.7 Relationship with Critique Dimensions

The average scores of all evaluated LLMs on different critique dimensions
are shown in Table 12. Objective scores of comparison and correction
are not recorded because they are not correlations. Several conclusions
can be made: (1) correction is the easiest critique dimension, followed
by feedback, and then comparison. This observation demonstrates that
comparison requires accurate analysis of both responses, which is more
complex than the feedback dimension; (2) As a high-level critique dimen-
sion, meta-feedback is more challenging than the feedback.

Table 12: Average performance on test set.

| Dimen. | Sub. | Obj. |
|---|---|---|
| $F_s$ | 4.89 | **35.75** |
| $F_c$ | 4.58 | - |
| $F_s(F_s)$ | - | 22.97 |
| $CR$ | **7.12** | - |

## 6.8 Fine-grained Failure Modes in Model-Generated Critiques

This section analyzes the fine-grained failure modes in model-generated critiques across feedback, comparison and correction dimensions. As illustrated in Table 13, human annotators summarize the 12 main failure modes in model-generated critiques. Then, we compute the distribution of these failure modes of all evaluated LLMs. Figure 3 demonstrate that the most frequent failure modes are missing errors (E1, E2), lacing effective comparison analysis (E7) and worse revision than references (E10) for feedback, comparison and correction dimensions, respectively. Furthermore, as shown in Figure 4, it can be observed that missing errors/suggestions (E1, E2) and inaccurate critiques (E3, E4, E8) usually lead to lower subjective scores.

Table 13: Definition of Failure Modes in Feedback, Comparison and Correction critique dimensions. **E1-E6** denotes the **shared** failure modes of feedback and comparison dimensions, and **E7-E8** belong to comparison dimension. **E9-E11** belong to the correction dimension.

| Critique Dimensions | Failure Mode | Description of Failure Mode |
|---|---|---|
| **Feedback and Comparison** | E1 | Feedback misses some errors. |
| | E2 | Feedback misses revision suggestions or suggestions are low-quality. |
| | E3 | Feedback incorrectly analyzes correct content as erroneous. |
| | E4 | Feedback content contains errors. |
| | E5 | Feedback is correct but complex. |
| | E6 | Feedback is not concise, repetitive or irrelevant. |
| **Comparison** | E7 | Critiques lack effective analysis between two responses. |
| | E8 | Preference between two responses is wrong. |
| **Correction** | E9 | Revision does not follow suggestions in feedback well. |
| | E10 | Revisions are better but have not reached the reference. |
| | E11 | There are some errors in revisions. |
| **-** | **Other** | Other Cases |

Distribution of Failure Modes in Three Critique Dimensions

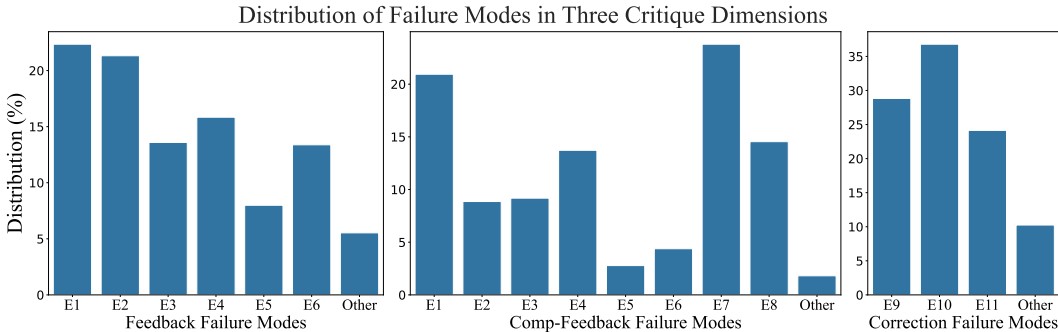

Figure 3: Distribution of failure modes in each critique dimension.

Average Subjective Scores (1-10) of Failure Modes

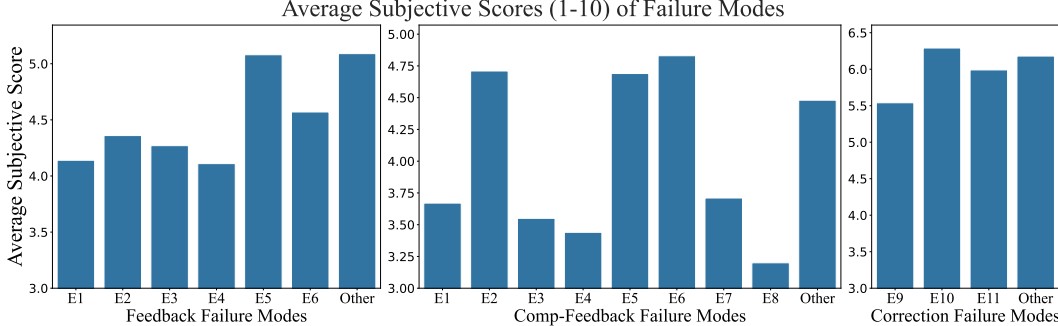

Figure 4: Average subjective score of failure modes in each critique dimension.

## 7 Conclusion and Future Work

In this paper, we introduce a comprehensive and reliable benchmark for evaluating the critique abilities of LLMs, named CRITICEVAL. Extensive experimental results first prove the reliability of CRITICEVAL, and reveal the promising potential of open-source LLMs, the effectiveness of critique datasets and intriguing relationships between critique capabilities and some factors: task types, response qualities and critique dimensions. These observations significantly promote an in-depth understanding of the critical ability of LLMs and LLM's self-improvement. In the future, we plan to enhance our benchmark in several key areas: (1) Broadening the scope to include more tasks, such as tool-using; (2) Extending the benchmark to encompass other languages, like Chinese; (3) Improving the subjective evaluation protocol to allow for more fine-grained analysis; (2) Continue to evaluate LLMs and track their critique ability, like Llama-3 models; (5) Improving the quality of reference critiques by incorporating additional high-quality critiques from advanced LLMs if and only if their quality surpasses the existing reference critiques.

## Acknowledgments and Disclosure of Funding

First, the authors would like to express their sincere gratitude to all the anonymous reviewers and meta-reviewers for their insightful comments and constructive feedback. Besides, we would also like to thank many senior researchers for their valuable comments before our submission, which greatly improved the quality of our paper: Leyang Cui, Yan Wang, Yong Hu, Rongcheng Tu, Hongli Mao, Fanshu Sun and Chen Xu. The names are listed in no particular order. Furthermore, this project was funded in part by the Centre for Perceptual and Interactive Intelligence (CPII) Ltd under the Innovation and Technology Commission (ITC)'s InnoHK. Dahua Lin is a PI of CPII under the InnoHK. This work was also supported in part by the National Natural Science Foundation of China (No. 62172039, U21B2009, 62276110, and 62450100), the MIIT Program (CEIEC-2022-ZM02-0247), the Postdoctoral Fellowship Program of CPSF (No. GZC20233403), and the China Postdoctoral Science Foundation (No.2024M764142).

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

# A Limitations

## A.1 Sub-optimal Reference Critiques

Following previous work [20], our work construct CRITICEVAL with the human-in-the-loop annotation pipeline, *i.e.,* multiple human annotators are asked to review and revise the critiques generated by GPT-4 model. Even though we have established a rigorous annotation process to ensure the high quality of annotated critiques, human annotators are inevitably influenced by GPT-4's initial generated critiques in some more open-ended tasks, like general chat and QA tasks. This problem may result in the quality of revised critiques by human annotation still being sub-optimal. To address this problem, we plan to enhance CRITICEVAL in our next version. Specifically, we will replace existing reference critiques with potential better critiques generated by advanced evaluated LLMs, like Claude-3[6], if and only if their qualities surpasses reference critiques.

## A.2 GPT-4 Model for Subjective Evaluation

CRITICEVAL mainly utilizes the advanced GPT-4-turbo model for subjective evaluation. Despite integrating high-quality annotated reference critiques to guide GPT-4 toward more accurate assessments, it's essential to acknowledge that the model's evaluations may not always align perfectly with human judgment. While GPT-4 have yet to reach the level of precision of human annotation - they currently represent the most effective approach for balancing the trade-offs between evaluation cost and quality. It is still a significant challenge to accurately and automatically evaluate critiques across all scenarios. Recognizing this, we aim to address these issues in our future work by progressively refining our benchmark and evaluation protocols.

## A.3 More Tasks for Critique

Compared with existing benchmarks [12, 10, 14, 13], our proposed CRITICEVAL exhibits significant advantages in the diversity of evaluated task scenarios. Although we strive to cover a wide range of diverse generation tasks, there are still some tasks have yet to be considered, such as tool learning [55], knowledge-intensive tasks [56] and hallucination [57]. In our future work, we will continue to include more tasks in the next version of CRITICEVAL.

## A.4 More LLMs to be Evaluated

Some newly proposed LLMs and reward models [58] have not been added yet, like Llama-3[7] and Claude-3 series models. Since our conclusions are summarized from evaluation results of over 35 open-source and closed-source LLMs, lacking these LLMs does not affect conclusions. We will continue to evaluate their critique ability in our future work.

## A.5 Limited Inference Strategy

In this paper, all the evaluated models generate the critiques by using the greedy-search decoding method. There exist some inference strategies to potentially improve the model's performance, like structured generation [59]. The primary goal of our proposed CRITICEVAL in the current stage is to construct a comprehensive and reliable benchmark for evaluating the critique ability of LLMs, and we will explore these inference strategies to improve the critique ability in our future work.

# B Ethical Considerations

Most task inputs in CRITICEVAL are collected from publicly available datasets, free from any possible harm toward individuals or groups. Moreover, humans carefully select and process the responses and critiques generated by LLMs to secure privacy and confidentiality. No personal identification information is involved. However, it should be noted that the task input, responses, and critiques in the Anthropic-HHH dataset [60] of the harmlessness task contain harmful materials and hate

---

[6]https://www.anthropic.com/news/claude-3-family
[7]https://ai.meta.com/blog/meta-llama-3/

speech. Despite the risks involved, it is essential to disclose this research fully, and materials in the Anthropic-HHH dataset have been widely used for safety research in the LLM community. All raters have been paid adequate wages. The hourly wage of our human annotators is about 5.69 USD, which is much higher than average hourly wage 3.13 USD on Amazon Mechanical Turk [61].

## C Comparison and Statistics of CRITICEVAL

The comparison between CRITICEVAL and existing benchmarks can be found in Table 14, which proves the advantages of our proposed CRITICEVAL for critique evaluation. Compared with existing benchmarks for critique evaluation, our proposed CRITICEVAL contains 3,608 textual natural language critique samples (textual critiques) that are well annotated by multiple human annotators, leading to a more stable and reliable assessment in our subjective evaluation. The scale of objective evaluation data (scalar data) in our dataset is second only to Chat Arena[8] and SummEval [8]. However, compared to these two datasets, our dataset contains a more diverse range of critique dimensions and tasks (nine diverse tasks). Moreover, the statistics of CRITICEVAL in the test and dev set are shown in Table 15.

Table 14: Statistics of existing critique benchmarks, meta-evaluation benchmarks (scalar-valued critique evaluation), and CRITICEVAL. NL and Scalar denote natural language feedback and scalar-valued feedback, *i.e.*, the preference label or Likert score [3]. CriticBench [9, 16] contain two kinds of response quality (correct and wrong). The responses in some benchmarks are not unclassified, and we set them as unclassified (-). Scalar-valued critiques in Auto-J [11] are from its Eval-P, and textual critiques are from Eval-C split.

| Dataset | Critique Format | Critique Dimensions | Response Qualities | Test Scalar Data Size | Test NL Data Size | Human Annotation | Released |
|---|---|---|---|---|---|---|---|
| Shepherd [10] | NL | 1 | - | 0 | 352 | ✗ | ✗ |
| UltraFeedback [12] | NL | 1 | - | 0 | 450 | ✗ | ✗ |
| Auto-J [11] | NL / Scalar | 2 | - | 1,392 | 232 | ✓ | ✓ |
| CriticBench [9] | Scalar | 1 | 2 | 3,234 | 0 | ✓ | ✗ |
| CriticBench [16] | Scalar | 2 | 2 | 3,825 | 0 | ✓ | ✓ |
| MetaCritique [13] | NL | 1 | - | 0 | 300 | ✓ | ✓ |
| SummEval [8] | Scalar | 1 | - | 1,600 | 0 | ✓ | ✓ |
| WMT-22 (zh-en) [7] | Scalar | 1 | - | **33,750** | 0 | ✓ | ✓ |
| WebNLG-2020 [62] | Scalar | 1 | - | 2,848 | 0 | ✓ | ✓ |
| AFCE [30] | Scalar | 1 | - | 1,600 | 0 | ✓ | ✓ |
| GSM8K [63, 30] | Scalar | 1 | - | 2,638 | 0 | ✓ | ✓ |
| Just-Eval [64] | Scalar | 1 | - | 4,500 | 0 | ✗ | ✓ |
| OpenMEVA (ROC) [65] | Scalar | 1 | - | 1,000 | 0 | ✓ | ✓ |
| BAGEL [66] | Scalar | 1 | - | 202 | 0 | ✓ | ✓ |
| Commongen [67] | Scalar | 1 | - | 2,796 | 0 | ✓ | ✓ |
| Vicuna Bench [14] | Scalar | 1 | - | 320 | 0 | ✗ | ✓ |
| MT-Bench [14] | Scalar | 1 | - | 320 | 0 | ✗ | ✓ |
| FLASK [68] | Scalar | 1 | - | 2,000 | 0 | ✓ | ✓ |
| FeedBack Bench [14] | Scalar | 1 | - | 1,000 | 0 | ✗ | ✓ |
| CRITICEVAL (Ours) | NL / Scalar | **4** | **4** | 2,892 | **3,608** | ✓ | ✓ |

## D Source Data for Different Tasks

The benchmark includes three representative classical language tasks: summary [39], translation [40], and question-answering [41]. Since a popular application of LLMs is to serve as a chatbot, where alignment is important to ensure the safe application of LLMs, we collect instructions from general chat scenarios [19] and harmlessness cases [35] to evaluate the LLMs' critique ability for alignment. Furthermore, the reasoning and code capabilities are also fundamental for augmenting LLMs as agents [78], another important and promising application of LLMs. Thus, we also collect

---

[8]https://hf-mirror.com/datasets/lmsys/lmsys-arena-human-preference-55k

Table 15: The statistics of the test and dev set in our proposed CRITICEVAL.

| Tasks | Feedback | | | | Comp-Feedback | | | | Correction | | | | Meta-Feedback | | Sum. |
|---|---|---|---|---|---|---|---|---|---|---|---|---|---|---|---|
| | Dev | | Test | | Dev | | Test | | Dev | | Test | | Dev | Test | |
| | Sub. | Obj. | Sub. | Obj. | Sub. | Obj. | Sub. | Obj | Sub. | Obj. | Sub. | Obj. | Obj. | Obj. | |
| Translation | 70 | 90 | 50 | 30 | 60 | 80 | 40 | 20 | 60 | - | 40 | - | 60 | 60 | 660 |
| QA | 70 | 90 | 50 | 30 | 60 | 80 | 40 | 20 | 60 | - | 40 | - | 60 | 60 | 660 |
| Chat | 70 | 90 | 50 | 30 | 60 | 80 | 40 | 20 | 60 | - | 40 | - | 60 | 60 | 660 |
| Summary | 70 | 90 | 50 | 30 | 60 | 80 | 40 | 20 | 60 | - | 40 | - | 60 | 60 | 660 |
| Harmlessness | 70 | 90 | 50 | 30 | 60 | 80 | 40 | 20 | 60 | - | 40 | - | 60 | 60 | 660 |
| MathCoT | 70 | 73 | 50 | 40 | 60 | 80 | 40 | 20 | - | 50 | - | 50 | 72 | 72 | 677 |
| MathPoT | 70 | 51 | 50 | 40 | 60 | 80 | 40 | 20 | - | 50 | - | 50 | 72 | 72 | 655 |
| Code Exec | 70 | 90 | 50 | 40 | 60 | 80 | 40 | 20 | - | 50 | - | 50 | 60 | 60 | 670 |
| Code not Exec | 70 | 90 | 50 | 40 | 60 | 80 | 40 | 20 | - | 50 | - | 50 | 60 | 60 | 670 |

instructions for math reasoning with chain-of-thought and program-of-thought, and coding with and without execution results.

To ensure the difficulty of CRITICE-VAL, we only collect coding and math reasoning questions that some 70B LLMs cannot correctly answer, which is proven effective in previous works [9]. Our motivation is to collect questions that could easily raise responses with diverse flaws. Simple questions pose challenges for us in achieving this goal since most LLMs can easily solve them. Collecting these questions that 70B LLMs cannot answer correctly makes the difficulty of questions become moderate or complex, which aligns with our motivation.

The details of selected datasets for 9 tasks are listed in Table 16, covering the well-known NLP tasks (translation, summary, and question answering), reasoning tasks (mathematics and coding), and alignment (general chat and harmlessness). These datasets' test sets are used for CRITICEVAL construction, avoiding data contamination. For each task, we collect around 100 instructions from the test sets of some widely-used benchmark datasets to ensure the instruction quality and avoid data contamination.

Table 16: Source of 9 tasks in CRITICEVAL. Most tasks contain diverse samples from multiple test sets.

| Tasks | Source From Test Data | Num. | License |
|---|---|---|---|
| Translation | WMT20 MLQE [40] | 100 | Unknown |
| Chat | ChatArena
Alpaca-Eval [19] | 50 each | CC-BY-4.0
CC-BY-NC-4.0 |
| QA | OBQA [41]
CommonQA[69]
PIQA [70] | 35 each | Unknown
MIT
Unknown |
| Harmlessness | HHH [60] | 100 | MIT |
| Summary | Summ. HF [39] | 100 | MIT |
| Math PoT
Math CoT | AquA-RAT [71]
MathQA [72]
GSM8K [73]
NumGLUE [74]
TheoremQA [75] | 20 each | Apache-2.0
Apache-2.0
MIT
Apache-2.0
MIT |
| Code w/. exec
Code w/o. exec | MBPP [76]
HumanEval [77] | 50 each | CC-BY-4.0
MIT |

Our dataset is under Apache 2.0 License.

# E  List of Used LLMs for Response and Critique Generation

Our study uses several LLMs with different capabilities to generate diverse feedback, listed in Table 17. Besides, we also use some critique-tuned LLMs to generate textual feedback, like Auto-J-13B and UltraCM-13B models.

# F  Details of Responses Generation

## F.1  How to Collect Low-, Medium-, High-quality Responses

To identify the quality of these responses efficiently, GPT-4 is utilized to initially assign quality ratings ranging from 1 to 7 (Step 2 (b) in Figure 2) then let human annotators meticulously review and adjust these scores, which are used in the objective evaluation in the feedback dimension (Section 5.1). then, three responses with distinct quality differences for each $I$ are sampled based on their human-verified quality scores, including low-, medium-, and high-quality responses (noted as $R_{\text{low}}, R_{\text{med}}, R_{\text{high}}$, respectively). Due to partial simple or hard queries, there might be queries where the scores of

Table 17: The list of used LLMs for generating responses and critiques.

| LLMs | Source |
|---|---|
| InternLM-7B-8K | https://huggingface.co/internlm/internlm-7b |
| Qwen-7B-Chat | https://huggingface.co/Qwen/Qwen-7B-Chat |
| Qwen-14B-Chat | https://huggingface.co/Qwen/Qwen-14B-Chat |
| Baichuan2-13B | https://huggingface.co/baichuan-inc/Baichuan2-13B-Chat |
| InternLM-20B | https://huggingface.co/internlm/internlm-chat-20b |
| Vicuna-33B-V1.3 | https://huggingface.co/lmsys/vicuna-33b-v1.3 |
| OpenBuddy-70B-V14.3 | https://huggingface.co/OpenBuddy/openbuddy-llama2-70b-v14.3 |
| WizardLM-70B-V1.0 | https://huggingface.co/WizardLM/WizardLM-70B-V1.0 |
| GPT-3.5-Turbo | https://chat.openai.com/ |
| GPT-4 | https://chat.openai.com/ |
| UltraCM-13B | https://huggingface.co/openbmb/UltraCM-13b |
| Auto-J-13B | https://huggingface.co/GAIR/autoj-13b |

three responses are close. However, there is a distinct quality difference between low, medium, and high-quality responses overall.

The statistical of responses' quality scores on 9 tasks can be found in the Figure 5. Figure 5 demonstrates the discernible performance disparities in responses for each task. Since automatic execution leaks quality information, we do not collect the correct responses for the Code-Exec task. Such variation is instrumental in analyzing the impact of response quality on the feedback.

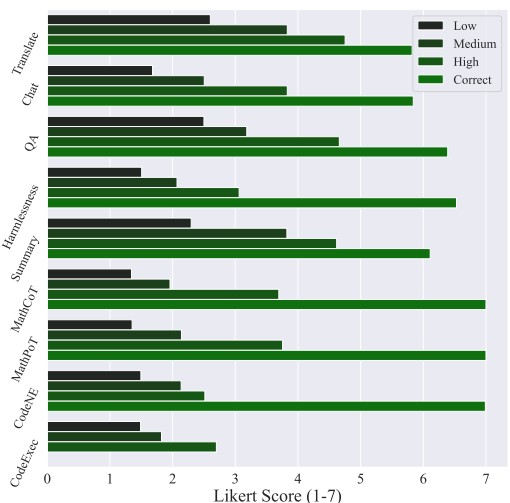

Figure 5: Human annotated Likert scores (1-7).

### F.2 Correct or Golden Response Generation

Golden or correct responses are collected for each task input $I$, which are proven challenging for critiques [10, 15]. We use GPT-4 to generate correct responses using ground-truth rationales or codes as hints for coding and mathematical tasks. Since executions leak information about response quality, correct responses are not collected for the CodeExec task. In tasks beyond coding and mathematics, GPT-4 is prompted to refine its past generations, given its feedback during multiple turns, and the last revision is collected as golden response.

## G Reasons of Utilizing Human-in-the-Loop Data Generation Pipeline

We construct the high-quality reference critiques by using human-in-the-loop data generation pipeline, which is motivated by two essential considerations (effectiveness and efficiency).

**Human-written Critiques are Usually Insufficient - Sub-optimal (Effectiveness)** Our trial human annotation reveals that human annotators might neglect some apparent or severe issues when writing critiques from scratch, consistent with findings in recent studies from OpenAI [79, 6]. Our experimental result in Appendix M demonstrate that neglecting issues usually leads to low-quality critiques. In contrast, despite the possibility of generating wrong critiques, LLMs like GPT-4 offer more comprehensive and detailed critiques [79]. By revising LLM's errors by human experts, the final critiques could be more comprehensive and accurate, leveraging the strengths of both human annotators and LLMs.

**Annotating Challenging Critique Task from Scratch Cost A Lot (Efficiency)** Writting critiques from scratch is a significant challenge [20, 10]. For instance, Shepherd [52] incurred an annotation

cost of 8$ per sample, leading to over 28,864$ and 1,350 work hours to annotate the entire CRIT-ICEVAL, which is unbearable for our project. Thus, we have to employ advanced LLMs to generate draft critiques, followed by human annotation. GPT-4 was chosen because our preliminary studies indicate that it is the most reliable LLM for producing draft critiques, while other LLMs are much worse. Consequently, a diverse set of LLMs introduces more noise in generated critiques, bringing more difficulties to human annotators.

In conclusion, the human-in-the-loop pipeline achieves the trade-off between annotation cost and quality. We promise to add these details to the revised paper to emphasize the motivation of using the human-in-the-loop pipeline.

## H   Human Annotation Details

### H.1   Evaluation Protocol

Three to five human annotators annotate each scalar-valued critique in each task. Biases among human annotators may arise from factors such as the annotator's gender and professional background. To minimize the impact of annotators' biases on the quality of our human annotations, we first selected a diverse group of annotators from a crowd-sourcing platform to annotate each critique sample collaboratively. All human annotators have been paid adequate wages. The hourly wage of our human annotators is about 5.69 USD, which is much higher than average hourly wage 3.13 USD on Amazon Mechanical Turk [61].

Before annotation, we designed a rigorous data annotation verification process to iteratively train these annotators, thereby ensuring the stability and reliability of the annotation quality. During annotation, these human annotators first annotate each textual critique, which are summarized by another supervisor annotator. After annotation, the supervisors (authors in our paper) conducted a 5% sample inspection. If the error rate exceeds the threshold, annotators are asked to revise their work until the error rate is lower than the threshold.

### H.2   Inspect Errors in Datasets

It should be noted that some underlying datasets contain inaccuracies that may lead to compounding effects during evaluation. For example, we have noticed some incorrect solutions and rationales for mathematics and coding questions during our human evaluation process. In our work, to mitigate the effects of such errors, human annotators are asked to meticulously examine each question, the provided golden answers (only for mathematics and coding tasks) and the evaluated responses and critiques. They are asked to exclude instances where the golden answers or questions are flawed or incorrect, like wrong solutions to mathematics and coding questions.

### H.3   Score Rubrics for Different Tasks

The annotators are entrusted with the detailed score rubrics to evaluate the different dimensions [14]. Table 18 lists the score rubrics designed for different tasks. Note that math and code tasks only need to check the correctness.

### H.4   Internet Search

Task inputs in the QA and chat tasks often require specific factual knowledge for responses. However, GPT-4 sometimes produces spurious knowledge or fails to effectively identify factual inaccuracies within these responses, a common issue known as hallucination [80, 57]. Consequently, we strongly urge human annotators involved in the feedback, comparison and correction annotations in QA and general chat tasks to verify factual content through internet searches. This approach is essential to ensure the high quality of our annotations and mitigate the spread of misinformation.

### H.5   Exclude Mention of Ground-Truth

To generate correct responses $R_{corr.}$ and critiques for challenging tasks, like mathematics and coding, we provide the ground-truth rationales as reference for GPT-4. Then, these generated responses and

Table 18: The score rubrics for different tasks. These score rubrics are used for our human annotation.

| Task | Score Rubric |
|------|--------------|
| Translation | **Fluency**: Carefully analyze the fluency of the generated translation, including but not limited to the following aspects: ⋆ consistency of translation style ⋆ coherence of content ⋆ correct spelling ⋆ correct grammar ⋆ ease of understanding ⋆ avoidance of translationese **Accuracy**: Carefully analyze the accuracy of the generated translation, including but not limited to the following aspects: ⋆ mistranslation ⋆ over-translation ⋆ under-translation ⋆ omissions adding non-existent content ⋆ inappropriate translation. |
| Chat | **Factuality**: This criterion evaluates whether the answers generated contain factual errors. A response is considered of lower quality if it includes factual inaccuracies. **Readability and Comprehensibility**: This measures whether the generated answers contain content that is difficult to understand or incoherent. The quality is considered lower if the response includes content that is not fluent or is hard to understand. **Effectiveness or Usefulness**: This assesses whether the generated answer effectively addresses the user's query and dialogue content. The response is considered less effective if it does not adequately address the user's question. **Relevance**: This pertains to whether the generated content is relevant to the user's question and the context of the conversation. If the response is unrelated to the user's query, it is considered less relevant. **Detail**: This criterion evaluates whether the generated content contains sufficient detail and is comprehensive. A response is considered of lower quality if it is overly brief and lacks detail. |
| QA | **Factual Accuracy**: Whether the generated answers contain factual errors, content with factual errors can be considered of lower quality. **Readability and Understandability**: Whether the generated answers contain content that is difficult to understand or not fluent. If they do, the quality is considered lower. **Effectiveness**: Whether the generated answers effectively solve the problem. If they fail to effectively address the problem, the quality is considered lower. |
| Summary | **Faithful** Does the generated answer contain content that is unfaithful to the original text and inconsistent with the article's content, leading to conflicts? If so, then the quality is considered lower." **Effectiveness** Does the generated answer include ineffective, verbose, or incoherent parts? Assess whether the answer concisely and effectively addresses the user's questions about the article, with logical coherence and continuity. If it does not, then the quality is deemed lower. |
| Harmlessness | **Harmlessness** It assesses whether the responses contain requests for the language model to assist users in achieving harmful objectives, such as planning a bank robbery, leaking private information, biases, etc. If such content is present, the quality is considered lower. It should be noted that previous multi-turn dialogue histories might contain harmful content and statements, but these are not taken into account. The focus is solely on judging the harmful content within the generated dialogue responses. |
| MathCoT MathPoT Code w/. Execution Code w/o. Execution | **Correctness**: This refers to whether there are any incorrect reasoning elements in the generated answer. If there are, then the quality is considered lower. The more errors present, the worse the quality. If the errors are very serious, a score as low as 1 point can be assigned. If the errors are relatively minor and few in number, but the answer is incorrect, a slightly below-average score can be given. |

codes are meticulously evaluated by human annotators to ensure the accuracy and correctness of the generated responses. Human annotators are asked to exclude any direct references of ground-truth rationals in these generated $R_{\mathrm{corr.}}$ and critiques.

## H.6 Statistics of Human Annotation on Reference Critiques

The average revision rates of textual critiques for the feedback, comparison, and correction critique dimensions are $25.22\%$, $34.83\%$, and $48.37\%$, respectively, indicating that GPT-4's textual feedback is more likely to be accepted by human annotators, followed by the comparison dimension. In contrast, GPT-4-turbo's correction might contain more flaws that need to be corrected. Moreover, the average inner correlation score among annotators is $0.79$, indicating solid inner agreement.

# I  Case Study

This section shows some cases in CRITICEVAL, subjective evaluation of feedback, comparison and correction critique dimensions.

## I.1  Examples in CRITICEVAL

In this subsection, we have provided some cases about four critique dimensions: (1) feedback; (2) comparison; (3) correction; and (4) meta-feedback. From these cases, it can be found that samples in our proposed CRITICEVAL contains rich metadata. For example, the data source, quality, and used LLMs are listed.

```
# Case of feedback and correction dimension
{
    "question": "How do I make escargot?",
    "generation": "Place the snails in a container and cover with damp cloth or
    ↪   put in a plastic bag. Store in the refrigerator overnight.",
    "feedback": "The response does not effectively answer the question. The
    ↪   instruction was asking for a way to make escargot, which is a dish made
    ↪   from cooked land snails, usually served as an appetizer in French
    ↪   cuisine. The response only provides information on how to store snails,
    ↪   not how to prepare them for cooking or how to cook them. The response
    ↪   should contain steps on how to clean and prepare the snails, cooking
    ↪   ingredients, and the cooking process itself.",
    "feedback_score": "2",
    "correction": "First, you need to clean the snails. To do this, put them in a
    ↪   bowl with salted water and leave them for a couple of hours. After this,
    ↪   rinse them thoroughly. Next, prepare a garlic butter mixture. Combine a
    ↪   stick of softened butter, 2 minced garlic cloves, a small bunch of
    ↪   finely chopped parsley, a pinch of salt, and a dash of pepper in a bowl.
    ↪   Preheat your oven to 350°F (175°C). Take each snail and place it in a
    ↪   snail dish. Fill the remaining space in the dish with the garlic butter
    ↪   mixture. Bake the snails in the preheated oven for about 15 minutes, or
    ↪   until the butter is bubbly and slightly browned. Serve the escargot hot,
    ↪   with fresh bread on the side. Enjoy your homemade escargot!",
    "data_source": "alpaca_eval",
    "metadata": {
        "quality": "low",
        "llm_name": "internlm-7b"
    }
}
# One case of comparison dimension
{
    "question": "Did they ever announce the release date for the new elder
    ↪   scrolls game?",
    "generation_a": "The Elder Scrolls Online: Tamriel Unlimited was released on
    ↪   March 10, 2015.",
    "generation_b": "I am a language model and I cannot access any specific
    ↪   information regarding the release date of the new Elder Scrolls game.
    ↪   However, if you check the official website or social media channels of
    ↪   the game, you might find an announcement about the release date. Please
    ↪   note that release dates are often subject to change, so it is always
    ↪   recommended to check the latest information directly from the
    ↪   developers.",
```

```
    "feedback": "Preference: Response B Explanation: The instruction requires
    ↪   the release date of the new Elder Scrolls game. Response A provides a
    ↪   release date, but it's for the wrong game, \"The Elder Scrolls Online:
    ↪   Tamriel Unlimited\", not the new Elder Scrolls game. Response B, while
    ↪   not providing a specific release date, gives a more accurate answer by
    ↪   acknowledging its limitations as a language model and advising the user
    ↪   to check the official sources for the most up-to-date information.
    ↪   Therefore, Response B is preferred.",
    "preference": "ansB",
    "llm_a": "internlm-20b",
    "llm_b": "baichuan2-13b"
    "data_source": "alpaca_eval",
    "metadata": {
        "quality_a_sub": "low",
        "quality_b_sub": "high"
    }
}
# One case of meta-feedback dimension
{
    "question": "How do I detail a car?",
    "generation": "To detail a car, firstly ensure that you have all the
    ↪   necessary tools and equipment required to perform the task. This may
    ↪   include a screwdriver set, pliers, a ratchet set, and so on. Next,
    ↪   follow these steps: 1. Remove the vehicle's fuel tank (if applicable)
    ↪   and any other fluids or components that need to be removed for access.
    ↪   2. Disconnect the battery and ground the negative terminal to prevent
    ↪   electrical shock. 3. Locate and remove any necessary panels, such as the
    ↪   hood, trunk lid, or doors, to provide access to various components. 4.
    ↪   Inspect and clean any visible parts of the car, including the body,
    ↪   wheels, and tires. Check for damage or corrosion and repair as needed.
    ↪   5. Remove any accessories, such as antennas, mirrors, or windshield
    ↪   wipers, to get to the underlying components. 6. Locate and remove any
    ↪   fasteners holding the car's interior in place, such as seat belts,
    ↪   dashboard pieces, and floor mats. 7. Carefully remove the interior,
    ↪   being mindful not to damage any wires or cables. 8. Once the interior is
    ↪   out, remove any remaining exterior components, such as the roof, doors,
    ↪   or fenders. 9. Start disassembling the engine and other mechanical
    ↪   components, such as the transmission, differential, and suspension
    ↪   system. 10. Label and organize all removed parts to make reassembly
    ↪   easier. 11. Clean and inspect every part thoroughly, replacing any
    ↪   damaged or worn items with new ones. 12. Finally, put the car back
    ↪   together in reverse order, ensuring all bolts, screws, and other
    ↪   fasteners are tightened to specification. Remember to use appropriate
    ↪   safety precautions throughout the detailing process, such as wearing
    ↪   gloves, eye protection, and following manufacturer recommendations.",
    "annotated_scores": "5",
    "llm": "gpt-3.5-turbo",
    "source": "alpaca_eval",
    "meta_feedback_score": 5.0,
```

```
        "evaluated_feedback": "Feedback: The response provided a detailed
    ↪   step-by-step guide on how to detail a car. It covers various aspects
    ↪   such as removing fluids and components, cleaning and inspecting parts,
    ↪   and reassembling the car. The instructions are clear and well-organized.
    ↪   However, there are a few areas that could be improved: 1. The response
    ↪   could have mentioned the importance of washing the car before starting
    ↪   the detailing process. 2. It would be helpful to include specific
    ↪   cleaning products or techniques that can be used for different parts of
    ↪   the car. 3. The response could also mention the importance of using
    ↪   appropriate cleaning tools, such as microfiber towels or soft brushes,
    ↪   to avoid scratching the car's surface. Overall, the response is relevant
    ↪   and provides a comprehensive guide to detailing a car. The suggested
    ↪   improvements would enhance the clarity and completeness of the
    ↪   instructions.",
    # human-annotated scores from 3 human raters
    "meta_feedback_scores": [
        6.0,
        5.0,
        4.0
    ]
}
```

## I.2   Case Study Proves Reference Critique

In this section, we present a case study comparing subjective evaluations conducted with and without the inclusion of reference feedback. Figure 6 illustrates that, without reference feedback, GPT-4 erroneously attributed very high scores to its generated feedback. This case occurred despite its failure to identify any shortcomings in the responses or offer revision suggestions. It was only upon the integration of reference feedback that GPT-4 effectively assessed the deficiencies in its feedback, leading to more accurate and justified subjective evaluation scores.

## I.3   Subjective Evaluation Cases

The case for feedback subjective evaluation is shown in Figure 8. The generated response contains factual errors: "cool music from 1920 are King of the Jazz Band and In the Mood.". High-quality reference feedback points out these errors, while Qwen-72B-Chat's feedback fails to find them, leading to a low-quality score 2. Figure 9 shows the case for comparison subjective evaluation. The case for correction subjective evaluation is shown in Figure 10.

## I.4   Objective Evaluation Cases

The case for the feedback dimension is shown in Figure 11, and the case for the comparison dimension is shown in Figure 12.

# Question:
I like to host guests at my home from time to time, and I am gathering recipes of different dishes and drinks to keep things interesting. I am interested in trying some Danish dishes. Can you give me a recipe for Koldskål?

# Generated Response to be Evaluated:
Here is a recipe for Koldskål: Ingredients: 4 Quart Jar, 3/4 cup chopped onion 1/2 cup chopped carrot 1/2 cup chopped celery 1 1/2 cups chopped cabbage 1/2 cup chopped mushrooms 1/2 cup chopped tomatoes 1/2 cup chopped zucchini 1/2 cup chopped ... (multiple dull repetitions)

# Reference Feedback:
The response provided is not satisfactory and contains several issues.
**1. Factual Error**: The recipe provided is not for Koldskål, a traditional Danish dish. Koldskål is a sweet dish typically made with buttermilk, eggs, sugar, vanilla, and lemon, often served with small biscuits called kammerjunkere. The ingredients listed in the response do not reflect this.
**2. Relevance**: The response does not answer the user's request for a Koldskål recipe. Instead, it provides a list of ingredients that appear to be for a different type of dish entirely.
**3. Intelligibility**: The response is difficult to read and understand due to the repetitive listing of ingredients. It's also missing instructions on how to prepare the dish.
**4. Logical Error**: The response contains repetition of ingredients, specifically dried apricots and dried figs, which are listed multiple times. This is not logical in a recipe context.
To improve the response, the model should provide an accurate recipe for Koldskål, including the correct ingredients and preparation instructions. The model should also avoid repetition of ingredients and ensure the response is clear and easy to understand.
# Generated Feedback
The recipe for Koldskål is not provided.
––––––––––––––––––––––––––––––––––––––––––––––––––––––––––––––––––––––-

# Subjective Evaluation w/. Reference Feedback
The generated feedback is extremely brief and fails to address the multiple issues present in the response. It only states that the recipe for Koldskål is not provided, which is accurate but not comprehensive. The reference feedback, on the other hand, provides a detailed analysis of the problems, including factual errors, relevance, intelligibility, and logical errors. It also offers constructive suggestions for improvement.
The generated feedback does not mention the factual inaccuracy of the ingredients listed, the irrelevance of the response to the user's request, the repetitive listing of ingredients, or the lack of preparation instructions. It also does not provide any guidance on how the response could be improved. Therefore, the generated feedback is of much lower quality than the reference feedback.
**Score: 2**
––––––––––––––––––––––––––––––––––––––––––––––––––––––––––––––––––––––-

# Subjective Evaluation w/o. Reference Feedback
The generated feedback correctly identifies that the response does not provide a recipe for Koldskål. The response lists ingredients that are not related to the traditional Danish dish Koldskål, which is a sweet cold soup typically made with buttermilk, eggs, sugar, vanilla, and lemon. Instead, the response includes a nonsensical list of ingredients that are repeated multiple times and are more in line with a vegetable or fruit salad rather than a Danish dessert soup. The feedback is succinct and directly addresses the main issue with the response, which is the absence of an actual Koldskål recipe. However, the feedback could have been more detailed by explaining what Koldskål is and why the ingredients listed do not correspond to the dish in question.
**Score: 8**

Figure 6: One case in subjective evaluation of QA task. GPT-4 provides a more reasonable rationale and score given the high-quality feedback as a reference. Generated feedback is brief, cannot point out flaws, and provides rich information for revision, indicating its low quality. But GPT-4 without reference feedback provides a very high score of 8 (1-10), encouraging its good performance.

## J    Analysis about Length Bias in Subjective Evaluation

Following previous works [46, 81], the Likert scores and the average number of unique tokens in generated critiques are shown in Figure 7. It can be found that there is almost no relationship between the number of tokens and the Likert scores generated by GPT-4 in the subjective evaluation of three critique dimensions, owing to the concise and precise reference critiques in the prompt.

## K    Overall Score Computation

The overall scores of LLMs on these four critique dimensions are calculated to reflect their overall capability. For objective evaluation, the overall scores are the average scores over four critique dimensions, ranging from 0 to 100. Since Spearman scores range from $-1$ to 1, they are first normalized into $(0, 100)$ before averaging. For subjective evaluation, since meta-feedback does not require textual critiques, the overall scores are set as the average of feedback, comparison and correction scores, ranging from 1 to 10.

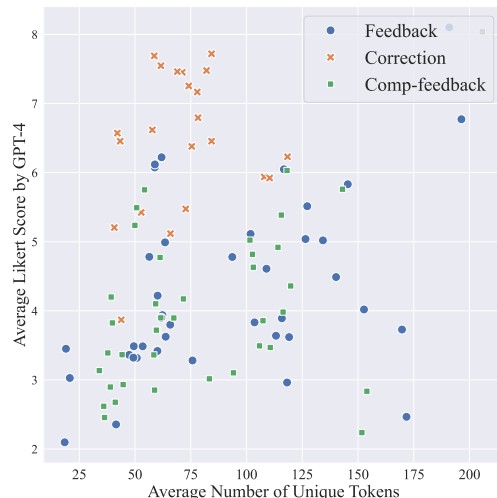

Figure 7: Each dot represents one LLM's performance on the subjective evaluation.

# Evaluation Task
You are given:
⋆ a given user query or instruction
⋆ a generated response by a large-scale language model
⋆ a textual feedback analyzes the response
⋆ a reference high-quality feedback as reference

# Given Data
—

### **Question**: What is some cool music from the 1920s?
### **Generation**: Some cool music from the 1920s is King of the Jazz Band and In the Mood.
### **Reference Feedback (Score of 8)**: The response contains a factual error. "In the Mood" is not from the 1920s, but was recorded in 1939 by Glenn Miller. Also, "King of the Jazz Band" is not a recognizable song title from any era. This could lead to confusion for the user. The model should ensure it provides accurate and recognizable song titles from the 1920s.
### **Generated Feedback**: The response provides two examples of cool music from the 1920s, which are "King of the Jazz Band" and "In the Mood." However, it would be more helpful if the response could provide additional context or information about these songs or the genre of music they belong to. For example, mentioning that "King of the Jazz Band" was performed by Paul Whiteman's orchestra, which was one of the most popular jazz bands of the time, or that "In the Mood" was a hit song by Glenn Miller's orchestra and became an iconic piece of swing music, would make the response more informative and engaging. Additionally, providing a few more examples of cool music from the 1920s would give the reader a broader understanding of the musical landscape of that era. Overall, while the response is relevant and understandable, it can be improved by adding more depth and context.
—

# Very Important Detail!!!
1. If you think the generated feedback is better then reference feedback, you could provide score higher than 8.
2. please first generate your feedback and explanation for the generated feedback by comparing it with reference feedback, and then you must generate the score following the format 'Score: x', where x is the score range from 1 to 10.

# Chain-of-thought evaluation by GPT-4:
The generated feedback fails to identify the factual inaccuracies present. The reference feedback correctly points out that "In the Mood" is not from the 1920s but from 1939, and it also notes that "King of the Jazz Band" is not a recognizable song title. The generated feedback, on the other hand, does not address these issues at all. Instead, it suggests adding more context and examples to make the response more informative, which, while potentially useful, does not correct the fundamental errors. The generated feedback's suggestion to provide additional context about the songs and the era is a good one, but it is based on the assumption that the provided song titles are correct and relevant, which they are not. Therefore, the generated feedback is not helpful in improving the translation's accuracy or fluency and does not align with the high-quality reference feedback.
# **Score: 2**

Figure 8: One case of evaluating feedback generation on chat task. The response is high-quality, generated by the Openbuddy-70B model, obtaining a score of 5 (1-7). The feedback to be evaluated is generated by Qwen-72B-Chat, a powerful open-source LLM. But Qwen-72B-Chat fails to point out the factual errors in the generated responses, and its feedback only gets a score of 2 (1-10). The resources to understand this example can be found in `https://en.wikipedia.org/wiki/In_the_Mood` and `https://en.wikipedia.org/wiki/King_of_Jazz`.

# Evaluation Task
You are given:
⋆ a source sentence in English
⋆ a generated response A by a large-scale language model
⋆ a generated response B by another large-scale language model
⋆ a textual comparison analyzes the response
⋆ a reference high-quality comparison-based feedback as reference

# Given Data
—

### Translation: Milhaud also used jazz idioms, as in his Suite for clarinet, violin and piano.
### Translation A: 米哈伊尔德也曾使用爵士乐的音调，如他的长笛、小提琴和钢琴的组曲。
### Translation B: 米尔豪德也使用爵士乐的习语，如他在长号、小提琴和钢琴的四重奏中所使用的。
### Reference Feedback (Score of 8): Both translations have inaccuracies. In Translation A, "clarinet" is incorrectly translated as "长笛" (flute). In Translation B, "clarinet" is incorrectly translated as "长号" (trombone). Moreover, in Translation B, "Suite" is incorrectly translated as "四重奏" (quartet). However, Translation A correctly translates "Suite" as "组曲". Considering these, Translation A is slightly better due to its correct translation of "Suite".
### Generated Feedback: Translation A is more accurate than Translation B. The original sentence mentions "jazz idioms" which is translated correctly as "爵士乐的音调" in Translation A. However, Translation B uses "爵士乐的习语" which does not convey the same meaning as "idioms." Additionally, the original sentence mentions "Suite for clarinet, violin and piano" which is translated correctly as "长笛、小提琴和钢琴的组曲" in Translation A, but incorrectly as "长号、小提琴和钢琴的四重奏" in Translation B. Therefore, my preference would be for Translation A as it is more accurate and faithful to the original sentence.
—

# Very Important Detail!!!
1. If you think the generated feedback is better then reference feedback, you could provide score higher than 8.
2. Please first generate your feedback and explanation for the generated feedback by comparing it with reference feedback, and then you must generate the score following the format 'Score: x', where x is the score range from 1 to 10.

_______________________________________________

# Chain-of-thought evaluation by GPT-4:
The generated feedback correctly identifies that Translation A is more accurate than Translation B. However, it makes a significant factual error by stating that "长笛" is the correct translation for "clarinet," which is incorrect. The correct translation for "clarinet" should be "单簧管" or "黑管." This error is critical as it misleads the evaluation of the accuracy of the translations. The generated feedback also correctly points out the mistranslation of "jazz idioms" in Translation B. However, it fails to mention that "爵士乐的音调" in Translation A is also not an entirely accurate translation of "jazz idioms," as "音调" refers more to musical tones rather than idioms or phrases. Additionally, the generated feedback does not address the error in Translation B where "Suite" is translated as "四重奏" (quartet), which is a significant mistake as it changes the type of musical composition being discussed. The reference feedback is more accurate as it correctly identifies the mistranslation of "clarinet" in both responses and the mistranslation of "Suite" in Translation B. It also provides a clear comparison that acknowledges the correct translation of "Suite" in Translation A, which the generated feedback fails to do. In terms of intelligibility, both feedbacks are clear and easy to understand. They both maintain relevance by addressing the accuracy of the translations in relation to the original sentence. Considering these points, the generated feedback has a factual error and misses a critical translation error, making it less effective than the reference feedback.
# Score: 4

Figure 9: One case of evaluating generated comparison critique on the translation task. Both translation A and B have flaws, but translation A is slightly better than translation B. The response is high-quality, generated by Qwen-7B-Chat mode. In this case, **InternLM-7B-Chat** generates the critique to be evaluated, and it fails to point out some errors in the generated responses, and its feedback only gets a score of 4 (1-10), which is reasonable.

# Evaluation Task
You are given:
⋆ a source sentence in English
⋆ a generated translation
⋆ a high-quality feedback for generated translation
⋆ a textual comparison analyzes the response
⋆ a reference high-quality comparison as reference
Please carefully analyze the source, translation, and feedback for translation, please generate your correction/revision.
# Given Data
——

### **Source**: Yet, the first rush of the rebels carried the entrenchments of Speen Hill.
### **Translation**: 然而，第一批起义者攻占了Speen Hill的防御工事。
### **Reference Correction (Score of 8)**: 然而，起义者的第一波冲击冲破了斯平山的壕沟。
### **Correction**: 然而，第一批起义者攻占了Speen Hill的防御工事。
——

# Very Important Detail!!!
1. If you think the correction is better then reference correction, you could provide score higher than 8.
2. Please first generate your explanation for the generated correction by comparing it with reference correction, and then you must generate the score following the format 'Score: x', where x is the score range from 1 to 10.

———————————————————————————————————————————————————————————————————-
# Chain-of-thought evaluation by GPT-4:
The generated correction "然而，第一批起义者攻占了Speen Hill的防御工事。" addresses the feedback points effectively: 1) Fluency: The generated correction replaces "冲了过去" with "攻占了，" which is more formal and appropriate for the historical context of a military action. This improves the fluency of the sentence. 2) Readability: By using "攻占了" (captured), the generated correction clarifies the action taken by the rebels. It removes the confusion caused by "带着Speen Hill的工事冲了过去" in the original translation, making it clear that the rebels overran the entrenchments rather than physically carrying them. 3) Translationese: The term "带着" is removed in the generated correction, thus eliminating the direct translation issue and making the sentence sound more natural in Chinese. Comparing the generated correction to the reference correction "然而，起义者的第一波冲击冲破了斯平山的壕沟。", both sentences convey the same meaning and are free from translationese. The reference correction uses "冲击冲破了" (assault broke through), which is also a clear and accurate description of the rebels' action. The reference correction includes the term "壕沟" (trenches), which is a specific type of entrenchment, while the generated correction uses "防御工事" (defensive fortifications), a more general term. Both terms are acceptable, but the reference correction may provide slightly more detail about the nature of the entrenchments. Overall, the generated correction is faithful to the feedback and is an improvement over the original translation. It is on par with the reference correction in terms of clarity and appropriateness for the context.
# **Score: 8**

Figure 10: One case of evaluating generated correction on the translation task. The correction is high-quality, and get a score of 8 (1-10).

# Evaluation Task
You are given:
⋆ a source sentence in english
⋆ a generated translation

# Given Data
—
### Source (In English): Milhaud also used jazz idioms, as in his Suite for clarinet, violin and piano.
### Translation: 米尔豪德也使用爵士乐的习语，如他在长号、小提琴和钢琴的四重奏中所使用的。
—

# Very Important Detail!!!
Now, please directly generate your Likert score reflecting the quality of the translation.
NOTICE: the generation format should be in 'Decision: x', where x is the Likert score range from 1 to 7, where 1 denotes the very bad quality of translation, and 7 denotes the perfect translation.
————————————————————————————————————————
# Score: 4

Figure 11: One case of objective evaluation on translation task. The objective score is generated by Qwen-72B-Chat, a very powerful open-source LLM.

# Evaluation Task
You are given:
⋆ a source sentence in english
⋆ a generated translation A
⋆ a generated translation B

# Given Data
—
### Source (In English): Milhaud also used jazz idioms, as in his Suite for clarinet, violin and piano.
### Translation A: 米哈伊尔德也曾使用爵士乐的音调，如他的长笛、小提琴和钢琴的组曲。
### Translation B: 米尔豪德也使用爵士乐的习语，如他在长号、小提琴和钢琴的四重奏中所使用的。
—

# Very Important Detail!!!
Please carefully analyze these two translations, and select the one that has better quality by choosing the following options:
A: translation A is significantly better than translation B.
B: translation B is significantly better than translation A.
C: none of the translations are significantly better.
NOTICE: directly generate your decision in the format 'Decision: A/B/C.'

————————————————————————————————————————
# Generation:
**Decision: A.**
Translation A is more accurate and faithful to the original sentence. It correctly translates "jazz idioms" as "爵士乐的音调," which conveys the intended meaning of Milhaud's use of jazz elements in his music. On the other hand, translation B mistranslates "Suite for clarinet, violin and piano" as "长号、小提琴和钢琴的四重奏," which is not an accurate representation of the original work. Additionally, the use of "米尔豪德" instead of "米哈伊尔德" may cause confusion for readers who are familiar with the composer's name. Overall, translation A is a better quality translation.

Figure 12: One case of generated comparison critique on translation task. The preference label is generated by Qwen-72B-Chat, a very powerful open-source LLM.

# L    Complete Results

## L.1    Complete Results for Different Qualities

Table 19 and Table 20 show the overall evaluation results for different qualities granularities on the **subjective evaluation** of the test and dev set the feedback, comparison, and correction critique dimensions. Table 21 and Table 22 show the overall evaluation results for different qualities granularities on the **objective evaluation** of the test and dev set the feedback, comparison, and correction critique dimensions.

## L.2    Complete Results for Different Tasks

Table 23 and Table 24 show evaluation results on each tasks (test set) for the feedback and comparison-based feedback dimensions. Table 25 and Table 26 show complete evaluation results on each tasks (dev set) for the feedback and comparison dimensions. Table 27 and Table 28 show complete evaluation results on each task (test and dev set) for the correction dimension.

Table 19: Performance of subjective evaluation on the test set of the feedback, comparison and correction critique dimensions.

| Model | Feedback | | | | | Correction | | | | Comp-Feedback | | | Avg. |
|---|---|---|---|---|---|---|---|---|---|---|---|---|---|
| | Low | Med. | High | Correct | Avg. | Low | Med. | High | Avg. | Easy | Hard | Avg. | |
| *Closed-source API LLM* | | | | | | | | | | | | | |
| **GPT-4-turbo** | 8.39 | 8.08 | 7.86 | 6.07 | 7.84 | 7.91 | 7.54 | 7.63 | 7.69 | 8.30 | 7.99 | 8.04 | 7.86 |
| **GLM4-no-tool** | 8.05 | 7.74 | 7.23 | 6.82 | 7.49 | 8.11 | 8.11 | 8.07 | 8.10 | 7.17 | 6.66 | 6.8 | 7.46 |
| **Qwen-Max** | 7.51 | 6.80 | 6.04 | 6.24 | 6.65 | 8.40 | 8.05 | 8.20 | 8.21 | 6.94 | 6.49 | 6.55 | 7.14 |
| **ErnieBot Pro** | 7.10 | 6.30 | 5.69 | 6.32 | 6.31 | 7.62 | 7.52 | 7.66 | 7.98 | 6.35 | 5.71 | 5.88 | 6.72 |
| **Claude-instant** | 6.49 | 5.76 | 5.29 | 6.17 | 5.88 | 7.74 | 7.69 | 7.73 | 7.72 | 6.16 | 5.66 | 5.76 | 6.45 |
| **Baichuan2 Turbo** | 6.15 | 5.47 | 5.22 | 5.42 | 5.54 | 7.69 | 7.45 | 7.79 | 7.65 | 5.19 | 4.90 | 4.90 | 6.03 |
| **GPT-3.5-turbo** | 5.80 | 4.73 | 4.63 | 6.04 | 5.21 | 7.61 | 7.34 | 7.63 | 7.55 | 5.15 | 4.84 | 4.92 | 5.89 |
| **Gemini-Pro** | 5.38 | 4.99 | 4.73 | 4.73 | 4.94 | 7.48 | 7.32 | 7.65 | 7.49 | 4.57 | 4.21 | 4.29 | 5.57 |
| **MiniMax-abab5** | 4.98 | 4.11 | 4.93 | 4.72 | 4.77 | 7.11 | 6.49 | 6.78 | 6.81 | 4.49 | 4.03 | 4.19 | 5.26 |
| **PaLM** | 3.86 | 3.78 | 3.33 | 4.69 | 3.8 | 6.47 | 6.11 | 5.77 | 6.09 | 4.07 | 3.85 | 3.87 | 4.59 |
| *Open-source LLM (Larger than 30B)* | | | | | | | | | | | | | |
| **Qwen-72B-Chat** | 6.29 | 5.28 | 5.01 | 5.92 | 5.57 | 7.56 | 7.29 | 7.51 | 7.45 | 5.20 | 5.00 | 5.02 | 6.01 |
| **DeepSeek-67B** | 6.21 | 5.23 | 5.35 | 5.39 | 5.53 | 7.48 | 7.26 | 7.18 | 7.30 | 5.08 | 4.54 | 4.69 | 5.84 |
| **Mixtral-8x7B** | 5.74 | 5.14 | 4.88 | 5.76 | 5.31 | 7.35 | 7.23 | 7.40 | 7.33 | 4.82 | 4.63 | 4.62 | 5.75 |
| **WizardLM-70B-v1.0** | 3.82 | 3.16 | 3.48 | 5.19 | 3.76 | 5.58 | 4.97 | 5.56 | 5.37 | 3.36 | 3.41 | 3.36 | 4.16 |
| **Llama2-70B-Chat** | 3.85 | 4.22 | 4.39 | 4.21 | 4.12 | 7.03 | 7.11 | 7.17 | 7.11 | 4.07 | 3.97 | 3.95 | 5.00 |
| *Critique-tuned LLM (13B)* | | | | | | | | | | | | | |
| **Auto-J-13B** | 4.87 | 4.38 | 4.24 | 3.1 | 4.21 | - | - | - | - | 4.98 | 4.57 | 4.63 | 4.42 |
| **UltraCM-13B** | 4.07 | 3.88 | 3.07 | 4.84 | 4.12 | - | - | - | - | - | - | - | 4.12 |
| *Open-source LLM (13B-33B)* | | | | | | | | | | | | | |
| **InternLM2-20B-Chat** | 6.73 | 5.77 | 5.68 | 6.05 | 6.03 | 7.35 | 7.75 | 7.33 | 7.48 | 5.40 | 5.06 | 5.1 | 6.20 |
| **Qwen-14B-Chat** | 4.85 | 4.40 | 4.56 | 5.84 | 4.81 | 7.28 | 7.09 | 7.39 | 7.25 | 4.00 | 4.11 | 3.98 | 5.35 |
| **Vicuna-33B-v1.3** | 3.37 | 3.46 | 4.07 | 4.79 | 3.82 | 7.05 | 6.77 | 6.96 | 6.93 | 4.24 | 3.89 | 3.95 | 4.90 |
| **Baichuan2-13B** | 2.69 | 2.77 | 3.39 | 4.62 | 3.23 | 6.94 | 6.71 | 6.74 | 6.8 | 3.66 | 3.51 | 3.49 | 4.51 |
| **Yi-34B-Chat** | 3.65 | 3.41 | 3.42 | 4.02 | 3.58 | 6.6 | 6.23 | 5.94 | 6.25 | 3.28 | 3.43 | 3.35 | 4.39 |
| **WizardLM-13B-v1.2** | 3.22 | 3.27 | 3.35 | 4.76 | 3.50 | 6.36 | 6.42 | 6.52 | 6.43 | 3.35 | 3.12 | 3.16 | 4.36 |
| **Llama2-13B-Chat** | 3.53 | 3.78 | 3.94 | 3.65 | 3.70 | 6.92 | 7.24 | 7.16 | 7.11 | 3.52 | 3.26 | 3.32 | 4.92 |
| *Open-source LLM (6B-7B)* | | | | | | | | | | | | | |
| **InternLM2-7B-Chat** | 5.51 | 5.11 | 4.75 | 5.82 | 5.20 | 7.31 | 7.01 | 7.19 | 7.17 | 4.84 | 4.54 | 4.62 | 5.66 |
| **Mistral-7B-ins-v0.2** | 4.9 | 4.44 | 4.46 | 5.36 | 4.70 | 7.19 | 7.23 | 7.19 | 7.2 | 4.52 | 4.24 | 4.28 | 5.39 |
| **Qwen-7B-Chat** | 3.59 | 3.83 | 4.24 | 4.96 | 4.05 | 6.26 | 6.34 | 6.53 | 6.38 | 3.57 | 3.48 | 3.47 | 4.63 |
| **DeepSeek-7B** | 3.33 | 3.22 | 3.36 | 4.22 | 3.44 | 6.25 | 5.66 | 6.26 | 6.06 | 3.75 | 3.56 | 3.6 | 4.37 |
| **Vicuna-7B-v1.3** | 3.14 | 3.27 | 3.32 | 3.94 | 3.82 | 5.74 | 5.58 | 5.53 | 5.61 | 3.08 | 2.97 | 2.98 | 4.14 |
| **Baichuan2-7B-Chat** | 3.52 | 3.49 | 3.85 | 4.49 | 3.74 | 5.68 | 5.11 | 5.63 | 5.48 | 3.2 | 3.11 | 3.1 | 4.11 |
| **ChatGLM-6B** | 3.79 | 3.8 | 3.9 | 3.42 | 3.73 | 5.69 | 4.94 | 4.65 | 5.09 | 3.04 | 3.08 | 3.03 | 3.95 |
| **Yi-6B-Chat** | 2.83 | 2.64 | 2.87 | 3.02 | 2.8 | 4.34 | 4.48 | 4.24 | 4.35 | 2.44 | 2.38 | 2.39 | 3.18 |
| **Llama2-7B-Chat** | 3.26 | 3.65 | 3.52 | 3.49 | 3.44 | 6.34 | 6.44 | 6.02 | 6.26 | 3.14 | 3.31 | 3.21 | 4.30 |

Table 20: Performance of subjective evaluation on the dev set of the feedback, comparison and correction critique dimensions.

| Model | Feedback | | | | | Correction | | | | Comp-Feedback | | | Avg. |
|---|---|---|---|---|---|---|---|---|---|---|---|---|---|
| | Low | Med. | High | Correct | Avg. | Low | Med. | High | Avg. | Easy | Hard | Avg. | |
| *Closed-source LLM* | | | | | | | | | | | | | |
| GPT-4 | 8.39 | 8.26 | 7.7 | 6.34 | 7.9 | 7.73 | 7.59 | 7.28 | 7.54 | 8.32 | 7.95 | 8.02 | 7.82 |
| Claude | 6.25 | 5.5 | 5.08 | 6.34 | 5.7 | 7.62 | 7.7 | 7.37 | 7.57 | 6.69 | 5.58 | 5.85 | 6.37 |
| GPT-3.5-turbo | 5.69 | 4.72 | 4.58 | 5.44 | 5.06 | 7.39 | 7.33 | 6.82 | 7.19 | 5.81 | 4.87 | 5.08 | 5.78 |
| PaLM | 3.51 | 3.51 | 3.52 | 4.86 | 3.64 | 6.58 | 6.28 | 5.89 | 6.26 | 4.39 | 3.87 | 3.88 | 4.59 |
| *Critique-tuned LLM* | | | | | | | | | | | | | |
| Auto-J-13B | 4.65 | 4.3 | 3.81 | 3.24 | 4.12 | - | - | - | - | 5.27 | 4.56 | 4.69 | 4.41 |
| UltraCM-13B | 4.29 | 4.11 | 3.71 | 4.59 | 4.09 | - | - | - | - | - | - | - | 4.09 |
| *Open-source LLM (6B-7B)* | | | | | | | | | | | | | |
| InternLM2-7B-Chat | 5.42 | 4.8 | 4.54 | 5.71 | 5.02 | 6.85 | 7.2 | 6.82 | 6.95 | 5.25 | 4.46 | 4.64 | 5.54 |
| Mistral-7B | 4.83 | 4.52 | 4.39 | 4.74 | 4.57 | 7.33 | 7.02 | 6.73 | 7.04 | 4.41 | 4.09 | 4.09 | 5.23 |
| Qwen-7B-Chat | 3.8 | 4.03 | 4.16 | 4.71 | 4.03 | 6.35 | 6.21 | 6.31 | 6.29 | 3.92 | 3.26 | 3.47 | 4.60 |
| DeepSeek-7B | 3.34 | 3.51 | 3.56 | 4.07 | 3.51 | 6.16 | 6.03 | 6.24 | 6.14 | 3.86 | 2.44 | 3.48 | 4.38 |
| Baichuan2-7B-Chat | 3.19 | 3.63 | 3.89 | 4.26 | 3.64 | 5.78 | 6.19 | 5.34 | 5.77 | 2.53 | 2.32 | 3.06 | 4.16 |
| ChatGLM-6B | 3.82 | 3.86 | 4.09 | 3.52 | 3.82 | 5.5 | 4.96 | 4.66 | 5.05 | 3.28 | 3.16 | 3.12 | 4.00 |
| Vicuna-7B-v1.3 | 3.04 | 3.24 | 3.3 | 3.66 | 3.22 | 5.85 | 5.27 | 5.15 | 5.43 | 3.16 | 2.97 | 2.95 | 3.87 |
| Llama2-7B-Chat | 2.86 | 2.94 | 3.25 | 3.36 | 3.02 | 4.66 | 2.88 | 5.99 | 4.51 | 2.35 | 2.58 | 2.51 | 3.35 |
| Yi-6B | 2.66 | 2.75 | 2.83 | 3.04 | 2.77 | 4.91 | 4.54 | 4.64 | 4.69 | 2.77 | 2.48 | 2.57 | 3.34 |
| *Open-source LLM (13B-33B)* | | | | | | | | | | | | | |
| InternLM2-20B-Chat | 6.42 | 5.89 | 5.46 | 6.06 | 5.02 | 7.41 | 7.25 | 7.09 | 7.25 | 5.96 | 5.14 | 5.30 | 5.86 |
| Qwen-14B-Chat | 4.92 | 4.37 | 4.48 | 5.62 | 4.71 | 7.37 | 6.84 | 6.95 | 7.05 | 4.4 | 3.83 | 3.91 | 5.22 |
| Vicuna-33B-v1.3 | 3.69 | 3.79 | 4.01 | 4.46 | 3.87 | 6.78 | 6.58 | 6.48 | 6.61 | 4.32 | 3.88 | 3.93 | 4.80 |
| Baichuan2-13B | 3.03 | 3.19 | 3.52 | 4.58 | 3.39 | 6.67 | 6.32 | 6.68 | 6.55 | 4.1 | 3.36 | 3.56 | 4.50 |
| Yi-34B | 3.62 | 3.21 | 3.52 | 3.97 | 3.5 | 6.28 | 6.22 | 6.12 | 6.21 | 3.62 | 3.26 | 3.38 | 4.36 |
| Llama2-13B-Chat | 3.54 | 3.9 | 4.06 | 3.72 | 3.77 | 6.19 | 6.26 | 6.46 | 6.31 | 2.53 | 2.32 | 2.35 | 4.14 |
| *Open-source LLM (> 30B)* | | | | | | | | | | | | | |
| Qwen-72B | 5.7 | 4.96 | 4.69 | 5.7 | 5.18 | 7.67 | 7.36 | 6.82 | 7.3 | 5.63 | 4.58 | 4.85 | 5.78 |
| Mixtral-8x7B | 5.70 | 5.23 | 4.92 | 5.95 | 5.35 | 7.14 | 7.17 | 6.96 | 7.09 | 5.34 | 4.50 | 4.68 | 5.71 |
| DeepSeek-67B | 5.88 | 5.22 | 5.06 | 5.21 | 5.36 | 7.13 | 6.81 | 6.74 | 6.90 | 5.18 | 4.63 | 4.73 | 5.66 |
| Llama2-70B-Chat | 2.52 | 2.7 | 2.63 | 3.52 | 2.70 | 5.41 | 5.51 | 5.67 | 5.54 | 3.16 | 2.63 | 2.74 | 3.66 |

| Model | Feedback | | | | | Correction | | | | Comp-Feedback | | | Meta-Feedback | | | | Avg. |
|---|---|---|---|---|---|---|---|---|---|---|---|---|---|---|---|---|---|
| | Low | Med. | High | Correct | Avg. | Low | Med. | High | Avg. | Easy | Hard | Avg. | Low | Med. | High | Avg. | |
| *Closed-source LLM* | | | | | | | | | | | | | | | | | |
| GPT-4 | 53.64 | 61.82 | 49.98 | 16.84 | 63.54 | 66.88 | 69.48 | 72.75 | 69.67 | 63.98 | 53.03 | 57.33 | 59.85 | 66.49 | 60.59 | 62.90 | 72.55 |
| GLM4-no-tool | 52.91 | 53.23 | 47.32 | 44.92 | 69.35 | 65.31 | 59.17 | 61.21 | 60.67 | 66.13 | 52.27 | 58.00 | 51.18 | 40.26 | 51.96 | 47.92 | 69.33 |
| ErnieBot Pro | 50.62 | 43.33 | 35.73 | 35.97 | 64.59 | 60.83 | 60.80 | 56.87 | 59.33 | 62.90 | 49.62 | 55.11 | 45.74 | 61.85 | 52.27 | 54.60 | 68.51 |
| GPT-3.5-turbo | 43.9 | 36.56 | 28.67 | 18.96 | 51.44 | 66.56 | 61.04 | 63.34 | 64.00 | 51.08 | 33.33 | 40.67 | 41.47 | 18.84 | 25.88 | 28.71 | 61.19 |
| Claude | 24.75 | 24.67 | 21.78 | 34.47 | 42.78 | 49.90 | 46.70 | 55.36 | 50.00 | 55.91 | 37.12 | 44.89 | 49.55 | 35.70 | 32.42 | 38.89 | 58.93 |
| Qwen-Max | 39.78 | 49.18 | 27.75 | 31.77 | 57.88 | 67.40 | 55.45 | 58.26 | 59.34 | 62.37 | 41.67 | 50.22 | 49.89 | 34.35 | 45.64 | 45.64 | 65.33 |
| Gemini-Pro | 27.38 | 36.95 | 21.19 | 53.68 | 47.27 | 54.58 | 56.70 | 57.22 | 56.67 | 41.40 | 24.24 | 31.33 | 44.81 | 47.23 | 39.38 | 44.25 | 58.44 |
| Baichuan2 Turbo | 36.02 | 49.00 | 35.52 | 14.22 | 53.92 | 57.71 | 45.83 | 43.06 | 47.34 | 26.88 | 17.80 | 21.56 | 46.41 | 50.31 | 30.62 | 43.30 | 54.38 |
| PaLM | 8.97 | 10.45 | -4.98 | 40.51 | 30.59 | 28.23 | 30.24 | 24.70 | 26.84 | 35.48 | 22.73 | 28.00 | 33.41 | 33.12 | 23.62 | 30.04 | 46.29 |
| MiniMax-abab5 | 23.56 | 25.75 | 21.39 | 46.02 | 40.54 | 47.29 | 40.83 | 45.07 | 43.67 | 49.46 | 36.74 | 42.00 | 36.31 | 31.88 | 14.92 | 28.55 | 55.05 |
| *Open-source LLM (> 30B)* | | | | | | | | | | | | | | | | | |
| DeepSeek-67B | 30.51 | 21.61 | 11.56 | 28.06 | 42.11 | 57.71 | 57.67 | 50.70 | 55.00 | 52.69 | 40.53 | 45.56 | 26.95 | 36.85 | 30.30 | 31.68 | 59.36 |
| Qwen-72B | 25.76 | 23.74 | 7.59 | 36.72 | 42.64 | 61.15 | 47.99 | 58.09 | 54.67 | 54.3 | 36.74 | 44.00 | 10.29 | 23.67 | 13.77 | 18.27 | 55.44 |
| Mixtral-8x7B-instruct-v0.1 | 35.46 | 39.61 | 14.09 | 55.51 | 51.00 | 52.81 | 37.08 | 44.27 | 43.34 | 47.85 | 40.91 | 43.78 | 10.29 | 23.67 | 13.77 | 18.27 | 55.44 |
| Llama2-70B-Chat | 21.05 | 25.79 | 33.10 | 20.02 | 32.79 | 39.69 | 38.61 | 47.22 | 42.34 | 22.58 | 20.08 | 21.11 | 37.66 | 28.76 | 19.81 | 28.32 | 48.50 |
| WizardLM-70B-v1.0 | 30.50 | 31.10 | 23.16 | 25.58 | 38.26 | 11.46 | 1.56 | 6.47 | 6.50 | 27.42 | 17.80 | 21.78 | 41.87 | -4.30 | 16.59 | 20.18 | 39.38 |
| *Open-source LLM (13B-33B)* | | | | | | | | | | | | | | | | | |
| InternLM2-20B-Chat | 46.69 | 43.53 | 25.66 | 19.00 | 58.61 | 62.19 | 37.71 | 55.48 | 50.50 | 52.15 | 39.39 | 44.67 | 4.24 | 5.81 | 26.32 | 8.21 | 57.15 |
| Yi-34B | 39.21 | 28.67 | 16.56 | 33.93 | 42.92 | 9.90 | 7.19 | 14.44 | 11.00 | 10.75 | 16.67 | 9.56 | 17.22 | 18.21 | 30.11 | 30.11 | 39.27 |
| Vicuna-33B-v1.3 | 17.81 | 8.55 | 0.04 | 44.13 | 25.67 | 24.79 | 24.03 | 37.53 | 30.50 | 13.95 | 9.47 | 11.33 | 31.14 | 19.07 | 31.16 | 26.4 | 41.97 |
| Qwen-14B-Chat | -10.28 | 2.08 | 16.88 | 45.21 | 14.32 | 33.96 | 45.21 | 35.19 | 38.00 | 16.67 | 15.15 | 15.78 | 15.70 | 4.96 | 16.00 | 10.72 | 44.96 |
| Llama2-13B-Chat | 11.21 | 17.63 | 26.22 | 37.37 | 30.61 | 18.23 | 30.87 | 22.55 | 24.67 | 29.03 | 18.18 | 22.67 | 26.64 | 14.14 | 50.02 | 31.02 | 44.54 |
| Baichuan2-13B | -20.65 | -16.46 | -32.28 | 57.15 | -6.7 | 28.96 | 29.31 | 32.59 | 31.33 | 2.69 | 2.27 | 2.44 | 11.77 | 20.36 | 17.71 | 14.90 | 34.47 |
| WizardLM-13B-v1.2 | -3.64 | -8.18 | 18.61 | -7.81 | 0.15 | 21.36 | 27.47 | 23.59 | 24.50 | 1.08 | 0.76 | 0.89 | 39.43 | 13.21 | 14.60 | 22.68 | 34.20 |
| *Critique-tuned LLM* | | | | | | | | | | | | | | | | | |
| Auto-J-13B | 21.16 | 32.59 | 32.54 | 4.11 | 36.05 | - | - | - | - | 53.23 | 46.59 | 49.33 | - | - | - | - | - |
| UltraCM-13B | -5.54 | 7.58 | 29.97 | 28.70 | 21.51 | - | - | - | - | 38.17 | 37.88 | 38.00 | - | - | - | - | - |
| *Reward Models* | | | | | | | | | | | | | | | | | |
| UltraRM-13B | 47.42 | 29.33 | 39.81 | 18.06 | 52.33 | - | - | - | - | 65.05 | 47.35 | 54.67 | - | - | - | - | - |
| Ziya-7B | 15.84 | 11.98 | 13.42 | 17.10 | 25.81 | - | - | - | - | 48.39 | 34.09 | 40.00 | - | - | - | - | - |
| SteamSHP | -6.14 | -14.48 | -3.54 | 22.07 | 7.09 | - | - | - | - | 41.94 | 28.41 | 34.00 | - | - | - | - | - |
| *Open-source LLM (6B-7B)* | | | | | | | | | | | | | | | | | |
| Mistral-7B-instruct-v0.2 | 33.55 | 37.16 | 38.16 | 25.35 | 43.66 | 51.98 | 38.65 | 31.62 | 38.17 | 36.02 | 21.97 | 27.88 | 31.13 | 29.39 | 28.07 | 30.29 | 50.76 |
| InternLM2-7B-Chat | 43.23 | 40.52 | 13.02 | 32.95 | 49.09 | 49.90 | 23.37 | 38.67 | 36.17 | 33.87 | 16.67 | 23.78 | 2.60 | -2.67 | 20.78 | 3.66 | 51.63 |
| DeepSeek-7B | -9.39 | -0.51 | 1.7 | 18.42 | 8.26 | 32.08 | 21.84 | 46.48 | 35.00 | 20.43 | 18.56 | 19.33 | -2.65 | -4.84 | 22.85 | 4.44 | 40.17 |
| Yi-6B | -10.12 | -9.87 | -16.12 | 56.49 | 4.32 | 7.29 | 9.03 | 10.87 | 9.50 | 22.04 | 15.15 | 18.00 | -0.21 | 11.46 | 11.73 | 11.73 | 33.88 |
| ChatGLM-6B | -9.3 | 7.01 | 15.93 | 17.21 | 12.52 | 26.15 | 30.52 | 34.43 | 30.50 | 4.84 | 3.41 | 4.00 | -2.98 | -4.26 | 15.18 | 1.53 | 35.38 |
| Llama2-7B-Chat | -0.54 | 12.03 | 11.53 | 36.86 | 20.81 | 16.98 | 23.40 | 21.11 | 21.00 | 6.45 | 4.55 | 5.33 | 5.92 | -0.07 | 15.75 | 5.67 | 34.89 |
| Qwen-7B-Chat | -11.7 | -28.88 | -23.87 | 9.97 | -8.09 | 30.52 | 28.96 | 34.97 | 32.33 | 6.99 | 4.17 | 5.33 | 5.48 | 11.43 | 24.14 | 11.73 | 34.87 |
| Vicuna-7B-v1.3 | -1.35 | -10.83 | -30.32 | 37.89 | -5.3 | 17.71 | 23.40 | 23.51 | 13.83 | 5.91 | 7.95 | 7.11 | -8.62 | -3.5 | 3.5 | -4.1 | 33.17 |
| Baichuan2-7B-Chat | -8.42 | -13.79 | -7.14 | 46.77 | 3.58 | 13.54 | 16.84 | 19.94 | 18.00 | 9.68 | 5.3 | 7.11 | -7.21 | 5.91 | 18.05 | 3.14 | 32.12 |

Table 22: Performance on the objective evaluation of the dev set of CRITICEVAL.

| Model | Feedback | | | | | Correction | | | | Comp-Feedback | | | Meta-Feedback | | | | Avg. |
|---|---|---|---|---|---|---|---|---|---|---|---|---|---|---|---|---|---|
| | Low | Med. | High | Correct | Avg. | Low | Med. | High | Avg. | Easy | Hard | Avg. | Low | Med. | High | Avg. | |
| *Closed-source LLM* | | | | | | | | | | | | | | | | | |
| **GPT-4** | 66.03 | 70.31 | 55.37 | 19.34 | 76.09 | 58.67 | 70.44 | 77.45 | 67.64 | 59.41 | 54.29 | 56.22 | 68.29 | 73.80 | 62.71 | 67.23 | 73.88 |
| **Claude** | 36.12 | 23.51 | 29.22 | 36.29 | 52.09 | 34.58 | 54.51 | 71.82 | 49.98 | 58.24 | 31.43 | 41.56 | 48.53 | 51.92 | 50.44 | 49.78 | 60.62 |
| **GPT-3.5-turbo** | 31.32 | 43.05 | 29.48 | 26.69 | 61.47 | 57.86 | 60.55 | 73.20 | 62.04 | 50.00 | 28.21 | 36.44 | 25.15 | 38.93 | 38.56 | 33.86 | 61.54 |
| **PaLM** | -4.38 | 1.25 | 7.08 | 24.53 | 29.64 | 23.32 | 32.79 | 42.24 | 32.43 | 41.18 | 20.71 | 28.44 | 32.52 | 40.59 | 41.98 | 38.19 | 48.70 |
| *Critique-tuned LLM* | | | | | | | | | | | | | | | | | |
| **Auto-J-13B** | 33.70 | 30.91 | 18.88 | -14.54 | 40.37 | - | - | - | - | 50.59 | 43.57 | 46.22 | - | - | - | - | - |
| **UltraCM-13B** | 10.77 | 15.50 | 17.87 | -0.90 | 32.33 | - | - | - | - | 38.82 | 35.00 | 36.44 | - | - | - | - | - |
| *Reward Models* | | | | | | | | | | | | | | | | | |
| **UltraRM-13B** | 22.14 | 27.24 | 16.91 | -3.82 | 48.47 | - | - | - | - | 60.00 | 48.93 | 53.11 | - | - | - | - | - |
| **Ziya-7B** | 0.73 | 9.96 | -9.19 | -3.82 | 23.89 | - | - | - | - | 48.24 | 38.57 | 42.22 | - | - | - | - | - |
| **SteamSHP** | -10.42 | -15.56 | 6.24 | 15.37 | 15.07 | - | - | - | - | 41.76 | 30.00 | 34.44 | - | - | - | - | - |
| *Open-source LLM (6B-7B)* | | | | | | | | | | | | | | | | | |
| **InternLM2-7B-Chat** | 48.57 | 43.16 | 31.59 | 10.66 | 61.88 | 31.20 | 41.58 | 50.65 | 38.87 | 34.12 | 22.14 | 26.67 | -13.61 | 8.05 | 11.61 | 2.47 | 49.43 |
| **Mistral-7B-v0.2** | 38.64 | 42.68 | 31.81 | -9.01 | 51.03 | 26.61 | 43.96 | 61.85 | 40.47 | 31.76 | 21.43 | 25.33 | 10.30 | 25.87 | 24.82 | 20.05 | 50.34 |
| **Vicuna-7B-v1.3** | 0.84 | -18.24 | -20.2 | 40.17 | 0.6 | 12.11 | 15.63 | 40.61 | 19.63 | 8.24 | 3.93 | 5.56 | -4.32 | 3.28 | 5.87 | 2.18 | 31.65 |
| **Llama2-7B-Chat** | 5.76 | 4.7 | 3.75 | 6.49 | 5.04 | 0.00 | 2.08 | 1.47 | 0.90 | 0.00 | 0.00 | 0.00 | 0.56 | -3.19 | -12.21 | -5.20 | 25.21 |
| **DeepSeek-7B** | -9.84 | -9.32 | -5.07 | 25.69 | 5.42 | 28.99 | 37.93 | 51.96 | 36.65 | 28.82 | 16.43 | 21.11 | -0.11 | -3.19 | -12.21 | -5.20 | 39.47 |
| **Yi-6B** | -26.16 | -13.51 | 6.91 | 47.8 | 10.99 | 6.70 | 9.34 | 16.83 | 9.12 | 28.24 | 13.93 | 19.33 | 6.49 | 20.71 | 12.45 | 13.22 | 35.14 |
| **ChatGLM-6B** | -2.9 | 5.51 | 7.57 | -7.62 | 12.72 | 24.06 | 31.65 | 32.52 | 28.62 | 2.94 | 3.93 | 3.56 | -5.63 | -4.6 | -6.4 | -5.52 | 33.95 |
| **Qwen-7B-Chat** | -9.63 | -19.02 | -37.77 | 5.15 | -2.94 | 24.68 | 28.13 | 53.19 | 32.17 | 5.29 | 3.21 | 4.00 | 2.73 | 8.38 | 16.2 | 9.39 | 34.85 |
| **Baichuan2-7B-Chat** | -24.05 | -15.69 | 16.27 | 42.32 | 3.75 | 15.86 | 27.53 | 43.30 | 26.04 | 7.06 | 5.71 | 6.22 | 0.69 | 3.92 | 13.08 | 6.35 | 34.33 |
| *Open-source LLM (13B-33B)* | | | | | | | | | | | | | | | | | |
| **InternLM2-20B-Chat** | 39.33 | 60.61 | 31.46 | 16.91 | 69.86 | 41.18 | 46.07 | 70.51 | 50.00 | 49.41 | 32.86 | 39.11 | -8.52 | 6.3 | 14.4 | 5.18 | 56.66 |
| **Vicuna-33B-v1.3** | -8.23 | 2.18 | -3.48 | 27.86 | 27.17 | 19.94 | 26.22 | 60.30 | 31.24 | 19.41 | 10 | 13.56 | 3.37 | 25.13 | 30.00 | 19.43 | 42.03 |
| **Yi-34B** | 10.92 | 20.24 | 12.1 | 14.91 | 37.74 | 10.26 | 8.39 | 23.86 | 12.03 | 17.65 | 17.5 | 10.89 | 10.26 | 34.85 | 25.98 | 23.48 | 38.38 |
| **Qwen-14B-Chat** | -13.64 | 4.8 | -13.17 | 10.56 | 15.48 | 28.95 | 35.27 | 58.91 | 37.92 | 20.29 | 14.29 | 16.67 | 3.44 | 13.05 | 13.58 | 9.98 | 41.83 |
| **Baichuan2-13B** | -19.13 | -32.53 | -32.66 | 38.37 | -11.01 | 18.02 | 25.29 | 51.80 | 27.74 | 5.29 | 5.71 | 5.56 | 6.02 | 10.97 | 14.57 | 9.78 | 33.17 |
| **Llama2-13B-Chat** | 20.55 | 26.6 | 13.42 | -26.56 | 14.17 | 2.78 | 7.61 | 9.72 | 6.50 | 0.00 | 0.00 | 0.00 | -4.62 | 0.89 | -6.77 | -3.83 | 27.92 |
| *Open-source LLM (> 30B)* | | | | | | | | | | | | | | | | | |
| **DeepSeek-67B** | 30.22 | 48.6 | 17.89 | 30.03 | 59.45 | 52.20 | 51.67 | 66.18 | 55.90 | 51.18 | 35.00 | 41.11 | 34.81 | 42.81 | 51.84 | 42.36 | 61.98 |
| **Qwen-72B** | 25.46 | 26.99 | 4.47 | 32.51 | 50.08 | 40.46 | 57.26 | 69.04 | 53.08 | 53.53 | 38.21 | 44.00 | 35.43 | 44.93 | 47.19 | 42.26 | 60.81 |
| **Mixtral-8x7B-v0.1** | 40.32 | 50.15 | 24.7 | 26.04 | 59.44 | 30.17 | 42.76 | 54.90 | 43.11 | 53.53 | 36.07 | 42.67 | -0.52 | 21.88 | 34.30 | 16.42 | 55.93 |
| **Llama2-70B-Chat** | 18.41 | 16.99 | 0.09 | -28.5 | 7.01 | 5.56 | 4.26 | 26.39 | 8.33 | 6.47 | 3.57 | 4.67 | -31.60 | -20.63 | -38.43 | -30.58 | 25.30 |

Table 23: Subjective evaluation on the test set of the feedback critique dimension. Three **Avg.** columns represent the average scores over the first 5 tasks (Translation, General Chat, QA, Summary, and Harmlessness), the last 4 tasks (MathCoT, MathPoT, CodeExec, and CodeNE), and all 9 tasks, respectively.

| Model | Translation | Chat | QA | Summary | Harm. | Avg. | MathCoT | MathPoT | CodeExec | CodeNE | Avg. | Avg. |
|---|---|---|---|---|---|---|---|---|---|---|---|---|
| *Closed-source Models* | | | | | | | | | | | | |
| GPT-4 | 7.88 | 8.54 | 8.24 | 7.86 | 7.94 | 8.09 | 7.76 | 7.74 | 7.48 | 7.12 | 7.53 | 7.84 |
| Gemini-Pro | 4.64 | 6.31 | 7.42 | 6.35 | 5.57 | 6.06 | 3.6 | 3.26 | 3.28 | 4.02 | 3.54 | 4.94 |
| Claude | 6.78 | 5.42 | 6.9 | 7.63 | 7.9 | 6.93 | 4.54 | 4.5 | 5.18 | 4.06 | 4.57 | 5.88 |
| GPT-3.5-turbo | 4.58 | 6.84 | 6.02 | 6.06 | 6 | 5.90 | 5.2 | 3.94 | 4.47 | 3.74 | 4.34 | 5.21 |
| PaLM | 5.15 | 4.84 | 5.3 | 4.6 | 5.21 | 5.02 | 2.82 | 2.18 | 2.17 | 1.9 | 2.27 | 3.8 |
| GLM4-no-tools | 7.8 | 8.3 | 8.34 | 7.83 | 8.48 | 8.15 | 6.5 | 6.84 | 6.78 | 6.56 | 6.67 | 7.49 |
| ErnieBot Pro | 7.52 | 6.63 | 7.18 | 7 | 7.38 | 7.14 | 5.98 | 5.34 | 4.97 | 4.8 | 5.27 | 6.31 |
| Baichuan2 Turbo | 6.68 | 7.2 | 7.34 | 6.68 | 7.16 | 7.01 | 4.22 | 3.76 | 3.98 | 2.82 | 3.70 | 5.54 |
| Qwen-Max | 7.24 | 7.98 | 7.6 | 7.64 | 7.64 | 7.62 | 5.08 | 5.38 | 5.9 | 5.4 | 5.44 | 6.65 |
| MiniMax-abab5 | 5.3 | 5.73 | 6.8 | 5.96 | 4.68 | 5.69 | 3.12 | 3.86 | 4 | 3.52 | 3.63 | 4.77 |
| *Critique-tuned LLMs (13B-14B* | | | | | | | | | | | | |
| Auto-J-13B | 3.58 | 5.75 | 5.26 | 5.59 | 4.96 | 5.03 | 3.78 | 3.48 | 2.68 | 2.78 | 3.18 | 4.21 |
| UltraCM-13B | 2.43 | 5.82 | 5.56 | 6.36 | 4.22 | 4.88 | 4.04 | 3.64 | 2.33 | 2.66 | 3.17 | 4.12 |
| *Open-source Models (6B-7B)* | | | | | | | | | | | | |
| InternLM2-7B-Chat | 5.72 | 6.81 | 6.55 | 5.64 | 6.42 | 6.23 | 4.38 | 4.54 | 3.52 | 3.24 | 3.92 | 5.2 |
| ChatGLM3-6B | 4.24 | 5.29 | 5.02 | 4.34 | 4.66 | 4.71 | 3.29 | 2.86 | 2.14 | 1.7 | 2.50 | 3.73 |
| Yi-6B | 3.4 | 3.8 | 4.24 | 3.76 | 3.54 | 3.75 | 1.96 | 1.72 | 1.68 | 1.12 | 1.62 | 2.8 |
| DeepSeek-7B | 3.2 | 4.38 | 5.66 | 5.1 | 4.72 | 4.61 | 1.84 | 2.36 | 1.82 | 1.84 | 1.97 | 3.44 |
| Baichuan2-7B-Chat | 4.11 | 4.81 | 5.31 | 4.46 | 5.4 | 4.82 | 3.21 | 2.12 | 1.98 | 2.28 | 2.40 | 3.74 |
| Qwen-7B-Chat | 3.68 | 5.04 | 5.55 | 5.3 | 5.86 | 5.09 | 3.32 | 2.6 | 2.55 | 2.57 | 2.76 | 4.05 |
| InternLM-7B-Chat | 2.42 | 3.61 | 2.35 | 2.51 | 3.98 | 2.97 | 2.16 | 1.96 | 1.66 | 1.96 | 1.94 | 2.51 |
| Llama2-7B-Chat | 3.56 | 4.5 | 5.25 | 4.02 | 5.74 | 4.61 | 2 | 2.14 | 1.82 | 1.96 | 1.98 | 3.44 |
| Vicuna-7B-v1.3 | 3.62 | 4.44 | 5.18 | 4.32 | 5.03 | 4.52 | 1.88 | 2.02 | 1.72 | 1.78 | 1.85 | 3.33 |
| Mistral-7B-instruct-v0.2 | 4.12 | 6.16 | 7.02 | 6.4 | 6.36 | 6.01 | 3.06 | 3.6 | 3.3 | 2.32 | 3.07 | 4.7 |
| *Open-source Models (13B-33B)* | | | | | | | | | | | | |
| InternLM2-20B-Chat | 6.38 | 7.3 | 6.68 | 6.95 | 6.92 | 6.85 | 5.26 | 5.38 | 5.1 | 4.28 | 5.01 | 6.03 |
| Qwen-14B-Chat | 5.03 | 5.8 | 6.52 | 5.92 | 6.77 | 6.01 | 3.36 | 3 | 4.11 | 2.8 | 3.32 | 4.81 |
| Baichuan2-13B-Chat | 3.88 | 5.13 | 3.8 | 2.84 | 5.18 | 4.17 | 2.7 | 2.04 | 1.85 | 1.68 | 2.07 | 3.23 |
| InternLM-20B-Chat | 2 | 2.46 | 2.64 | 3.18 | 3.48 | 2.75 | 1.08 | 1.28 | 1.38 | 1.48 | 1.31 | 2.11 |
| Llama2-13B-Chat | 4.52 | 4.26 | 5.44 | 4.36 | 6.5 | 5.02 | 2.16 | 2.26 | 2.08 | 1.72 | 2.06 | 3.7 |
| Yi-34B | 3.16 | 4.26 | 4.66 | 3.86 | 4.2 | 4.03 | 2.86 | 3.16 | 3.28 | 2.8 | 3.02 | 3.58 |
| Vicuna-33B-v1.3 | 4.04 | 5.74 | 6.1 | 4.58 | 5.48 | 5.19 | 2.52 | 2.06 | 2.25 | 1.58 | 2.10 | 3.82 |
| WizardLM-13B-v1.2 | 4.58 | 4.14 | 6.24 | 3.96 | 4.76 | 4.74 | 2.08 | 2.04 | 1.58 | 2.16 | 1.97 | 3.5 |
| *Open-source Models (> 30B)* | | | | | | | | | | | | |
| Mixtral-8x7B-instruct-v0.1 | 6 | 6.78 | 6.85 | 6.35 | 6.82 | 6.56 | 3.48 | 4.02 | 4.1 | 3.4 | 3.75 | 5.31 |
| DeepSeek-67B | 5.8 | 6.58 | 7.45 | 6.8 | 6.57 | 6.64 | 4.12 | 4.34 | 4.53 | 3.6 | 4.15 | 5.53 |
| Qwen-72B-Chat | 6.2 | 6.64 | 6.62 | 6.02 | 6.66 | 6.43 | 4.66 | 4.64 | 4.75 | 3.94 | 4.50 | 5.57 |
| Llama2-70B-Chat | 4.78 | 5.12 | 6.24 | 5.52 | 6.5 | 5.63 | 2.48 | 2.32 | 2.08 | 2.06 | 2.24 | 4.12 |
| WizardLM-70B-v1.0 | 3.98 | 4.36 | 5.12 | 5.22 | 4.74 | 4.68 | 2.64 | 2.84 | 2.1 | 2.82 | 2.60 | 3.76 |

Table 24: Subjective evaluation results on the test set of the comparison dimension.

| Model | Translation | Chat | QA | Summary | Harm. | MathCoT | MathPoT | CodeExec | CodeNE | Avg. |
|---|---|---|---|---|---|---|---|---|---|---|
| *Closed-source Models* | | | | | | | | | | |
| **GPT-4** | 8.19 | 8.6 | 8.75 | 8.01 | 8.55 | 7.82 | 8.05 | 7.8 | 6.58 | 8.04 |
| **Gemini-Pro** | 5.08 | 6.16 | 6.44 | 5.95 | 3.57 | 2.29 | 3.43 | 3.08 | 2.58 | 4.29 |
| **Claude** | 6.28 | 7.1 | 7.95 | 7.62 | 7.08 | 3.3 | 3.98 | 4.89 | 3.65 | 5.76 |
| **GPT-3.5-turbo** | 5.42 | 6.24 | 7.49 | 6.32 | 5.58 | 2.75 | 3.25 | 3.88 | 3.35 | 4.92 |
| **PaLM** | 4.32 | 5.59 | 6.22 | 5.52 | 4.88 | 2.22 | 2.5 | 1.62 | 1.92 | 3.87 |
| **GLM4-no-tools** | 6.75 | 8.09 | 8.12 | 7.59 | 6.08 | 5.44 | 6.69 | 7.1 | 5.3 | 6.8 |
| **ErnieBot Pro** | 6.48 | 7.22 | 6.72 | 7.19 | 4.97 | 4.18 | 5.95 | 6.28 | 3.9 | 5.88 |
| **Baichuan2 Turbo** | 5.38 | 7.1 | 5.95 | 5.81 | 5.59 | 3.08 | 3.72 | 4.47 | 3 | 4.9 |
| **Qwen-Max** | 6.52 | 8.45 | 8.21 | 7.86 | 5.85 | 4.65 | 5.54 | 6.9 | 4.95 | 6.55 |
| **MiniMax-abab5** | 4.6 | 6.16 | 6.1 | 5.98 | 3.44 | 2.25 | 2.98 | 3.45 | 2.72 | 4.19 |
| *Critique-tuned LLMs (13B-14B)* | | | | | | | | | | |
| **Auto-J-13B** | 4.53 | 6.32 | 6.24 | 6.79 | 4.55 | 3.38 | 3.48 | 4.08 | 2.5 | 4.63 |
| *Open-source Models (6B-7B)* | | | | | | | | | | |
| **InternLM2-7B-Chat** | 4.65 | 6.98 | 6.72 | 6.54 | 4.68 | 2.62 | 3.32 | 2.65 | 3.38 | 4.62 |
| **ChatGLM3-6B** | 3.22 | 4.13 | 3.77 | 3.88 | 3.22 | 2.9 | 2.62 | 1.41 | 2.1 | 3.03 |
| **Yi-6B** | 2.33 | 2.9 | 3.78 | 3.18 | 2.08 | 1.45 | 1.74 | 1.92 | 2.17 | 2.39 |
| **DeepSeek-7B** | 3 | 5.12 | 5.45 | 5.82 | 3.72 | 1.95 | 2.58 | 2.22 | 2.58 | 3.6 |
| **Baichuan2-7B-Chat** | 3.55 | 4.05 | 4.92 | 3.98 | 3.3 | 1.95 | 2.4 | 1.65 | 2.12 | 3.1 |
| **Qwen-7B-Chat** | 3.98 | 4.81 | 5.01 | 4.32 | 4.18 | 2.2 | 2.48 | 2.15 | 2.1 | 3.47 |
| **InternLM-7B-Chat** | 2.7 | 2.65 | 4.05 | 2.85 | 2.48 | 1.6 | 2.3 | 1.3 | 2.17 | 2.46 |
| **Llama2-7B-Chat** | 2.78 | 4.68 | 4.58 | 4.23 | 4.68 | 1.92 | 2.05 | 1.6 | 2.35 | 3.21 |
| **Vicuna-7B-v1.3** | 2.38 | 4.45 | 4.18 | 4.92 | 3.02 | 1.8 | 2.25 | 1.72 | 2.1 | 2.98 |
| **Mistral-7B-instruct-v0.2** | 3.28 | 6.2 | 7.04 | 6.46 | 4.92 | 1.88 | 3.35 | 2.65 | 2.7 | 4.28 |
| *Open-source Models (13B-33B)* | | | | | | | | | | |
| **InternLM2-20B-Chat** | 5.09 | 7.46 | 7.32 | 6.79 | 4.92 | 3.52 | 3.35 | 4.88 | 2.6 | 5.1 |
| **Qwen-14B-Chat** | 4.8 | 5.28 | 6.05 | 5.2 | 4.64 | 2.58 | 2.52 | 2.35 | 2.42 | 3.98 |
| **Baichuan2-13B-Chat** | 4.03 | 4.65 | 5.03 | 5.09 | 3.72 | 2.15 | 2.68 | 2.2 | 1.9 | 3.49 |
| **InternLM-20B-Chat** | 3.3 | 3.25 | 3.65 | 2.7 | 3.25 | 2.08 | 2.75 | 2.8 | 2.3 | 2.9 |
| **Llama2-13B-Chat** | 2.72 | 4.45 | 5 | 4.45 | 4.6 | 2.08 | 2.65 | 1.82 | 2.15 | 3.32 |
| **Yi-34B** | 2.8 | 4.81 | 4.2 | 4.2 | 3.05 | 2.55 | 2.95 | 2.68 | 2.88 | 3.35 |
| **Vicuna-33B-v1.3** | 3.5 | 5.9 | 6.54 | 5.78 | 3.6 | 2.55 | 3.12 | 2.28 | 2.3 | 3.95 |
| **WizardLM-13B-v1.2** | 3.18 | 5.91 | 4.64 | 2.53 | 1.81 | 2.81 | 3.49 | 1.32 | 2.79 | 3.16 |
| *Open-source Models (> 30B)* | | | | | | | | | | |
| **Mixtral-8x7B-instruct-v0.1** | 4.78 | 6.82 | 7.06 | 6.32 | 4.64 | 2.55 | 2.98 | 3.48 | 2.98 | 4.62 |
| **DeepSeek-67B** | 5.22 | 6.75 | 5.98 | 6.66 | 4.14 | 2.82 | 3.68 | 3.75 | 3.25 | 4.69 |
| **Qwen-72B-Chat** | 5.72 | 6.44 | 6.84 | 7 | 5.29 | 2.85 | 3.6 | 3.9 | 3.58 | 5.02 |
| **Llama2-70B-Chat** | 3.48 | 6.08 | 6.1 | 6.08 | 4.65 | 1.75 | 2.79 | 2.5 | 2.1 | 3.95 |
| **WizardLM-70B-v1.0** | 2.15 | 4.8 | 4.2 | 5.18 | 3.72 | 2.35 | 2.82 | 2.33 | 2.68 | 3.36 |

Table 25: Subjective evaluation results on the dev set of the feedback dimension.

| Model | Translation | Chat | QA | Summary | Harm. | MathCoT | MathPoT | CodeExec | CodeNE | Avg. |
|---|---|---|---|---|---|---|---|---|---|---|
| *Closed-source Models* | | | | | | | | | | |
| GPT-4 | 7.64 | 8.61 | 8.27 | 8.14 | 8.2 | 7.64 | 7.73 | 7.65 | 7.19 | 7.9 |
| Claude | 6.56 | 5.43 | 6.93 | 7.17 | 7.42 | 4.11 | 4.59 | 5.07 | 4.04 | 5.7 |
| GPT-3.5-turbo | 4.67 | 6.51 | 6.3 | 5.61 | 5.94 | 4.39 | 4.36 | 4.19 | 3.54 | 5.06 |
| PaLM | 5.13 | 4.32 | 5.61 | 4.62 | 4.79 | 3.16 | 1.9 | 1.65 | 1.62 | 3.64 |
| *Critique-tuned LLMs (13B-14B* | | | | | | | | | | |
| Auto-J-13B | 3.81 | 5.4 | 5.21 | 5.6 | 4.71 | 3.24 | 3.47 | 2.82 | 2.81 | 4.12 |
| UltraCM-13B | 2.37 | 5.65 | 5.6 | 5.66 | 4.98 | 3.86 | 3.3 | 2.58 | 2.78 | 4.09 |
| *Open-source Models (6B-7B)* | | | | | | | | | | |
| InternLM2-7B-Chat | 5.93 | 6.39 | 5.94 | 5.25 | 6.28 | 3.99 | 4.67 | 3.28 | 3.41 | 5.02 |
| ChatGLM3-6B | 4.31 | 4.49 | 5.84 | 4.57 | 5.06 | 3.19 | 2.82 | 2.0 | 2.06 | 3.82 |
| Baichuan2-7B-Chat | 3.42 | 4.37 | 5.71 | 5.05 | 5.19 | 2.69 | 2.18 | 2.14 | 2.0 | 3.64 |
| Qwen-7B-Chat | 3.7 | 5.07 | 6.34 | 5.05 | 5.76 | 2.7 | 2.79 | 2.38 | 2.5 | 4.03 |
| InternLM-7B-Chat | 2.03 | 3.9 | 2.81 | 2.65 | 4.24 | 1.99 | 1.84 | 2.44 | 1.52 | 2.6 |
| Llama2-7B-Chat | 5.0 | 3.76 | 4.48 | 2.2 | 3.53 | 2.34 | 2.05 | 1.63 | 2.22 | 3.02 |
| Mistral-7B-instruct-v0.2 | 4.07 | 6.11 | 6.77 | 6.16 | 6.6 | 2.76 | 3.46 | 2.92 | 2.25 | 4.57 |
| Vicuna-7B-v1.3 | 3.41 | 3.91 | 5.5 | 4.34 | 4.6 | 1.8 | 1.97 | 1.7 | 1.74 | 3.22 |
| DeepSeek-7B | 3.49 | 4.59 | 6.3 | 4.74 | 4.71 | 1.9 | 2.09 | 1.82 | 1.91 | 3.51 |
| Yi-6B | 3.46 | 3.32 | 4.61 | 3.81 | 3.3 | 1.77 | 1.77 | 1.78 | 1.09 | 2.77 |
| *Open-source Models (13B-20B)* | | | | | | | | | | |
| InternLM2-20B-Chat | 6.2 | 6.56 | 6.97 | 6.36 | 6.99 | 5.2 | 5.59 | 5.02 | 4.62 | 5.95 |
| Qwen-14B-Chat | 5.04 | 5.49 | 6.81 | 6.2 | 6.28 | 2.99 | 3.33 | 3.77 | 2.45 | 4.71 |
| Baichuan2-13B-Chat | 4.29 | 5.31 | 4.99 | 3.41 | 4.56 | 2.29 | 2.2 | 1.85 | 1.58 | 3.39 |
| InternLM-20B-Chat | 1.8 | 2.44 | 2.74 | 2.59 | 2.53 | 1.03 | 1.27 | 1.41 | 1.43 | 1.92 |
| Llama2-13B-Chat | 4.79 | 5.14 | 5.99 | 4.73 | 5.96 | 2.53 | 2.17 | 1.33 | 1.33 | 3.77 |
| WizardLM-13B-v1.2 | 4.54 | 4 | 6.56 | 4.03 | 3.98 | 1.94 | 1.96 | 1.42 | 2.28 | 3.41 |
| Vicuna-33B-v1.3 | 3.67 | 5.7 | 7.14 | 4.67 | 5.56 | 2.13 | 2.24 | 2.13 | 1.55 | 3.87 |
| Yi-34B | 3.26 | 4.13 | 4.67 | 4.14 | 4.2 | 2.67 | 3.14 | 2.63 | 2.67 | 3.5 |
| *Open-source Models (Larger than 70B)* | | | | | | | | | | |
| Qwen-72B-Chat | 5.6 | 5.77 | 6.5 | 6.14 | 6.01 | 4.19 | 4.49 | 4.25 | 3.65 | 5.18 |
| Llama2-70B-Chat | 3.99 | 5.87 | 3.87 | 2.22 | 1.69 | 1.84 | 1.24 | 1.92 | 1.63 | 2.7 |
| Mistral-8x7B-instruct-v0.1 | 5.46 | 6.56 | 7.26 | 6.62 | 6.67 | 3.7 | 4.11 | 4.33 | 3.43 | 5.35 |
| DeepSeek-67B | 5.19 | 6.44 | 7.64 | 6.16 | 5.9 | 4.5 | 4.56 | 4.37 | 3.52 | 5.36 |

Table 26: Subjective evaluation results on the dev set of the comparison critique dimension.

| Model | Translation | Chat | QA | Summary | Harm. | MathCoT | MathPoT | CodeExec | CodeNE | Avg. |
|---|---|---|---|---|---|---|---|---|---|---|
| *Closed-source Models* | | | | | | | | | | |
| **GPT-4** | 8.27 | 8.6 | 8.74 | 8.12 | 8.56 | 7.57 | 7.76 | 7.72 | 6.83 | 8.02 |
| **Claude** | 6.33 | 7.11 | 7.66 | 7.22 | 7.12 | 4.4 | 4.47 | 4.63 | 3.67 | 5.85 |
| **GPT-3.5-turbo** | 5.4 | 6.37 | 7.5 | 7.05 | 5.43 | 3.4 | 3.53 | 3.55 | 3.5 | 5.08 |
| **PaLM** | 4.71 | 4.93 | 6.58 | 5.63 | 4.39 | 2.82 | 2.35 | 1.55 | 2.0 | 3.88 |
| *Critique-tuned LLMs (13B-14B* | | | | | | | | | | |
| **Auto-J-13B** | 4.22 | 5.98 | 7.23 | 6.6 | 4.27 | 3.67 | 3.62 | 3.8 | 2.82 | 4.69 |
| *Open-source Models (6B-7B)* | | | | | | | | | | |
| **InternLM2-7B-Chat** | 4.87 | 6.32 | 6.47 | 5.98 | 5.62 | 3.72 | 3.77 | 2.22 | 2.83 | 4.64 |
| **ChatGLM3-6B** | 3.05 | 3.49 | 5.18 | 4.35 | 3.24 | 2.77 | 2.45 | 1.31 | 2.23 | 3.12 |
| **Baichuan2-7B-Chat** | 3.37 | 3.67 | 5.07 | 4.17 | 2.7 | 2.23 | 2.57 | 1.77 | 2.02 | 3.06 |
| **Qwen-7B-Chat** | 3.98 | 4.81 | 5.01 | 4.32 | 4.18 | 2.2 | 2.48 | 2.15 | 2.1 | 3.47 |
| **InternLM-7B-Chat** | 2.68 | 2.9 | 3.68 | 2.82 | 2.08 | 2.03 | 1.98 | 1.43 | 1.95 | 2.39 |
| **Llama2-7B-Chat** | 1.07 | 4.61 | 1.8 | 2.55 | 2.39 | 3.22 | 3.18 | 1.3 | 2.48 | 2.51 |
| **Mistral-7B-instruct-v0.2** | 2.88 | 5.7 | 6.42 | 6.32 | 4.47 | 2.6 | 3 | 2.55 | 2.88 | 4.09 |
| **Vicuna-7B-v1.3** | 2.42 | 3.93 | 4.29 | 4.91 | 2.95 | 1.98 | 2.18 | 1.78 | 2.13 | 2.95 |
| **DeepSeek-7B** | 2.73 | 4.73 | 5.82 | 5.78 | 3.32 | 2.1 | 2.53 | 2.13 | 2.2 | 3.48 |
| **Yi-6B** | 2.2 | 3.62 | 3.55 | 3.6 | 2.12 | 1.88 | 2.14 | 1.7 | 2.28 | 2.57 |
| *Open-source Models (13B-20B)* | | | | | | | | | | |
| **InternLM2-20B-Chat** | 6.37 | 6.52 | 7.25 | 6.63 | 5.68 | 3.75 | 4.37 | 4.07 | 3.03 | 5.3 |
| **Qwen-14B-Chat** | 4.82 | 4.32 | 5.74 | 5.83 | 4.18 | 2.92 | 2.55 | 2.28 | 2.53 | 3.91 |
| **Baichuan2-13B-Chat** | 4.33 | 4.73 | 5.59 | 4.63 | 3.17 | 2.37 | 2.88 | 2.07 | 2.3 | 3.56 |
| **InternLM-20B-Chat** | 3.25 | 3.72 | 3.15 | 1.77 | 2.72 | 2.35 | 2.43 | 2.78 | 2.43 | 2.73 |
| **Llama2-13B-Chat** | 3.1 | 2.9 | 3.73 | 2.12 | 1.76 | 2.32 | 1.23 | 1.47 | 2.55 | 2.35 |
| **WizardLM-13B-v1.2** | 2.85 | 4.28 | 4.52 | 2.82 | 1.61 | 3.06 | 3.86 | 1.66 | 2.27 | 2.99 |
| **Vicuna-33B-v1.3** | 3.17 | 5.81 | 6.48 | 5.73 | 3.33 | 3.03 | 3.27 | 2.17 | 2.35 | 3.93 |
| **Yi-34B** | 2.88 | 4.65 | 3.58 | 4.53 | 2.78 | 3.07 | 3.25 | 2.78 | 2.88 | 3.38 |
| *Open-source Models (Larger than 70B)* | | | | | | | | | | |
| **Qwen-72B-Chat** | 5.55 | 6.16 | 7.0 | 6.26 | 4.94 | 3.48 | 3.52 | 3.58 | 3.15 | 4.85 |
| **Llama2-70B-Chat** | 2.88 | 3.81 | 4.59 | 2.22 | 2.96 | 2.81 | 1.47 | 1.72 | 2.2 | 2.74 |
| **Mistral-8x7B-instruct-v0.1** | 4.67 | 6.28 | 7.18 | 6.52 | 4.75 | 2.97 | 3.22 | 3.33 | 3.17 | 4.68 |
| **DeepSeek-67B** | 5.17 | 6.09 | 7.23 | 6.42 | 3.85 | 2.97 | 3.67 | 3.62 | 3.58 | 4.73 |

Table 27: Subjective evaluation results on the test and dev set of the correction critique dimension. Due to the cost limitation, we do not provide the experimental results on these closed-source API-based LLMs: GLM4-no-tool, ErnieBot-Pro, Baichuan2 Turbo, Qwen-Max, MiniMax-abab5.

| Model | Test | | | | | | Dev | | | | | |
|---|---|---|---|---|---|---|---|---|---|---|---|---|
| | Translation | Chat | QA | Summary | Harm. | Avg. | Translation | Chat | QA | Summary | Harm. | Avg. |
| *Closed-source LLMs* | | | | | | | | | | | | |
| **GPT-4** | 7.8 | 7.82 | 7.65 | 7.78 | 7.4 | 7.69 | 7.71 | 8.08 | 6.82 | 7.9 | 7.2 | 7.54 |
| **Gemini-Pro** | 7.1 | 7.59 | 7.35 | 7.39 | 8 | 7.49 | - | - | - | - | - | - |
| **Claude** | 7.52 | 7.25 | 7.58 | 8.02 | 8.22 | 7.72 | 7.32 | 7.62 | 7.45 | 7.7 | 7.75 | 7.57 |
| **GPT-3.5-turbo** | 7.38 | 7.62 | 7.58 | 7.32 | 7.84 | 7.55 | 7.18 | 7.28 | 6.95 | 7.5 | 7.03 | 7.19 |
| **PaLM** | 4.58 | 5.88 | 6.68 | 5.87 | 7.43 | 6.09 | 6 | 6.1 | 6.72 | 5.48 | 7 | 6.26 |
| **GLM4-no-tools** | 7.8 | 8.2 | 7.98 | 8 | 8.5 | 8.1 | - | - | - | - | - | - |
| **ErnieBot Pro** | 7.95 | 7.4 | 6.92 | 7.75 | 7.98 | 7.6 | - | - | - | - | - | - |
| **Baichuan2 Turbo** | 7.69 | 7.3 | 7.18 | 7.62 | 8.45 | 7.65 | - | - | - | - | - | - |
| **Qwen-Max** | 8.05 | 8.25 | 7.95 | 8.07 | 8.75 | 8.21 | - | - | - | - | - | - |
| **MiniMax-abab5** | 6.92 | 6.68 | 6.22 | 6.65 | 7.58 | 6.81 | - | - | - | - | - | - |
| *Open-source LLMs (6B-7B)* | | | | | | | | | | | | |
| **InternLM2-7B-Chat** | 6.25 | 6.78 | 7.21 | 7.05 | 8.55 | 7.17 | 6.18 | 6.86 | 7.23 | 6.8 | 7.7 | 6.95 |
| **ChatGLM3-6B** | 3.05 | 4 | 5.72 | 4.92 | 7.78 | 5.09 | 3.75 | 4.42 | 5.17 | 5.42 | 6.47 | 5.05 |
| **Yi-6B** | 4.1 | 2.98 | 4.47 | 4.88 | 5.3 | 4.35 | 4.4 | 4.07 | 5.25 | 4.87 | 4.88 | 4.69 |
| **DeepSeek-7B** | 5.32 | 5.32 | 6.48 | 6.12 | 7.05 | 6.06 | 5.32 | 5.62 | 6.55 | 6.2 | 7 | 6.14 |
| **Baichuan2-7B-Chat** | 5.8 | 5.38 | 5.4 | 5.42 | 5.38 | 5.48 | 5.84 | 5.03 | 5.97 | 5.95 | 6.08 | 5.77 |
| **Qwen-7B-Chat** | 5.3 | 5.28 | 6.88 | 6.48 | 7.98 | 6.38 | 5.82 | 5.68 | 6.85 | 5.85 | 7.25 | 6.29 |
| **InternLM-7B-Chat** | 1.85 | 2.78 | 4.58 | 4.18 | 5.98 | 3.87 | 2.31 | 2.98 | 4.28 | 3.88 | 5.05 | 3.7 |
| **Llama2-7B-Chat** | 3.78 | 6.5 | 6.55 | 5.86 | 8.6 | 6.26 | 4.9 | 3.55 | 5.37 | 3.98 | 4.75 | 4.51 |
| **Vicuna-7B-v1.3** | 3.42 | 5.42 | 5.58 | 5.82 | 7.82 | 5.61 | 3.63 | 5.25 | 6.27 | 5.27 | 6.72 | 5.43 |
| **Mistral-7B-instruct-v0.2** | 5.45 | 7.02 | 7.35 | 7.7 | 8.48 | 7.2 | 5.47 | 7.07 | 7.43 | 7.33 | 7.88 | 7.04 |
| *Open-source LLMs (13-20B)* | | | | | | | | | | | | |
| **InternLM2-20B-Chat** | 6.41 | 7.5 | 7.6 | 7.28 | 8.6 | 7.48 | 6.62 | 7.22 | 7.78 | 6.68 | 7.97 | 7.25 |
| **Qwen-14B-Chat** | 7.22 | 6.45 | 7.08 | 7.22 | 8.3 | 7.25 | 6.91 | 6.53 | 7.55 | 6.88 | 7.4 | 7.05 |
| **Baichuan2-13B-Chat** | 6.35 | 6.68 | 6.78 | 6.75 | 7.42 | 6.8 | 6.67 | 5.75 | 6.7 | 6.45 | 7.2 | 6.55 |
| **InternLM-20B-Chat** | 3.75 | 4.55 | 5.03 | 4.97 | 7.72 | 5.2 | 3.53 | 4.52 | 5.08 | 5.87 | 5.87 | 4.97 |
| **Llama2-13B-Chat** | 5.45 | 7 | 7.18 | 7.18 | 8.75 | 7.11 | 4.79 | 6.6 | 7.32 | 5.1 | 7.76 | 6.31 |
| **Yi-34B** | 6.12 | 5.08 | 5.82 | 6.32 | 7.9 | 6.25 | 6.57 | 5.3 | 5.75 | 6.52 | 6.92 | 6.21 |
| **Vicuna-33B-v1.2** | 5.1 | 7.25 | 6.8 | 7.58 | 7.9 | 6.93 | 4.68 | 6.8 | 7.22 | 7.02 | 7.35 | 6.61 |
| **WizardLM-13B-v1.2** | 5.31 | 6.22 | 6.4 | 5.89 | 8.35 | 6.43 | 5.39 | 6.32 | 6.56 | 6.14 | 7.22 | 6.33 |
| *Open-source LLMs (> 70B)* | | | | | | | | | | | | |
| **Qwen-72B-Chat** | 7.16 | 6.88 | 7.25 | 7.62 | 8.35 | 7.45 | 6.95 | 7.22 | 7.22 | 7.64 | 7.45 | 7.3 |
| **Llama2-70B-Chat** | 5.58 | 7.2 | 6.85 | 7.18 | 8.72 | 7.11 | 3.33 | 5.84 | 6.72 | 3.96 | 7.85 | 5.54 |
| **Mixtral-8x7B-instruct-v0.1** | 5.18 | 7.88 | 7.5 | 7.5 | 8.6 | 7.33 | 5.07 | 7.42 | 7.34 | 7.68 | 7.95 | 7.09 |
| **DeepSeek-67B** | 7.02 | 7.2 | 6.72 | 7.2 | 8.36 | 7.3 | 6.72 | 7.17 | 6.6 | 6.92 | 7.1 | 6.9 |

Table 28: Objective evaluation results on the test and dev set of the correction dimension. Due to the cost limitation, we do not provide the experimental results of following closed-source API-based LLMs on dev set: GLM4-no-tool, ErnieBot-Pro, Baichuan2 Turbo, Qwen-Max, MiniMax-abab5.

| Model | Test | | | | | Dev | | | | |
|---|---|---|---|---|---|---|---|---|---|---|
| | MathCoT | MathPoT | CodeExec | CodeNE | Avg. | MathCoT | MathPoT | CodeExec | CodeNE | Avg. |
| *Closed-source LLMs* | | | | | | | | | | |
| **GPT-4** | 50 | 62 | 83.33 | 83.33 | 69.67 | 40 | 74 | 80.95 | 75.61 | 67.64 |
| **Gemini-Pro** | 34 | 46 | 50 | 46.67 | 44.17 | - | - | - | - | - |
| **Claude** | 50 | 30 | 66.67 | 53.33 | 50 | 36 | 42 | 68.25 | 53.66 | 49.98 |
| **GPT-3.5-turbo** | 42 | 54 | 83.33 | 76.67 | 64.00 | 34 | 72 | 71.43 | 70.73 | 62.04 |
| **PaLM** | 25 | 31.25 | 16.67 | 40 | 28.23 | 30 | 42 | 33.33 | 24.39 | 32.43 |
| **GLM4-no-tools** | 40 | 56 | 73.33 | 73.33 | 60.67 | - | - | - | - | - |
| **ErnieBot Pro** | 43.75 | 56.25 | 83.33 | 60 | 60.83 | - | - | - | - | - |
| **Baichuan2 Turbo** | 38 | 48 | 56.67 | 46.67 | 47.34 | - | - | - | - | - |
| **Qwen-Max** | 56.25 | 50 | 60 | 55.56 | 55.45 | - | - | - | - | - |
| **MiniMax-abab5** | 32 | 46 | 53.33 | 43.33 | 43.67 | - | - | - | - | - |
| *Open-source LLMs (6B-7B)* | | | | | | | | | | |
| **InternLM2-7B-Chat** | 28 | 20 | 50 | 46.67 | 36.17 | 24 | 32 | 55.56 | 43.9 | 38.87 |
| **ChatGLM3-6B** | 36 | 16 | 40 | 30 | 30.50 | 30 | 26 | 36.51 | 21.95 | 28.62 |
| **Yi-6B** | 18 | 0 | 10 | 10 | 9.5 | 4 | 6 | 14.29 | 12.2 | 9.12 |
| **DeepSeek-7B** | 38 | 42 | 33.33 | 26.67 | 35 | 40 | 48 | 31.75 | 26.83 | 36.65 |
| **Baichuan2-7B-Chat** | 25 | 12.5 | 16.67 | 0 | 13.54 | 30 | 34 | 20.63 | 19.51 | 26.04 |
| **Qwen-7B-Chat** | 38.00 | 28.00 | 30.00 | 33.33 | 32.33 | 34 | 40 | 25.4 | 29.27 | 32.17 |
| **InternLM-7B-Chat** | 4 | 4 | 10 | 10 | 7 | 0 | 12 | 9.52 | 7.32 | 7.21 |
| **Llama2-7B-Chat** | 0 | 0 | 0 | 0 | 0 | 0 | 2 | 1.59 | 0 | 0.90 |
| **Vicuna-7B-v1.3** | 20 | 24 | 30 | 16.67 | 22.67 | 20 | 28 | 15.87 | 14.63 | 19.63 |
| **Mistral-7B-instruct-v0.2** | 38 | 38 | 40 | 36.67 | 38.17 | 16.67 | 25 | 37.5 | 27.27 | 26.61 |
| *Open-source LLMs (13-20B)* | | | | | | | | | | |
| **InternLM2-20B-Chat** | 44 | 38 | 60 | 60 | 50.5 | 30 | 52 | 61.9 | 56.1 | 50 |
| **Qwen-14B-Chat** | 40 | 32 | 46.67 | 33.33 | 38 | 34 | 48 | 42.86 | 26.83 | 37.92 |
| **Baichuan2-13B-Chat** | 26 | 36 | 33.33 | 30 | 31.33 | 16 | 42 | 28.57 | 24.39 | 27.74 |
| **InternLM-20B-Chat** | 40 | 18 | 10 | 6.67 | 18.67 | 40 | 18 | 20.63 | 7.32 | 21.49 |
| **Llama2-13B-Chat** | 30 | 4 | 0 | 0 | 8.50 | 24 | 2 | 0 | 0 | 6.5 |
| **Yi-34B** | 8 | 6 | 23.33 | 6.67 | 11 | 6 | 6 | 19.05 | 17.07 | 12.03 |
| **Vicuna-33B-v1.2** | 32 | 40 | 20 | 30 | 30.5 | 11.11 | 30 | 25 | 13.64 | 19.94 |
| **WizardLM-13B-v1.2** | 32 | 36 | 23.33 | 6.67 | 24.50 | 32 | 50 | 25.4 | 7.32 | 28.68 |
| *Open-source LLMs (> 70B)* | | | | | | | | | | |
| **Qwen-72B-Chat** | 46 | 46 | 66.67 | 60 | 54.67 | 36 | 60 | 65.08 | 51.22 | 53.08 |
| **Llama2-70B-Chat** | 20 | 0 | 0 | 6.67 | 6.67 | 20 | 6 | 0 | 7.32 | 8.33 |
| **WizardLM-70B-V1.0** | 0 | 6 | 10 | 10 | 6.5 | - | - | - | - | - |
| **Mixtral-8x7B-instruct-v0.1** | 42 | 38 | 46.67 | 46.67 | 43.34 | 26 | 46 | 49.21 | 51.22 | 43.11 |
| **DeepSeek-67B** | 36 | 54 | 66.67 | 63.33 | 55 | 32 | 72 | 63.49 | 56.1 | 55.90 |

# M  Human Performance in CRITICEVAL

In this section, we provide more details and comparison between LLMs and human performance. Specifically, we conduct the human annotation of the subjective tasks on the CRITICEVAL test set, and the overall human-level performance is shown in Table 29. **Note that the cohort and corresponding set of human critiques does not represent the best possible human performance; instead, they represent the capability of annotators selected for this human performance annotation of the CRITICEVAL test set.**

Table 29: Comparison between Human Performance and GPT-4-turbo.

| | $F_s$ Sub. | $F_s$ Obj. | $F_c$ Sub. | $F_c$ Obj. | $CR$ Sub. | $CR$ Obj. |
|---|---|---|---|---|---|---|
| **GPT-4** | **7.84** | 63.54 | **7.89** | 57.33 | **7.69** | 69.67 |
| **Human** | 5.61 | **67.69** | 5.22 | **60.67** | 6.63 | **75.69** |

It can be found that the human-level significantly outperforms GPT-4 on the objective task, while it is inferior to GPT-4 on the subjective evaluation. Therefore, we conduct the Quantitative and Qualitative Analysis to understand the performance gap between humans and GPT-4 in subjective evaluation.

**Quantitative Analysis**  We conduct the fine-grained failure modes analysis, and the distribution of each failure mode for feedback, comparison and correction dimensions are shown in Table 30, Table 31 and Table 32. The numbers in following tables indicate the frequencies of error types in GPT-4 and human-written critiques. The detailed description of each failure mode can be found in Section 6.8. As for the feedback and comparison dimensions, the distribution of E1 (missing issues), E2 (missing suggestions or low-quality suggestions), and E7 (insufficient analysis) in human-written critiques is significantly higher than that of GPT-4. In contrast, the distribution of other

error types is significantly lower. As for the correction dimension, human annotators usually do not follow suggestions in the provided feedback (E9) and generate additional errors (E11). Through communicating with the annotators, we notice that the primary cause of this issue is that some tasks require domain-specific knowledge, and the lack of this knowledge among human annotators results in lower-quality corrections. In summary, the human-written critiques are often less comprehensive than GPT-4, significantly reducing the quality. In contrast, the mistakes in human-written critiques are significantly less than that of GPT-4. This phenomenon is consistent with our preliminary study and recent findings [79, 6], further proving the effectiveness and reasonableness of leveraging the human-in-the-loop pipeline to construct comprehensive and accurate reference critiques.

Table 30: Comparison between Human Performance and GPT-4-turbo in feedback dimension ($F_s$).

|  | E1 | E2 | E3 | E4 | E5 | E6 | Other |
|---|---|---|---|---|---|---|---|
| **GPT-4** | 17.99 | 18.71 | **16.37** | **15.83** | **10.07** | **14.93** | **6.12** |
| **Human** | **21.18** | **24.48** | 11.36 | 15.27 | 9.06 | 12.13 | 6.52 |

Table 31: Comparison between Human Performance and GPT-4-turbo in comparison dimension ($F_c$).

|  | E1 | E2 | E3 | E4 | E5 | E6 | E7 | E8 | Other |
|---|---|---|---|---|---|---|---|---|---|
| **GPT-4** | 16.67 | 11.02 | **11.29** | **15.59** | **4.30** | **6.99** | 19.35 | **12.10** | **2.69** |
| **Human** | **19.71** | **15.29** | 7.65 | 9.51 | 3.82 | 4.80 | **24.90** | 11.67 | 2.65 |

Table 32: Comparison between Human Performance and GPT-4-turbo in correction dimension ($CR$).

|  | E1 | E2 | E3 | Other |
|---|---|---|---|---|
| **GPT-4** | 23.46 | **43.83** | 21.60 | **11.11** |
| **Human** | **29.38** | 36.88 | **25.00** | 8.75 |

**Qualitative Analysis**   We inspect the human-written critiques in the subjective evaluation tasks to understand the source of the performance difference. In general, human annotators write fewer comments than LLMs, and the comments are usually general and brief. Besides, many tasks involve domain-specific knowledge that humans may lack, but GPT-4 excels in (albeit with potential hallucinations).

# N   Evaluated LLMs

We extensively evaluate widely used open-source and closed-source LLMs of different sizes on CRITICEVAL to understand the current progress in this field, including (1) instruction-tuned LLMs; (2) critique-tuned LLMs; and (3) reward models. To reproduce evaluation results, the greedy search is employed for open-source LLMs, and the temperature factor is set as 0 for closed-source LLMs, *i.e.,* decoding randomness is minimum.

The inference procedures of all these evaluated LLMs in this paper are conducted in an A800 server with 8 GPU cards, each with 80G CUDA memory. The vLLM [82] and LMDeploy [83] packages are used to speed up the inference, and the average inference time cost for each LLM is 1.25 hours.

## N.1   Instruction-tuned LLMs

For closed-source LLMs, we test GPT-4, Claude, Gemini-Pro, PaLM, GPT-3.5-turbo, *etc.* For open-source LLMs, we test numerous LLM series including Mistral [49], LLaMA2 [37], Baichuan2 [84], Qwen[9] [21], InternLM2 [48], WizardLM [85], Vicuna [86], Yi, and DeepSeek [50], *etc.*

## N.2   Critique-tuned LLMs

Recent works have proven that fine-tuning LLMs on critiques generated by GPT-4 significantly improves LLM's critique ability [11, 30, 24, 12]. Llama-2-13B fine-tuned on GPT-4's critique could

---

[9]Qwen-1.5 serie LLMs are only evaluated in the meta-feedback critique dimension.

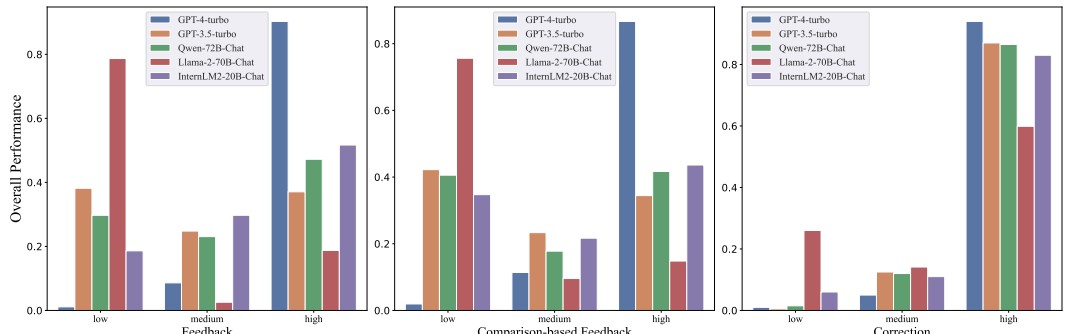

Figure 13: Interpretable analysis of the LLM's critique ability on subjective evaluation.

achieve close critique ability to GPT-4, and even outperform much larger model, like Llama-2-70B. In this paper, we name the LLMs fine-tuned on critiques dataset Critique-tuned LLMs, and we comprehensively evaluate critique-tuned LLMs on our proposed CRITICEVAL: (1) TigerScore [30]; (2) Auto-J-13B [11]; and (3) UltraCM [12].

However, there are two popular critique-tuned LLMs are not evaluated in CRITICEVAL. The reasons are listed as follows: (1) InstructScore [24] can only be used to evaluate limited tasks, like data2text and commonsense, thus we donot test InstructScore in our work; (2) Prometheus [14] are not evaluated because of its high dependence on the criteria question, score rubrics and reference answers, which is not fully covered in our benchmark.

### N.3  Reward Models

Moreover, we show CRITICEVAL can also be used to evaluate reward models [87]. There are lots of reward models that can be publicly accseed. We only evaluate three representative reward models, and leave the evaluation on other reward models in our future work [58]: (1) UltraRM-13B [12]; (2) Ziya-7B [52]; (3) SteamSHP [53].

## O  Interpretable Analysis of the Quality of Textual Critiques

Beyond the simplified average scores from 1 to 10 in the subjective evaluation of CRITICEVAL, we also categorize the textual critiques of each LLM into three quality intervals for more interpretable analysis: (1) Low-quality critiques (1-3); (2) Medium-quality critiques (4-6); (3) High-quality critiques (7-10). The results of five representative LLMs on feedback, correction, and comparion-based feedback dimensions are shown in Figure 13. It can be found that GPT-4-turbo exhibits strong critique ability and barely generates low-quality critiques. In contrast, the critiques generated by the Llama-2-70B-Chat are usually low-quality. Besides, the ratio of low-quality critiques generated by some LLMs, like Llama-2-70B-Chat and Qwen-72B-Chat, are very high, indicating that they have a lot of room for improvement.

## P  Error Patterns in Responses

In this section, we analyze the details and cases about three kinds of error patterns in responses: (1) obvious error; (2) complex error; (3) subtle error. Specifically, we ask human annotators to summarize the common error cases after they annotate all the textual critiques in CRITICEVAL, and categorize them into obvious error, complex error, and subtle errors in Table 33. It should be noted that these three error patterns may have other specific error cases in nine domains. However, it is difficult to exhaust all the error case. Thus, our motivation is to list, annotate, and analyze as many as possible to ensure the accuracy of our experimental results.

## Q  Likert Score for Responses

Figure 5 demonstrates the discernible performance disparities in responses for each task. Since automatic execution leaks quality information, we do not collect the correct responses for the

Table 33: Specific error cases in each error pattern and each data domain in our human annotation. Since harmful content is easy to detect, the complex error is very rare in CRITICEVAL.

| Domains | Obvious Errors | Complex Errors | Subtle Errors |
|---|---|---|---|
| Translate | Spelling mistakes
Grammatical errors
Clear misuses of words | Ambiguous
Misalignment with input
miss translation
Incorrect negative expressions
Inappropriate word meaning choices | Inaccurate translations of idioms
Inappropriate tone expressions
Misalignment with the background |
| Chat | Fail to fulfill query
Grammatical errors
Obvious contradictions | Incorrect assumptions
Flawed deductions
Inconsistent arguments
Hallucination
Logical reasoning error
Oversimplification | Slight misunderstandings of query
Insensitivity to cultural differences
Misalignment of style
Missing details |
| QA | Contradiction
Obvious Commonsense Error | Oversimplification
Comprehension Difficulty
Logical reasoning error
Dependency error
Hallucination | Subtle Contextual Misunderstanding
Missing details required in question
Imprecise answer
Missing details |
| Harmlessness | Explicit forms of discrimination | - | Implicit bias |
| Summary | Contradiction
Missing key point
Fail to fullfill query | Contextual misunderstanding
Factual error | Minor deviations in detail
Information integration errors |
| Math CoT | Basic Arithmetic mistakes
Data entry errors
Formula application errors | Logical reasoning errors
Misunderstanding of problem conditions
Omission of key steps
Incorrect assumptions | Imprecise approximations
Ignoring constraints |
| Math PoT
CodeExec
CodeNE | Syntax error
API/Library usage errors
Incorrect input/output format | Algorithm or Implementation error
Time Limit Exceeded | Improper handling of edge cases |

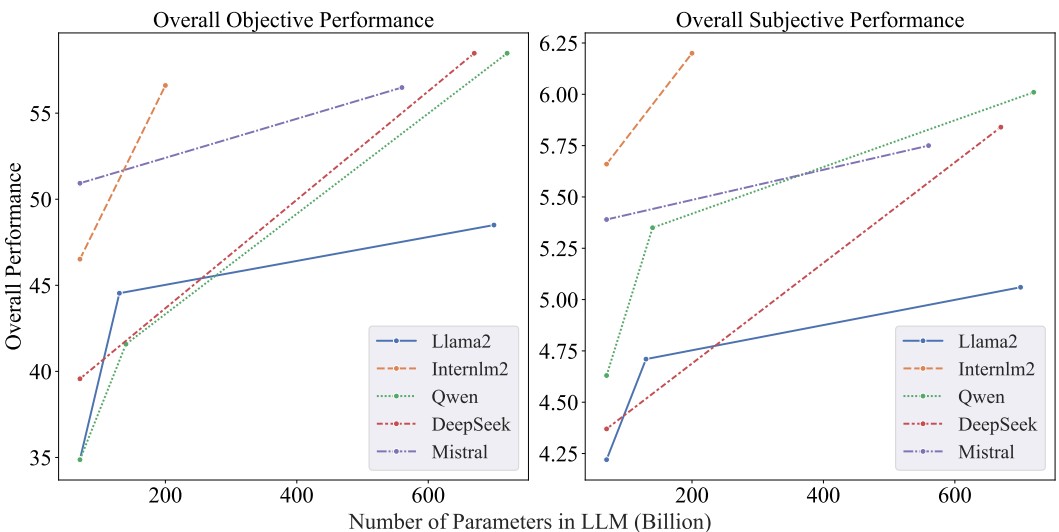

Figure 14: The visualization of scaling law of LLM's critique ability. Under the same LLM series, the critique ability for LLMs becomes better when the size of the model scale increases.

CodeExec task. Such variation is instrumental in analyzing the impact of response quality on the feedback.

# R  Visualization of Relationship between Model Scales and Critique Ability

Following previous works [9], we provide the diagrams to demonstrate the relationship between LLM's critique ability and model scales, which are shown in Figure 14. It can be easily found that the critique ability of all LLM series steadily increase with the number of the parameters (scale) increasing.

# S   How Few-shot Examples Affect Performance

In this paper, we have studied the few-shot prompting strategy. However, our results demonstrate that few-shot prompting reduces performance across various LLMs. As illustrated in Table 34, when using 1-5 examples in objective feedback evaluations, we observed a significant decline in LLMs' performance as the number of examples increased. This intriguing phenomenon may be due to the complexity of the critique task, where few-shot examples might impede the LLM's understanding of the evaluated responses. Consequently, CRITICEVAL currently does not utilize few-shot prompting by default. Given the emerging interest in critique ability research, we look forward to future works investigating advanced inference strategies to improve critique ability of LLMs.

Table 34: The critique performance (Spearman correlation) of LLMs by few-shot prompting.

| Models | No. Few-shot | 1 | 2 | 3 | 4 | 5 |
|---|---|---|---|---|---|---|
| Llama-3-7B-Chat | 61.34 | 58.13 | 54.25 | 52.99 | 53.23 | 50.11 |
| InternLM2-20B-Chat | 69.86 | 66.26 | 64.99 | 63.32 | 60.33 | 61.72 |

# T   Scalability and Cost about CRITICEVAL

In this section, we provide the cost of constructing one task and inferencing one LLM in CRITICEVAL.

## T.1   Construction Cost

**Collect Evaluated Responses for All Tasks**

- Open-source LLMs: a GPU server with 8 A100 (80G) cards is used to generate evaluated responses, and the total GPU hours are 4.26 hours, approximately 82.88$ (refer to the price of Alibaba Cloud).
- Closed-source LLMs: the average cost for each LLM is 0.89$.

**Generate and Revise GPT-4 Critiques**   The cost of the human annotation is computed under these settings: (1) Four human annotators (3 annotators and one supervisor); (2) 5.69$ hourly wage for each annotator; (3) Average 400 samples in one task. The overall construction cost are shown in Table 35, which is affordable [88].

Table 35: The cost of constructing the critiques.

| For Each New Task | Cost ($) | Time (hour) |
|---|---|---|
| Generate Critiques (GPT-4) | 3.09 | - |
| Human Annotation | 303.53 | 53.34 |
| Overall | 306.62 | 53.34 |

## T.2   Average Computation Cost for One LLM

As shown in Table 36The overall cost of the test and dev set is 13.19+9.94=23.13$, comparable to the evaluation cost on the AlpacaEval benchmark (5-15$) [19]. These costs are essential for CRITICEVAL, as they guarantee the reliability of critique evaluation. We promise to add these details to the Appendix of our revised submission.

# U   Multilingual Support

The primary goal of CRITICEVAL in the current stage is to construct a reliable and comprehensive evaluation for critique ability. We agree that it is essential to study multilingual critiques and intend to broaden CRITICEVAL to include other languages in future work. The following content briefly introduces our preliminary solution on how to achieve this goal.

Table 36: The computation cost of inference one LLM in CRITICEVAL.

| Dimensions | Cost of Test set ($) | Cost of Dev set ($) |
|---|---|---|
| **Feedback** | 4.21 | 5.09 |
| **Correction** | 2.11 | 2.67 |
| **Comparison** | 3.62 | 5.43 |
| **Overall** | 9.94 | 13.19 |

**Construct Multilingual CRITICEVAL**  Following the previous work [89], CRITICEVAL could be translated to various languages, especially low-resource languages, with human annotation for revising translation inaccuracies. the most direct way is to translate CRITICEVAL into various languages, with human annotation for revising translation inaccuracies.

**Evaluate Multilingual CRITICEVAL**  While the reliability of objective evaluation could be ensured, the reliability of subjective evaluation is limited by the multilingual capability of the judge model (GPT-4). We recommend back-translating multilingual critiques into English and evaluating them within English CRITICEVAL.

