# OpenReview forum: "CriticEval: Evaluating Large-scale Language Model as Critic"
_NeurIPS.cc/2024/Conference — NeurIPS 2024 poster_

### Official Review · Reviewer_xDZX · 2024-07-05

**Soundness:** 4
**Presentation:** 4
**Contribution:** 4
**Rating:** 7
**Confidence:** 4

**Summary:**

This work introduces a benchmark for using LLMs are critics.

The benchmark covers four settings:
1. Providing feedback
2. Correction of a response with/without feedback
3. Comparison of two responses for a given query
4. Providing meta-feedback (feedback on feedback)

**Strengths:**

Extensive experiments across models and setups.

**Weaknesses:**

For weakness I would repeat the limitations I list below.

**Questions:**

Did you consider structured generation?

Besides positional bias, did you explore any other biases such as bias to length, style or certain words?

Did you consider few-shot performance?

**Limitations:**

Two minor limitations:
- As far as I can tell structured generation was not used which has been shown to improve performance
- As far as I can tell few-shot performance was not considered.

---

> ### Author Rebuttal · Authors · 2024-08-07
>
> We really appreciate your valuable suggestions and insightful questions. We will address your concerns as follows.
>
> ---
>
> **Q1: Did you consider structured generation?**
>
> **A1:** We appreciate the reviewer’s attention to the details of our evaluation. Structured generation, such as JSON output, is relatively new but crucial for applications of LLMs, and we have not considered it yet. While the structured generation is not used, our conclusions are still reliable due to the same generation settings of all evaluated LLMs.
>
> The primary goal of our proposed CriticEval in the current stage is to construct a comprehensive and reliable benchmark for evaluating critique ability. We will supplement discussions about it in the Limitation. Thanks for your valuable suggestions.
>
> ---
>
> **Q2: Besides positional bias, did you explore any other biases such as bias to length, style or certain words?**
>
> **A2:** Yes, in addition to positional bias, we have investigated length bias, which is discussed in Section 5.2 (lines 198-200) and Appendix I. Figure 10 (Appendix I) reveals no significant correlation between the count of unique tokens and the Likert scores from GPT-4’s subjective evaluations across three critique dimensions [1,2], likely due to the human-annotated reference critiques used in the prompt.
>
> Regarding other potential biases, such as style and specific word preferences, our human annotations haven't observed substantial influence from these factors.
>
> ---
>
> **Q3: Did you consider a few-shot performance?**
>
> **A3:** Yes, we have studied the few-shot prompting strategy. However, our results demonstrate that few-shot prompting reduces performance across various LLMs. For example, when using 1-5 examples in objective feedback evaluations, we observed a significant decline in LLMs' performance as the number of examples increased.
>
> |Spearman Correlation|No Few-shot|1|2|3|4|5|
> |-|-|-|-|-|-|-|
> |**Llama-3-7B-Instruct**|**61.34**|58.13|54.25|52.99|53.23|50.11|
> |**InternLM2-20B-Chat**|**69.86**|66.26|64.99|63.32|60.33|61.72|
>
> This intriguing phenomenon may be due to the complexity of the critique task, where few-shot examples might impede the LLM’s understanding of the evaluated responses. Consequently, CriticEval currently does not utilize few-shot prompting by default. Given the emerging interest in critique ability research, we look forward to future works investigating advanced inference strategies to improve critique ability of LLMs.
>
> ---
>
> ### References
>
> [1] AlpacaEval: An Automatic Evaluator of Instruction-following Models
>
> [2] How Far Can Camels Go? Exploring the State of Instruction Tuning on Open Resources

---

> > ### Comment · Reviewer_xDZX · 2024-08-13
> >
> > Thanks for responding to my questions.

---

### Official Review · Reviewer_kJKF · 2024-07-12

**Soundness:** 3
**Presentation:** 2
**Contribution:** 2
**Rating:** 5
**Confidence:** 5

**Summary:**

This paper introduces CriticEval, a benchmark designed to comprehensively and reliably evaluate the critique ability of large language models (LLMs). It assesses critique capabilities across four dimensions (feedback, comparison, correction, and meta-feedback) and nine diverse task scenarios, using both scalar-valued and textual critiques for responses of varying quality. The benchmark was constructed using a human-in-the-loop pipeline, with initial critiques generated by GPT-4 and refined by human experts. CriticEval employs both objective metrics and subjective evaluation by GPT-4 with human-annotated reference critiques. Key findings from evaluating 35 LLMs include: GPT-4's high correlation with human judgments when using reference critiques, some open-source LLMs approaching closed-source models in performance, and insights into how critique difficulty varies by task type, response quality, and critique dimension.  CriticEval could be used as a comprehensive and reliable tool for assessing LLM critique capabilities.

**Strengths:**

- CRITICEVAL evaluates critique ability across multiple dimensions (feedback, comparison, correction, meta-feedback) and diverse task scenarios, providing a more holistic assessment than existing benchmarks.
- The benchmark combines GPT-4 generated critiques with human expert refinement and employs both objective metrics and subjective evaluation with human-annotated reference critiques. This approach ensures a more reliable and accurate evaluation of LLM critique abilities.
- The paper presents results from evaluating 35 open-source and closed-source LLMs, offering valuable insights into the current state of LLM critique capabilities. It reveals interesting relationships between critique difficulty and factors like task type, response quality, and critique dimension.

**Weaknesses:**

- The biggest concern of the dataset is that it relies heavily on GPT-4 for initial critique generation and evaluation. This could introduce a bias favoring models such as GPT-4 and models trained on GPT-4 distilled data. The human-in-the-loop process might not fully mitigate this bias, especially if annotators are influenced by GPT-4's initial outputs. A more diverse set of models or purely human-generated critiques for the benchmark could have provided a more neutral evaluation framework.
- The paper doesn't adequately address the scalability of CRITICEVAL for evaluating future language models. Additionally, the reliance on GPT-4 and human experts for evaluation might make it challenging for other researchers to fully reproduce or extend the benchmark.
- Insufficient analysis of failure modes: While the paper presents overall performance metrics, it doesn't delve deeply into the specific ways in which models fail at critique tasks. A more detailed error analysis could provide valuable insights into the limitations of current LLMs and guide future research more effectively.
- Lack of comparison to human performance: The paper doesn't provide a clear comparison between LLM performance and human performance on these critique tasks.

**Questions:**

How did you evaluate the human's performance during the benchmark construction phase?

**Limitations:**

Yes

---

> ### Author Rebuttal · Authors · 2024-08-07
>
> We really appreciate your valuable suggestions and insightful questions. We will address your concerns as follows.
>
> ---
>
> **Q1：... relies on GPT-4 for evaluation. This could introduce a bias favoring models such as GPT-4 ...**
>
> **A1:** Please refer to **"Global Response - Overcome Bias of GPT-4 Judge"** for more details. In summary, we supply the human-annotated critiques to improve GPT-'s reliability, and experimental results in the meta-feedback dimension have validated its reliability (Section 6.2).
>
> ---
>
> **Q2: ... a more diverse set of models or purely human-generated critiques provide a more neutral evaluation framework.**
>
> **A2:** As described in Appendix A.1, human annotators might be influenced by GPT-4' initial critiques. **We wish to emphasize that our decision to employ a human-in-the-loop rather than pure human annotation is motivated by two essential considerations (effectiveness and efficiency)**:
> #### **1. Human-written Critiques are Not Comprehensive (Effectiveness)**
> Our human annotation reveals that annotators might neglect some apparent or severe issues when writing critiques from scratch, consistent with findings in recent studies [1,2]. Neglecting issues usually leads to low-quality critiques. Specifically, experimental results in Table 2 of  **Global Response - Fine-grained Failure Modes** proves that missing issues in critiques lead to low subjective scores.
>
> In contrast, despite the possibility of generating inaccurate critiques, LLMs like GPT-4 offer more comprehensive and detailed critiques [1,2]. By revising LLM's errors by human experts, the final critiques could be more comprehensive and accurate, leveraging the strengths of both human annotators and LLMs [1,2].
>
> As detailed in Appendix G.5, our human annotations exhibit significant revisions (25.22%, 34.83%, and 48.37%) on GPT-4's initial critiques, effectively alleviating potential bias and noise.
> #### **2. Annotating Critique Task from Scratch Cost A Lot (Efficiency)**
> Writting critiques from scratch is a significant challenge [3,4]. For instance, Shepherd [4] incurred an annotation cost of 8\\$ per sample, leading to over 28,864\\$ and 1,350 work hours to annotate the entire CriticEval, which is unbearable for our project. Thus, we have to employ advanced LLMs to generate draft critiques, followed by human annotation.
>
> **GPT-4 is chosen because our preliminary studies indicate that it is the most reliable LLM for producing draft critiques, while other LLMs are much worse (Table 2). Consequently, a diverse set of LLMs introduces more noise in draft critiques, bringing more difficulties to human annotators.**
>
> In conclusion, the human-in-the-loop pipeline achieves the trade-off between annotation cost and quality. We promise to add these details to the revised paper to emphasize the motivation of using the human-in-the-loop pipeline. Thanks for your detailed review and insightful question.
>
> ---
>
> **Q3: The paper doesn't adequately address the scalability of CriticEval ... reliance on GPT-4 and humans make it challenging to reproduce or extend CriticEval**
>
> **A3:** Please refer to Section **"Global Response - Scalability and Cost"** for our explanations. Our explanations reveal that the cost of reproducing and extending CriticEval is comparable to established benchmarks like AlpacaEval and AlpacaFarm.
>
> ---
>
> **Q4: Insufficient analysis of failure modes ...**
>
> **A4:** Thanks for your very valuable suggestions. We have conducted coarse-grained and fine-grained analyses of failure modes before. Due to space limitations, we only included the coarse-grained analysis in Section 6.6.
>
> To address your concern, we have supplemented our findings in **"Global Response - Fine-grained Failure Modes."** This supplementary material reveals intriguing phenomena. For example, the most frequent failure modes are missing errors, lacking effective comparison and worse revision than references for feedback, comparison and correction. Besides, inaccurate critiques usually lead to lower subjective scores. The revision that does not follow suggestions in feedback usually leads to the worst performance.
>
> We really appreciate your insightful suggestions and the opportunity to address them.
>
> ---
>
> **Q5: Lack of comparison to human performance ...**
>
> **A5:** As described in Q1-A1, collecting purely human-annotated critiques is challenging. Thus, human performance is not recorded in the current submission.
>
> We agree that human performance is valuable, and we are urgently working on annotating human-generated results on the test set. All annotators have an undergraduate level of education. Due to the massive workload of human annotation, subjective annotations are ongoing, and the objective scores have been provided below:
>
> ||Feedback (Corr.)|Comparison (Acc)|Correction (Pass Rate)|Meta-Feedback (Corr.)|
> |-|-|-|-|-|
> |GPT-4|63.54|57.33|69.67|62.9|
> |Human|**67.69**|**60.67**|**75.69**|**71.36**|
>
> Human performance slightly outperforms GPT-4 on four critique dimensions in the objective split. We promise to add full human performance results in the revised paper.
>
> Thanks for your valuable suggestions.
>
> ---
>
> **Q6: How did you evaluate human performance during benchmark construction?**
>
> **A6:** As briefly described in Appendix G.1, for the textual critiques, the supervisor's (authors) review and revise mechanism ensures the quality of human annotation meets our expectations. For the scalar-based critiques, the supervisors conducted a 5% sample inspection. If the error rate exceeds the threshold, annotators are asked to revise their work until the error rate is lower than the threshold. Besides, the inner-agreement among annotators are computed to make sure their judgments are consistent.
>
> ---
> ### References
>
> [1] LLM Critics Help Catch LLM Bugs (OpenAI)
>
> [2] Self-critiquing models for assisting human evaluators (OpenAI)
>
> [3] AlignBench: Benchmarking Chinese Alignment of Large Language Models
>
> [4] Shepherd: A Critic for Language Model Generation

---

> ### Author Response · Authors · 2024-08-11
> **The Complete Human-level Performance**
>
> Dear Reviewer kJKF,​
>
> We would like to thank you for the thoughtful and constructive feedback and appreciate that you agree on the strengths of our paper.​
>
> We provided details and analysis to address your concerns during the rebuttal. In this response, we complete the human performance annotation of the subjective tasks on the CriticEval test set, and the overall human-level performance is shown as follows. ​
>
> **Note that the cohort and corresponding set of human critiques do not represent the best possible human performance; instead, they represent the capability of annotators selected for this human performance annotation of the CriticEval test set.**
>
> ||Feedback Sub.|Feedback Obj.|Comparsion Sub.|Comparison Obj.|Correction Sub.|Correction Obj.|
> |-|-|-|-|-|-|-|
> |GPT-4|**7.84**|63.54|**7.89**|57.33|**7.69**|69.67|
> |Human|5.61|**67.69**|5.22|**60.67**|6.63|**75.69**|
>
> > As described in Section 5, objective scores (Obj.) for feedback, comparison, and correction are correlation, accuracy, and pass rate. Subjective scores (Sub.) for feedback, comparison, and correction are Likert scores (1-10) generated by GPT-4 with human-annotated critiques as references.
>
> **`Experimental Results:`** As shown in the Table above, it can be found that the human level significantly outperforms GPT-4 on the objective task, while it is inferior to GPT-4 on the subjective evaluation.
>
> ---
>
> Then, we conduct the Quantitative and Qualitative Analysis to understand the human performance in subjective evaluation.
>
> ### **Quantitative Analysis**
>
> The distribution of failure modes for humans and GPT-4 is shown in the following tables. The numbers in the tables indicate the frequencies of error types in critiques. The descriptions of failure modes are placed in Table 1 of **"Global Response - Fine-grained Failure Modes."**
>
> |Feedback|E1|E2|E3|E4|E5|E6|Other|
> |-|-|-|-|-|-|-|-|
> |GPT-4|17.99|18.71|**16.37**|**15.83**|**10.07**|**14.93**|6.12|
> |Human|**21.18**|**24.48**|11.36|15.27|9.06|12.13|**6.52**|
>
> |Comparison|E1|E2|E3|E4|E5|E6|E7|E8|Other|
> |-|-|-|-|-|-|-|-|-|-|
> |GPT-4|16.67|11.02|**11.29**|**15.59**|**4.3**|**6.99**|19.35|**12.1**|**2.69**|
> |Human|**19.71**|**15.29**|7.65|9.51|3.82|4.8|**24.9**|11.67|2.65|
>
> |Correction|E9|E10|E11|Other|
> |-|-|-|-|-|
> |GPT-4|23.46|**43.83**|21.6|**11.11**|
> |Human|**29.38**|36.88|**25**|8.75|
>
> **`Experimental Results:`** As for the feedback and comparison dimensions, the distribution of E1 (missing issues), E2 (missing suggestions or low-quality suggestions), and E7 (insufficient analysis) in human-written critiques is significantly higher than that of GPT-4. In contrast, the distribution of other errors (most are mistakes in critiques) is much lower.
>
> As for the correction dimension, human annotators usually do not follow suggestions in the provided feedback (E9) and generate additional errors (E11). Through communicating with the annotators, we notice that the primary cause of this issue is that some tasks require domain-specific knowledge. The lack of this knowledge among annotators leads to lower-quality corrections. This phenomenon aligns with the findings of recent work [1].
>
> **`Conclusion:`** The human-written critiques are often less comprehensive than GPT-4, significantly reducing the quality. In contrast, the mistakes in human-written critiques are significantly less than that of GPT-4. This phenomenon is consistent with our preliminary study and recent findings [1], further proving the reasonableness of using the human-in-the-loop pipeline to collect comprehensive and accurate critiques (**as described in Q2-A2**).
>
> ### **Qualitative Analysis**
>
> We inspect the human critiques in the subjective evaluation. We notice that human annotators generally write fewer comments than LLMs, and comments are usually general and brief. Besides, many tasks involve domain-specific knowledge that humans may lack, but GPT-4 excels in (albeit with potential hallucinations).
>
> ### **Summary**
>
> We provide the human-level performance for the CriticEval test set. The human performance is better than GPT-4 on the objective split, while it is inferior to GPT-4.
>
> **`We wish to emphasize that:`** Although our first submission lacks **pure** human performance, our reference critiques obtained through the human-in-the-loop pipeline could serve as very close candidates for the best possible human performance. During the subjective evaluation, the quality score of reference critique is anchored to 8 (overall 1-10 score range), serving as a relative scoring pivot, which is helpful in analyzing the performance gap between LLMs and human.
>
> We will supplement experimental results and analysis into our revised paper.
>
> We hope these responses address the concerns. We are happy to discuss further comments and suggestions. If the reviewer finds our response adequate, we would appreciate it if the reviewer considers raising the score.
>
> ### **Reference**
> [1] LLM Critics Help Catch LLM Bugs
>
> ---
>
> Best Regards,
>
> Submission 7061 Authors

---

> > ### Comment · Reviewer_kJKF · 2024-08-13
> >
> > I appreciate the author's response, which slightly resolves my concerns. However, I still have concerns about the bias in pure GPT-4 assisted annotation, which is a fundamental limitation of the methodology. Given the new results, I would like to slightly raise my overall score from 4 to 5.

---

> > > ### Author Response · Authors · 2024-08-13
> > > **Response to Reviewer**
> > >
> > > Dear Reviewer kJKF,
> > >
> > > We really appreciate your valuable feedback during the review phase, and thank you for raising the overall score assigned to our paper.
> > >
> > > Although we have made our best efforts, the bias of GPT-4 may still not be completely eliminated. The human-in-the-loop annotation pipeline is the trade-off solution for collecting comprehensive and accurate critiques by considering scalability and reliability. We will discuss it in more detail in the Limitation Section of our revised paper.
> > >
> > > If you have any further questions or concerns, we are more than happy to address them!
> > >
> > > Best Regards,
> > >
> > > Submission 7061 Authors

---

### Official Review · Reviewer_2SKd · 2024-07-13

**Soundness:** 3
**Presentation:** 4
**Contribution:** 4
**Rating:** 8
**Confidence:** 5

**Summary:**

The paper addresses the need for a comprehensive evaluation of the critique ability of large language models (LLMs) for self-improvement and alignment with human outcomes. Current evaluation methods are critiqued for their limited scope and reliability. The authors propose CRITICEVAL, a benchmark designed to evaluate LLMs across four critique dimensions: feedback, comparison, correction, and meta-feedback, covering nine diverse task scenarios including NLP, alignment, and reasoning tasks. While CRITICEVAL incorporates human-annotated references to enhance reliability, it heavily relies on GPT-4 for evaluations, raising concerns about generalisability across other LLMs. Evaluations of 35 LLMs demonstrate CRITICEVAL's effectiveness but also highlight the inherent difficulties in critiquing complex tasks and the inverse relationship between critique and response quality. Although promising, the reliance on human annotation and the potential biases introduced by using a single LLM for baseline evaluations could limit its broader applicability. The release of datasets and evaluation tools is a positive step towards fostering further research.

**Strengths:**

The work is a very comprehensive study on the critical evaluation problem and provides insight for those who would be in situations where they would want to build better LLM pipelines for use-cases where accuracy is important for fit for use.

The exploration of 35 different LLM providers more diversity and incorporating human feedback for quality ranking, it allows for wider evaluation than prior benchmarks.

I believe the work is well presented, written and argued. It is a very daunting project to get all the pieces together. It is harder, ironically, to evaluate it just as a paper because of all the moving parts, but the authors evaluations across the LLMs are appreciated.

**Weaknesses:**

Even with the mention of incorporating Chinese, I believe there should be a serious discussion on how such evaluation pipelines and datasets would function for low-resource languages (Chinese is not one). Many of the large LLMs have multilingual capabilities and their generation of responses are more likely to have higher error rates and as such being able to do critical evaluation in such use-cases is important.

The computational cost and real cost of setting up the CRITICEVAL pipelines should be spelled out.

It is not clear if IRB/Ethical clearance was obtained as the checklist response states that IRB *would* be easy to obtain. This is a concern as such I will be referring this for ethics review. It is appreciated that information was provided in Appendix B and G, but whether IRB was obtained or not should have a clear YES or NO statement.

**Questions:**

1. How do you protect against errors in the underlying tasks datasets. The explosion of many task evaluation sets, even with the best intentions of researchers means that we have compounding effects of source errors (e.g. in Translation), how would this affect your work and how would you mitigate against it?
2. Ultimately how many people are involved and computational cost of such an exercise as this affects replication ability of this study to cover some of the limitations you hilighted?

**Limitations:**

The limitations are written in Appendix A. They are clear.

---

> ### Author Rebuttal · Authors · 2024-08-07
>
> We really appreciate your valuable suggestions and insightful questions. Since some of your questions are similar to those of other reviewers, we will describe them in more detail in **Global Response**. We also provide some summary of these questions under your review comment to make our explanation clear.
>
> ---
>
> **Q1: ... it heavily relies on GPT-4 for evaluations ... The potential biases introduced by using a single LLM for evaluations limit broader applicability.**
>
> **A1:** Please refer to **"Global Response - Overcome Bias of GPT-4 Judge"** for our explanations. In summary, experimental results in the meta-feedback dimension prove the reliability of GPT-4 as a judge, while other LLMs are much worse. Besides, to further improve the reliability of subjective evaluation, GPT-4 is equipped with our human-annotated critiques, which is a trade-off solution for scalability and reliability of subjective evaluation.
>
> ---
>
> **Q2: ... I believe there should be a serious discussion on how such evaluation pipelines and datasets would function for low-resource languages ...**
>
> **A2:** Please refer to Section **"Global Response - Multilingual Support"** for our explanations. In summary, CriticEval could be extended to other low-resource languages at an affordable cost using translation and further human labor.
>
> ---
>
> **Q3: The computational cost and real cost of setting up the CRITICEVAL pipelines should be spelled out.**
>
> **A3:** Thanks for your valuable suggestion. Please refer to **"Global Response - Scalability and Cost"** for more details about the computation and construction cost of CriticEval. In summary, the construction and evaluation cost in CriticEval is comparable to established benchmarks, like AlpacaEval and AlpacaFarm.
>
> ---
>
> **Q4: It is not clear if IRB/Ethical clearance was obtained as the checklist response states that IRB would be easy to obtain ...**
>
> **A4:** Thanks for your highly meticulous review. We apologize for any confusion and misunderstanding of the meanings between the human subjects and the crowdsourcing. CriticEval is annotated by crowdsourcing, and no human subjects are studied in our work. Besides, as described in Appendix B, the hourly wage of crowdsourcing is much higher than that of Amazon Mechanical Turk. Thus, our work does not violate the NIPS Code of Ethics.
>
> Thank you for bringing this to our attention.
>
> ---
>
> **Q5: How do you protect against errors in the underlying tasks datasets ... how would this affect your work and how would you mitigate against it?**
>
> **A5:** We agree that some underlying datasets contain inaccuracies that may lead to compounding effects during evaluation. For example, we have observed some incorrect solutions and reasonings for mathematics and coding questions during our human evaluation process.
>
> In our work, to mitigate the effects of such errors, human annotators are instructed to examine each question meticulously, the provided golden answers and the evaluated responses and critiques. They are asked to exclude instances where the golden answers or questions are flawed or incorrect, like wrong solutions to mathematics and coding questions.
>
> We promise to supplement more details and cases in the Appendix. Thank you for raising this critical issue, and we appreciate the opportunity to strengthen our work through your valuable comments.
>
> ---
>
> **Q6: Ultimately how many people are involved and computational cost of such an exercise as this affects replication ability of this study to cover some of the limitations you hilighted?**
>
> **A6:** Please refer to Section **"Global Response - Scalability and Cost"** for more details about our explanations. Our explanations reveal that the cost of reproducing and extending CriticEval is comparable to established benchmarks like AlpacaEval and AlpacaFarm, ensuring replication ability for the research community.

---

> > ### Comment · Reviewer_2SKd · 2024-08-13
> > **Thank you for your resonses**
> >
> > Thank you for your responses.

---

### Official Review · Reviewer_Hopm · 2024-07-13

**Soundness:** 4
**Presentation:** 4
**Contribution:** 4
**Rating:** 9
**Confidence:** 4

**Summary:**

This study constructs a comprehensive framework for LLM-based evaluation, encompassing data construction, human/machine annotation, and result analysis. It defines a single evaluation framework that covers various tasks and response types. Although previous studies have evaluated different tasks and response types, they often lacked comprehensive analysis, making it difficult to understand the relationships between various factors during evaluation. This research aims to address this gap. Additionally, it examines the reliability of evaluations based on the types of judge models and target models, as well as the importance of human annotation data.
Through extensive experiments and comprehensive analysis, this study demonstrates consistent trends across various tasks and highlights the importance of utilizing human-annotated data in evaluations.

**Strengths:**

1. The proposed framework allows for the evaluation of various tasks and response types.
2. Through various experiments, the high evaluation capability of closed-source LLMs, regardless of response type and task, is confirmed.
3. The importance of human-annotated data is evident, showing consistency regardless of the evaluation model.
4. Additionally, extensive experiments analyze various aspects such as response quality, difficulty, and the capability of reward models as judge models.

**Weaknesses:**

1. One important metric for evaluation models, the ability to revise its generation with critique, is excluded. As mentioned in the paper, there is active research on improving generation performance through LLM’s self-feedback during the inference stage. It is necessary to evaluate each model’s ability to revise existing outputs when provided with critique.
2. Although this study addresses the translation task, it excludes the critique ability in other multilingual contexts. The critique ability of LLMs in various languages other than English is crucial for ensuring diversity, and additional data collection on this aspect seems necessary.

**Questions:**

1. What is the pattern of score changes when using feedback from other LLMs in Table 5, based on the quality of the feedback from these other LLMs?

**Limitations:**

Yes, this study clearly mentions its limitations in the conclusion.

---

> ### Author Rebuttal · Authors · 2024-08-07
>
> We really appreciate your valuable suggestions and insightful questions. Your questions are addressed as follows:
>
> ---
>
> **Q1: One important metric for evaluation models, the ability to revise its generation with critique, is excluded ... It is necessary to evaluate each model’s ability to revise existing outputs when provided with critique.**
>
> **A1:** We appreciate the reviewer's insightful comment regarding evaluating the ability to revise generation with critiques. We want to clarify that this capability is a central aspect of our work and has been thoroughly evaluated in our CriticEval benchmark, referred to as the correction ability ($CR$) in our paper. We acknowledge the reviewer's suggestion that "revision" is more precise than "correction" to describe this feature. We will update the terminology accordingly in the revised version of our paper, replacing "correction" with "revision."
>
> To further clarify the evaluation of the revision ability in our study, we highlight the following points:
> 1. Data Collection and Metric for Revision Evaluation:
> * Section 4.3 (line 155) details the process of collecting reference revisions.
> * Sections 5.1 (line 189) and 5.2 (lines 194-196) outline the objective and subjective metrics used to evaluate the model’s ability to revise generations.
>
> 2. Evaluating Revision Ability:
> Our evaluation of LLMs’ revision ability encompasses two scenarios: using golden feedback as input and using feedback generated by other LLMs.
> * **Golden Feedback as Input:** Subjective scores are computed by comparing the model’s revisions to human-annotated revisions, and the reliability of this subjective evaluation is substantiated in Section 6.2 (Table 4 and lines 240-246).
> * **Feedback from Other LLMs as Input:** We also conducted experiments to assess how LLMs perform with feedback produced by others, including themselves. As detailed in Section 6.5 (lines 311-326), we evaluated the average performance of the LLMs using three types of feedback: human-annotated, empty, and self-generated. The results indicate that LLMs face challenges in self-improvement, particularly in complex reasoning tasks.
>
> We hope this clarification addresses the reviewer’s concern and demonstrates the comprehensive nature of our evaluation approach. We are grateful for the opportunity to enhance the terminology and presentation of our findings, which we believe will further clarify the contributions of our work.
>
> ---
>
> **Q2: ... it excludes critique ability in other multilingual contexts ... and additional data collection on this aspect seems necessary.**
>
> **A2:** Please refer to Section "Global Response - Multilingual Support" for our explanations.
>
> In summary, our work is the first step to constructing a comprehensive and reliable evaluation. The multilingual feature is not considered because we mainly focus on the critique ability over common tasks in the current stage. But, the cost of extending CriticEval to other languages is affordable.
>
> ---
>
> **Q3: What is the pattern of score changes when using feedback from other LLMs in Table 5, based on the quality of the feedback from these other LLMs?**
>
> **A3:** We apologize for any confusion caused by Table 5 and the related content in Section 6.3. Since feedback and revision performance are entangled, it is essential to investigate how the quality of feedback affects the performance of LLM's revision ability.
>
> To explore this, we prompted the InternLM2-20B-Chat and Llama2-70B-Chat models to revise responses from CriticEval using three types of feedback with varying quality levels. To enhance clarity, we have restructured Table 5 as below: the first table presents results from the objective split of CriticEval, and the second table displays results from the subjective split.
>
> | Revision Model |Source of Feedback|Feedback Quality (1-10)|Objective Revision Performance|
> |-|-|-|-|
> |InternLM2-20B|Llama2-70B|2.24|7.15|
> |InternLM2-20B|InternLM2-20B|7.53|10.33|
> |InternLM2-20B|Human|**8**|**50.5**|
> |Llama2-70B|Llama2-70B|2.24|5.33|
> |Llama2-70B|InternLM2-20B|7.53|12.47|
> |Llama2-70B|Llama2-70B|**8**|**42.43**|
>
> | Revision Model |Source of Feedback|Feedback Quality (1-10)|Subjective Revision Performance|
> |-|-|-|-|
> |InternLM2-20B|Llama2-70B|5.63|5.71|
> |InternLM2-20B|InternLM2-20B|6.85|5.8|
> |InternLM2-20B|Human|**8**|**7.48**|
> |Llama2-70B|Llama2-70B|5.63|5.54|
> |Llama2-70B|InternLM2-20B|6.85|6.32|
> |Llama2-70B|Llama2-70B|**8**|**7.11**|
>
> These results indicate that as the quality of the feedback increases, both the objective and subjective revision performance improves. This trend demonstrates that higher-quality feedback is associated with more effective revisions.

---

### Author Rebuttal · Authors · 2024-08-07

# Global Response
We thank all the reviewers for their insightful and valuable comments. Below, we will address some common questions and concerns of reviewers.

---

## **1. Overcome Bias of GPT-4 Judge (kJKF, 2SKd)**
To mitigate bias of GPT-4 as a judge, our work has made two efforts:
### **1.1 In Construction Phase**
CriticEval collects high-quality critiques by human-in-the-loop pipeline, where human annotators review and revise GPT-4's initial critiques. **The human-annotated revisions for GPT-4's initial critiques are significant (25.22%, 34.83%, and 48.37%), as detailed in Appendix G.5.** Besides, the ground-truth answers for challenging tasks are fed to GPT-4 for high-quality draft critiques for GPT-4 (Appendix G.4).

**Furthermore, we introduce the meta-feedback critique dimension to independently validate GPT-4's evaluation reliability as a judge (Section 6.2 and Table 2). It is purely annotated by humans to evaluate critiques generated by various LLMs, including GPT-4, GPT-3.5-turbo, and two CritiqueLLMs, without the bias of GPT-4.** Our results, presented in Tables 2, 3, and 4, show that only GPT-4 with human-annotated critiques achieves a very high correlation with human judgment in meta-feedback, while the others are far behind GPT-4. These results justify GPT-4 as a reliable judge in subjective evaluations and a diverse set of LLMs as judges introduce more noise in draft critiques, bringing more difficulties to human annotators.
### **1.2 In Evaluation Phase**
Human-annotated critiques serve as reference critiques for subjective evaluation, ensuring GPT-4 does not prefer specific LLMs. Our human annotators haven't observed clear bias towards GPT-4 or LLMs fine-tuned on GPT-4's critiques.

Although we have made our best efforts, the bias of GPT-4 may still not be completely eliminated. **We emphasize that our evaluation method is a trade-off solution considering the scalability and reliability of the CriticEval subjective evaluation. We look forward to subsequent work that can address and resolve this issue.**

---

## **2. Scalability and Cost (kJKF, 2SKd)**
We notice that reviewers kJKF and 2SKd inquire about the scalability and the cost of CriticEval. Thus, we provide the cost of constructing one task and inferencing one LLM in CriticEval.
### **2.1 Construction Cost**
The construction cost consists of two parts:
1. **Collect Evaluated Responses for All Tasks**
* Open-source LLMs: a GPU server with 8 A100 (80G) cards is used to generate evaluated responses, and the total GPU hours are 4.26 hours, approximately 82.88\\$ (refer to the price of Alibaba Cloud).
* Closed-source LLMs: the average cost for each LLM is 0.89\\$.
2. **Generate and Revise GPT-4 Critiques**
|For Each New Task|Cost (\\$)|Time (hour)|
|-|-|-|
|Generate Critiques (GPT-4)|3.09|-|
|Human Annotation|303.53|53.34|
|Overall|306.62|53.34|

The cost of the human annotation is computed under these settings:
* Four human annotators (3 annotators and one supervisor)
* 5.69\\$ hourly wage for each annotator (Appendix G.1)
* Average 400 samples in one task.

In summary, the human annotation cost for one new task is affordable [1].
### **2.2 Average Computational Cost for One LLM**
|Dimension|Cost of Test set ($)|Cost of Dev set ($)|
|-|-|-|
|Feedback|4.21|5.09|
|Correction|2.11|2.67|
|Comparison|3.62|5.43|
|Overall|**9.94**|**13.19**|

The overall cost of the test and dev set is 13.19+9.94=23.13\\$, comparable to the evaluation cost on the AlpacaEval benchmark (5-15\\$) [2]. Note that these costs are essential for CriticEval, as they guarantee the reliability of critique evaluation. We promise to add these details to the Appendix of our revised submission.

---

## **3. Multilingual Support (Hopm, 2SKd)**

The primary goal of CriticEval in the current stage is to construct a reliable and comprehensive evaluation for critique ability. We agree that it is essential to study multilingual critiques and intend to broaden CriticEval to include other languages in future work, as described in Section 7 (lines 375-376).

Reviewer 2SKd suggests including a serious discussion on how to achieve this goal. The following content briefly introduces our preliminary solution.
### **3.1 Construct Multilingual CriticEval**

Following the previous work [3], CriticEval could be translated to various languages, especially low-resource languages, with human annotation for revising translation inaccuracies.
### **3.2 Evaluate Multilingual CriticEval**
While the reliability of objective evaluation could be ensured, the reliability of subjective evaluation is limited by the multilingual capability of the judge model (GPT-4). We recommend back-translating multilingual critiques into English and evaluating them within English CriticEval.

---

## **4. Fine-grained Failure Modes (kJKF)**

Reviewer kJKF offered valuable suggestions regarding a fine-grained analysis of failure modes. Coarse-grained failure modes have been analyzed in Section 6.6, and the fine-grained analysis is provided in the PDF file.

The analysis reveals that the most frequent failure modes are missing errors, lacing effective comparison analysis, and worse revision than references for feedback, comparison, and correction dimensions, respectively. Besides, inaccurate critiques usually lead to lower subjective scores, such as missing crucial errors and incorrect analysis content. The revision that does not follow suggestions in feedback usually leads to the worst performance.

---

### References
[1] AlpacaFarm: A Simulation Framework for Methods that Learn from Human Feedback

[2] Alpacaeval: An automatic evaluator of instruction-following models

[3] Language Models Are Multilingual Chain-of-Thought Reasoners

---

### Decision · Program_Chairs · 2024-09-25

**Decision:**

Accept (poster)

**Comment:**

This work presents an analysis of the evaluation capabilities of large language models (LLMs) at four different tasks: (1) pointwise evaluation of generated responses with textual and numerical assessments, (2) response correction, (3) pairwise response comparison; (4) meta-feedback – the evaluation of generated textual evaluations. Human annotations are collected on 9 specific application scenarios to facilitate the evaluation of the 4 evaluation tasks. A comprehensive evaluation is then performed on 35 LLMs using the collected human annotations as gold references with the assistance of GPT-4.

The reviewers mostly agree on the following strengths and contributions of this work:
1. The importance of new human annotation collection in facilitating the evaluation of LLMs’ evaluation capabilities on the four tasks studied.
2. The comprehensiveness of the evaluation performed, covering 35 LLMs, 9 application scenarios, and 4 evaluation tasks.
3. The analysis performed in this paper is thorough and offers interesting insights.

The major concerns the reviewers raised include:
1. The reliance on GPT-4 in the evaluation pipeline, considering GPT-4 may not be a reliable and unbiased evaluator.
2. Unclarity of the cost of the evaluation process.
3, Lack of discussion of multilingual evaluation.
During the rebuttal period, the authors provided responses that clarified some of the above concerns. Specifically, regarding the reliance on GPT-4, this work has already performed a meta-evaluation of GPT-4 regarding performing subjective evaluations in Section 6.2, which provides an assessment of its reliability. However, the highest correlation achieved between GPT-4 and human annotations is around 66.2, which may not be high enough to demonstrate that GPT-4 is a robust evaluator. That said, the use of GPT-4 is largely justified because it can greatly improve the evaluation efficiency.

This work could be improved by the following:
1. A clearer and more thorough discussion of the limitations and risks of using GPT-4 as the judge in evaluation.
2. Clear description and analysis of the human annotation quality. One major contribution of this work is the collected human annotation,
 and it is of great importance that the human annotations are reliable. However, there is a lack of quantitative analysis of the human annotation quality, such as the inter-annotator agreement rate of pointwise scoring, etc. These statistics are important to demonstrate the quality of the collected annotations. Related work such as LLMBar [1] put a great emphasis on this reliability since it determines whether the evaluation performed on the collected annotations is reliable.

[1] Zeng, Zhiyuan, et al. "Evaluating Large Language Models at Evaluating Instruction Following." ICLR